# Navigating Noise: A Study of How Noise Influences Generalisation and Calibration of Neural Networks

**Martin Ferianc**[*]                                                                                   *martin.ferianc.19@ucl.ac.uk*
*Department of Electronic and Electrical Engineering*
*University College London*

**Ondrej Bohdal**[*]                                                                                           *ondrej.bohdal@ed.ac.uk*
*School of Informatics*
*University of Edinburgh*

**Timothy Hospedales**                                                                                      *t.hospedales@ed.ac.uk*
*School of Informatics*
*University of Edinburgh*
*Samsung AI Center Cambridge*

**Miguel Rodrigues**                                                                                         *m.rodrigues@ucl.ac.uk*
*Department of Electronic and Electrical Engineering*
*University College London*

**Reviewed on OpenReview:** `https://openreview.net/forum?id=zn3fB4VVF0`

## Abstract

Enhancing the generalisation abilities of neural networks (NNs) through integrating noise such as MixUp or Dropout during training has emerged as a powerful and adaptable technique. Despite the proven efficacy of noise in NN training, there is no consensus regarding which noise sources, types and placements yield maximal benefits in generalisation and confidence calibration. This study thoroughly explores diverse noise modalities to evaluate their impacts on NN's generalisation and calibration under in-distribution or out-of-distribution settings, paired with experiments investigating the metric landscapes of the learnt representations across a spectrum of NN architectures, tasks, and datasets. Our study shows that AugMix and weak augmentation exhibit cross-task effectiveness in computer vision, emphasising the need to tailor noise to specific domains. Our findings emphasise the efficacy of combining noises and successful hyperparameter transfer within a single domain but the difficulties in transferring the benefits to other domains. Furthermore, the study underscores the complexity of simultaneously optimising for both generalisation and calibration, emphasising the need for practitioners to carefully consider noise combinations and hyperparameter tuning for optimal performance in specific tasks and datasets.

## 1 Introduction

Neural networks (NNs) have demonstrated remarkable capabilities across various tasks, yet they often grapple with overfitting to training data, resulting in suboptimal generalisation performance on unseen samples (Srivastava et al., 2014; Bishop, 1995; Sietsma & Dow, 1991). Addressing this issue, conventional techniques such as weight decay (Krogh & Hertz, 1991) and early stopping (Prechelt, 2002) have been employed to regularise NN training. Alongside these methods, the introduction of noise during the NN's training has

---
[*]Joint first authors.

emerged as a potent strategy to enhance generalisation (Sietsma & Dow, 1991; Neelakantan et al., 2017; Camuto, 2021; Kukačka et al., 2017). The concept of noise injections refers to the deliberate introduction of artificial perturbations into different aspects of NN training. Note that this is distinct from the concept of noise in the data itself which originates from the data collection process (Song et al., 2022). Diverging from weight decay and early stopping that modulate the model's search within the hypothesis space, noise injections embrace randomness during training, fostering exploration of a broader array of representations (He et al., 2019). The appeal of noise injections extends further due to their versatile applicability across diverse tasks, datasets, and NN architectures. These attributes establish noise injections as a convenient approach for enhancing NN's generalisation.

In addition to generalisation, confidence calibration is a desirable model property, especially in safety-critical applications where confidence scores must be aligned with the model's accuracy to make informed decisions (Guo et al., 2017). Empirically, noise injections have been shown to improve confidence calibration by improving the generalisation of the NNs in previously unseen circumstances and inherently reducing overconfidence in predictions (Guo et al., 2017; Müller et al., 2019; Hendrycks et al., 2020; Zhang et al., 2018; Gal & Ghahramani, 2016). However, the relationship between generalisation and calibration is not straightforward, and the two properties are often at odds with each other (Guo et al., 2017).

Various noise injection methodologies have been proposed, encompassing **activation** techniques such as Dropout (Srivastava et al., 2014; Gal & Ghahramani, 2016) and Gaussian Dropout (Kingma et al., 2015), **weight** noises such as DropConnect (Wan et al., 2013) or additive Gaussian noise (Blundell et al., 2015), **target** methods such as label smoothing (Szegedy et al., 2016), **input-target** strategies exemplified by MixUp (Zhang et al., 2018), **input** modifications such as AugMix (Hendrycks et al., 2020) or the standard horizontal flipping and center cropping (Krizhevsky et al., 2009), **model** approaches including weight perturbation (Ash & Adams, 2020), and **gradient** perturbations involving Gaussian noise (Neelakantan et al., 2017). Despite the diversity of these techniques, comprehensive and fair comparisons are scarce, leaving a gap in understanding which approach is helpful for specific datasets, tasks and models in conjunction with generalisation and calibration.

This study aims to systematically and comprehensively investigate the effects of widely used noise injection methods on NN generalisation and calibration across multiple datasets, tasks, and architectures. This exploration is predicated on the premise that while generalisation focuses on reducing overfitting and improving the model's predictive accuracy, calibration deals with aligning the model's confidence with its actual performance. Rather than focusing on improving state-of-the-art performance, we aim to provide a holistic view of the effects of noise injections on NNs' generalisation and calibration for the benefit of practitioners. To this end, we present the following contributions:

1. The first systematic empirical investigation into the impact of noise injections on NN generalisation and calibration across diverse datasets, tasks and NN architectures. Our exploration extends to evaluation under in-distribution (ID) and out-of-distribution (OOD) scenarios and their transferability across architectures and datasets.

2. A methodological framework for simultaneously combining various noise injection approaches.

3. Visualisation of the learnt representation landscape across noises, jointly comparing calibration and generalisation performance.

Our investigation reveals that certain types of noise aid in generalisation by introducing robustness against overfitting and variability in data and potentially improve calibration by mitigating overconfidence in predictions. The findings show that AugMix, weak augmentation and Dropout prove effective across diverse tasks, emphasising their versatility. Task-specific nuances in noise effectiveness, such as AugMix's superiority in computer vision (CV), Dropout in natural language processing (NLP) and Gaussian noise in tabular data regression, highlight the need for tailored approaches. Combining noises, careful hyperparameter tuning, and task-specific considerations are crucial for optimising NN's performance. Our code is publicly available at `https://github.com/martinferianc/noise`.

## 2 Related Work

In this study, we consider artificial addition of noise into various facets of NN training – including **input**, **target**, **input-target**, **activations**, **weights**, **gradients**, and **model** parameters. The noise application is denoted by $\alpha_{<\text{place}>}(\cdot, \delta)$, where $\alpha_{<\text{place}>}$ is the noise application methodology which can be executed at different places, e.g. $\alpha_{\text{input}}$ for input noise, $\alpha_{\text{target}}$ for target noise along with $\cdot$ arbitrary arguments, depending on the noise injection methodology. For example, under this definition, additive input Gaussian noise samples a Gaussian and adds it to the input $x$ as $\alpha_{\text{input}}(x, \delta) = x + \epsilon; \epsilon \sim \mathcal{N}(0, \sigma^2)$, where $\sigma^2$ is the hyperparameter in $\delta$. Note that this study focuses on *artificial* noise injections, which are purposely introduced during training, and not on *natural* noise, inherent in the data, e.g. label noise in the targets where the classifying label is incorrect. The natural noise needs to be addressed separately and we refer the reader to Song et al. (2022) for a review of strategies for learning with noisy labels. Under different noise placements, we review several noise injection strategies. The review focused on the most fundamental noise injection methodologies, which constitute the building blocks of more complex approaches and represent the noise injection category.

**Input Noise**: Pioneering work by Sietsma & Dow (1991) demonstrated the benefits of training with added input Gaussian noise, while Bishop (1995) established its linkage to regularisation in the least squares problems. In CV, weak augmentation, such as random cropping and horizontal flipping, has improved generalisation (Krizhevsky et al., 2009). AugMix, domain-specific to CV, applies a sequence of image processing operations to the input, bolstering robustness in OOD settings. From the adversarial robustness domain, ODS augments inputs conditioned on the prediction, aiming to diversify the inputs (Tashiro et al., 2020). **Target Noise**: Label smoothing (Pereyra et al., 2017) softens the one-hot classification targets by replacing the targets with a categorical distribution with the most mass on the correct class and the rest spread across the other classes through the addition of constant uniform noise, effectively improving NN's robustness (Müller et al., 2019). This differs from label noise, where the entire probability mass is on an incorrect class, and the NN must learn to ignore the errors (Song et al., 2022). **Input-Target Noise**: Variants of MixUp have exhibited efficacy in augmenting both generalisation and calibration (Zhang et al., 2018; Müller et al., 2019; Guo et al., 2019; Yao et al., 2022; Guo et al., 2017). MixUp adds noise to both the input and the target via linear interpolation between two samples and their targets, while CMixUp expands this approach to regression problems. **Activation Noise:** Widespread activation noise includes Dropout or Gaussian noise injections. Dropout (Srivastava et al., 2014; Noh et al., 2017) randomly deactivates activations through 0-1 noise, while Gaussian noise injections add noise to activations (Kingma et al., 2015; DeVries & Taylor, 2017). Bayesian NNs (Gal & Ghahramani, 2016) incorporate these injections during training and evaluation, in contrast to our work's focus solely on their application in training. **Weight Noise:** Unlike Dropout, DropConnect (Wan et al., 2013) randomly deactivates weights or connections between neurons, while Gaussian noise injections add noise to weights (Blundell et al., 2015). Note that we do not model the variance of the Gaussian noise through learnable parameters, as in (Blundell et al., 2015), but rather fix it through a searchable hyperparameter. We do this to ensure a fair comparison with other noise injection approaches, such as Dropout, which do not have learnable parameters and would require changing the model architecture to accommodate them. **Gradient Noise:** Annealed Gaussian noise added to gradients during training has demonstrated its efficacy in improving NN generalisation Neelakantan et al. (2017); Welling & Teh (2011); Zhou et al. (2019); Chaudhari & Soatto (2015); Wu et al. (2020). **Model Noise:** A recent contribution, Gaussian noise injection through periodic weight shrinking and perturbation Ash & Adams (2020), improves retraining generalisation.

In previous work, the impact of noise per injection type was studied. Poole et al. (2014) show that injecting noise at different layers of autoencoders implements various regularisation techniques and can improve feature learning and classification performance. Cohen et al. (2019) show that smoothing classifiers with Gaussian noise naturally induces robustness in the L2 norm. Wei et al. (2020) disentangle and analytically characterise the explicit regularisation effect from modifying the expected training objective and the implicit regularisation effect from the stochasticity of Dropout noise in NNs. Camuto (2021); Camuto et al. (2020) show that training NNs with Gaussian noise injections on inputs and activations regularises them to learn lower frequency functions, improves generalisation and calibration on unseen data but also confers robustness to perturbation. On one hand, Jang et al. (2021) show that training NNs on noisy images can

improve their robustness and match human behavioural and neural responses. On the other hand, Geirhos et al. (2018) demonstrate that adding specific noise to the input can surpass humans in generalisation on that specific noise, but not to other types of noise, while human vision is robust to a wide range of noise types. The results of Geirhos et al. (2018) are confirmed by the results of Kang et al. (2019), who show that robustness against one type of noise does not necessarily transfer to robustness against other types of noise. Furthermore, Kang et al. (2019) consider adversarial training, where a model is trained to be robust against noise-based adversarial attacks (Goodfellow et al., 2015). An adversarial attack is a specific type of noise injection during evaluation, where the noise is designed to fool the model. In comparison, our work focuses on the generalisation and confidence calibration performance of NNs with respect to domain shift in the data distribution, rather than adversarial attacks. We consider enhancing robustness to adversarial attacks through artificial noise injections as future work. Moreover, (Kukačka et al., 2017) provided a taxonomy of regularisation in NNs, covering multiple noise-based approaches.

The closest work to ours is (Chun et al., 2020), which considered regularisation commonly used during training and its impact on generalisation, confidence calibration and out-of-distribution detection in computer vision. While their focus was not noise-specific, as in our work, they overlap with our work by considering input noise: weak augmentation and Gaussian noise, target noise: label smoothing (Müller et al., 2019), input-target noise: MixUp (Zhang et al., 2018). They show that common regularisation techniques improve generalisation, confidence calibration and out-of-distribution detection. In comparison to (Chun et al., 2020), our work focuses on a broader set of noise injections, network architectures, datasets and tasks, evaluation of the weight landscapes, and in-depth noise combinations paired with comprehensive hyperparameter search.

Past work has studied noise injection techniques in isolation, mainly focused on generalisation alone, lacked comprehensive hyperparameter optimisation, and rarely evaluated the robustness of distribution shift. For example, only MixUp, AugMix and label smoothing have been studied in calibration (Guo et al., 2017; Müller et al., 2019; Guo et al., 2019; Yao et al., 2022; Chun et al., 2020). An exception to this is Chun et al. (2020), who studied generalisation, calibration and out-of-distribution detection for some noise injections. While promising, these methods require further unified analysis to determine their relationships across datasets, tasks, architectures and across a broader set of noise injections. Our work addresses these gaps by *1.)* studying the impact across datasets, tasks and architectures; *2.)* benchmarking the impact of noise injections' hyperparameters on transferability between datasets and architectures; *3.)* studying confidence-calibration in addition to generalisation; *4.)* performing a comprehensive hyperparameter search with fair comparisons; *5.)* evaluating robustness to distribution shift; *6.)* providing a methodological framework for combining and tuning various noise injection approaches across categories; and lastly *7.)* visualising the learnt representation or learning landscape across noise injections in 1D or 2D (Goodfellow et al., 2014; Li et al., 2018) across both generalisation and calibration.

## 3 Methodology

We establish a structured methodology to investigate noise injections' effects on NNs. The noise types are divided into **input**, **input-target**, **target**, **activation**, **weight**, **gradient** and **model**, and we enable their conditional deployment through probabilities $\{p_{noise}^i\}_{i=1}^S$ in the range $0 \leq p_{noise}^i \leq 1$, where $S$ denotes the number of noises.

The training allows simultaneous consideration of $S$ noise types, each associated with specific hyperparameters $\{\delta^i\}_{i=1}^S$ and an application function $\{\alpha_{<\text{place}>}^i(\cdot, \delta)\}_{i=1}^S$, where $\alpha_{<\text{place}>}^i$ is the noise application methodology which can be executed at different places, e.g. $\alpha_{\text{input}}$ for input noise, $\alpha_{\text{target}}$ for target noise along with $\cdot$ arbitrary arguments, depending on the noise injection methodology. The different noise types implement only the relevant $\alpha_{<\text{place}>}^i(\cdot, \delta)$ function, while others are ignored. We encourage the reader to refer to the code for the implementation details for each noise type. The probabilities $\{p_{noise}^i\}_{i=1}^S$ allow us to tune the frequency of applying each noise type, while the hyperparameters $\{\delta^i\}_{i=1}^S$ enable us to adjust the magnitude of each noise type. This enables us to tune both the magnitude and frequency of noise injections, unlike, for example, Dropout (Srivastava et al., 2014), which only allows the tuning of the magnitude, and it is applied every batch. The tuning of the frequency allows us to avoid conflicts between noises, as it can be set to 0 if the noise is conflicting with other noises.

Algorithm 1 provides a comprehensive overview of the training process, executed throughout $E$ epochs with $L$ batches processed per epoch. For every batch, input and target data $(x_b, y_b)$ are randomly drawn from the training dataset $\mathcal{D} = \{(x_b, y_b)\}_{b=1}^L$. For each noise in $S$, we sample a uniform random variable $\epsilon \sim U(0,1)$, and if $\epsilon < p_{noise}^i$, we enable noise $i$ with hyperparameters $\delta^i$ for the current batch $b$.

For each noise in $S$, we sample a uniform random variable $\epsilon \sim U(0,1)$, and if $\epsilon < p_{noise}^i$, we enable noise $i$ through setting the toggle $t^i$ to 1 for the current batch $b$. The enabled noises are applied in the order: *1.)* input, target, input-target, *2.)* weights, *3.)* activations, *4.)* gradients and *5.)* model through the apply_$\alpha_{<\text{place}>}(\cdot, t, \delta)$ procedure. The procedure sequentially iterates over the noise types in $S$ and applies the noise if the noise is enabled and the application function $\alpha_{<\text{place}>}^i$ exists. The user specifies the order of the noises in $S$.

Our approach accounts for networks of depth $D$, denoted by $\{f^d(\cdot, W^d)\}_{d=1}^D$, involving weights together with biases $W = \{W^d\}_{d=1}^D$ and activations $\{\phi^d(\cdot)\}_{d=1}^D$ to produce hidden states $\{z_b^d\}_{d=1}^D$. $z_0^b$ corresponds to the input $x_b$, while $z_b^D$ represents the output prediction $\hat{y}_b$.

For **input** noise, we explore Aug-Mix, ODS, weak augmentation: random cropping and horizontal flipping, and additive Gaussian noise injections (Hendrycks et al., 2020; Tashiro et al., 2020; Sietsma & Dow, 1991). For **input-target** we explore MixUp and CMixUp (Zhang et al., 2018; Yao et al., 2022). For **target** noise, we consider label smoothing, and the target noise also inherently involves MixUp and CMixUp (Zhang et al., 2018; Yao et al., 2022; Müller et al., 2019). The **activation** noise examines Dropout and additive Gaussian noise (Srivastava et al., 2014; Kingma et al., 2015) prior to activations for all linear or convolutional

---

**Algorithm 1** Training of a Neural Network with Noise

**Require:** Training dataset $\mathcal{D} = \{(x_b, y_b)\}_{b=1}^L$, $L$ batches, number of epochs $E$, network depth $D$, weights $W = \{W^d\}_{d=1}^D$, hidden states $z_b = \{z_b^d\}_{d=1}^D$, activations $\phi = \{\phi^d(\cdot)\}_{d=1}^D$, weighted operations $f = \{f^d(\cdot, W^d)\}_{d=1}^D$, $S$ noise types, probabilities of applying noise to a batch $p_{noise} = \{p_{noise}^i\}_{i=1}^S$, Noise hyperparameters $\delta = \{\delta^i\}_{i=1}^S$, Noise application functions $\alpha_{<\text{place}>} = \{\alpha_{<\text{place}>}^i(\cdot, \delta)\}_{i=1}^S$.

1: Initialise $W$ randomly
2: **for** $e = 1$ to $E$ **do**
3:    **for** $b = 1$ to $L$ **do**
4:       Randomly select a batch $(x_b, y_b)$ from $\mathcal{D}$
5:       Sample $\epsilon = \{\epsilon^i \sim U(0,1)\}_{i=1}^S$
6:       Set toggles $t = \{t^i = \epsilon^i < p_{noise}^i\}_{i=1}^S$
7:       **Input noise**: apply_$\alpha_{\text{input}}(x_b, t, \delta)$
8:       **Target noise**: apply_$\alpha_{\text{target}}(y_b, t, \delta)$
9:       **Input-target noise**: apply_$\alpha_{\text{input-target}}(x_b, y_b, t, \delta)$
10:      $z_b^0 = x_b$
11:      **for** $d = 1$ to $D$ **do**
12:         **Weight noise**: apply_$\alpha_{\text{weight}}(W^d, t, \delta)$
13:         Compute hidden state $z_b^d = f^d(z_b^{d-1}, W^d)$
14:         **Activation noise**: apply_$\alpha_{\text{activation}}(z_b^d, t, \delta)$ if $d < D$
15:         $z_b^d = \phi^d(z_b^d)$
16:      **end for**
17:      Assign predictions $\hat{y}_b = z_b^d$
18:      Compute loss $\mathcal{L}(\hat{y}^i, y^i)$ and gradients $\nabla_W \mathcal{L}$
19:      **Gradient noise**: apply_$\alpha_{\text{gradient}}(\nabla_W \mathcal{L}, t, \delta)$
20:      Update weights $W$
21:    **end for**
22:    **Model noise**: apply_$\alpha_{\text{model}}(W, e, t, \delta)$
23: **end for**
24: **Procedure:** apply_$\alpha_{<\text{place}>}(\cdot, t, \delta)$
25: **for** $i = 1$ to $S$ **do**
26:    **if** $t^i$ and $\alpha_{<\text{place}>}^i$ exists **then**
27:       $\alpha_{<\text{place}>}^i(\cdot, \delta^i)$
28:    **end if**
29: **end for**

---

layers, except the last layer. For **weight** noise, we consider Gaussian noise added to the weights (Blundell et al., 2015) or DropConnect (Wan et al., 2013) for all linear or convolutional layers, except the last layer. We consider **gradient** Gaussian noise added to all gradients of the loss function (Neelakantan et al., 2017). After the update of the weights, the **model** noise is applied to the weights, for which we consider shrinking the weights and adding Gaussian noise (Ash & Adams, 2020), but not in the last 25% of the training epochs. Out of these noises, label smoothing, MixUp and ODS are exclusive to classification, and CMixUp is applicable only in regression. AugMix and weak augmentation are exclusive to the CV data. The other noises are broadly applicable across tasks.

## 4 Experiments

Next, in Section 4.1 we present the concrete datasets, tasks and architectures used in our experiments, followed by experiments on ID data in Section 4.2, OOD data in Section 4.3, combined noises in Section 4.4, transferability in Section 4.5 and lastly the metric landscape visualisations in Section 4.6.

### 4.1 Experimental Settings

**Tasks, Architectures and Datasets:** We consider various setups, including computer vision (CV) classification and regression, tabular data classification and regression, and natural language processing (NLP) classification. For CV classification we include datasets such as CIFAR-10, CIFAR-100 (Krizhevsky et al., 2009), SVHN (Netzer et al., 2011), and TinyImageNet (Le & Yang, 2015), along with neural architectures such as a fully-connected (FC) net and ResNet (He et al., 2016). For CV regression, we introduce a rotated version of CIFAR-100 to predict the rotation angle, and we also use the WikiFace dataset (Rothe et al., 2015), where the aim is to predict the age based on the image of the face. We use the ResNet model in both cases. We deem the rotation prediction task compelling to evaluate since it is a common task in the literature for self-supervised pre-training (Gidaris et al., 2018). In the realm of tabular data classification and regression, we use an FC network and evaluate noises on diverse datasets, including Wine, Toxicity, Abalone, Students, Adult for classification and Concrete, Energy, Boston, Wine, Yacht for regression (Asuncion & Newman, 2007). We explore NLP classification using the NewsGroup and SST-2 datasets (Lang, 1995; Socher et al., 2013) paired with global pooling convolutional NN (Kim, 2014) and a transformer (Vaswani et al., 2017). The Appendix details the datasets and architectures and gives the complete numerical results.

**Metrics:** To assess the effectiveness of the noise injection methods in classification, we measure their performance using three metrics: Error $(\downarrow, \%)$, Expected Calibration Error (ECE) (Guo et al., 2017) $(\downarrow, \%)$ with 10 bins and the categorical Negative Log-Likelihood (NLL) $(\downarrow)$. For regression, we use the Mean Squared Error (MSE) $(\downarrow)$ and the Gaussian NLL $(\downarrow)$. We test the generalisation of the models by evaluating their performance on the ID test set. For CV classification and regression, we test the robustness of the models by assessing their performance on an OOD test set by applying corruptions (Hendrycks & Dietterich, 2019) to the ID test set. These corruptions include, for example, adding snow or fog to the image, changing the brightness or saturation of the image or blurring the image across 5 intensities. We created the OOD test set for tabular data by adding or multiplying the inputs with Gaussian or Uniform noise or by zeroing some of the input features with Bernoulli noise, similarly across 5 intensities. While vision data has ImageNet-C (Hendrycks & Dietterich, 2019), to the best of our knowledge, there is no similar benchmark for tabular data. Our methodology for introducing perturbations and zeroing out features is designed to simulate a wide range of potential distribution shifts in real-world scenarios, such as instrumentation errors, missing data, and adversarial attacks. We crafted the OOD evaluation to be similar to (Hendrycks & Dietterich, 2019), in terms of the magnitude and severity of the noise, allowing us to systematically evaluate the robustness of the models. To summarise the results, we collect the results for each approach for each dataset and metric and rank them relative to the no noise baseline. For example, -1 means that the approach is one rank better than the no noise baseline, and 1 means that the approach is one rank worse than the no noise baseline. We then average the ranks across the datasets for each task and metric.

**Hyperparameter Optimisation:** We first tune the learning rate and L2 regularisation of a no noise network, which are reused when tuning the HPs of each noise injection method. By tuning the learning rate and L2 regularisation, we wanted to simulate a realistic scenario where the practitioner seeks to add noise to their existing model and does not want to jointly tune the model's hyperparameters and the noise injection method. The tuning was performed with 1 seed, and the winning hyperparameters were retrained 3 times with different seeds. 10% of the training data was used as the validation set to select the best model, with validation NLL used as the selection objective to combine both generalisation and calibration. The tuning is performed using model-based Tree-structured Parzen Estimator method (Bergstra et al., 2011) with successive halving pruning strategy (Jamieson & Talwalkar, 2016). We evaluate 50 trials for each setting, which allows us to manage the trade-off between compute costs and a reasonable number of trials.

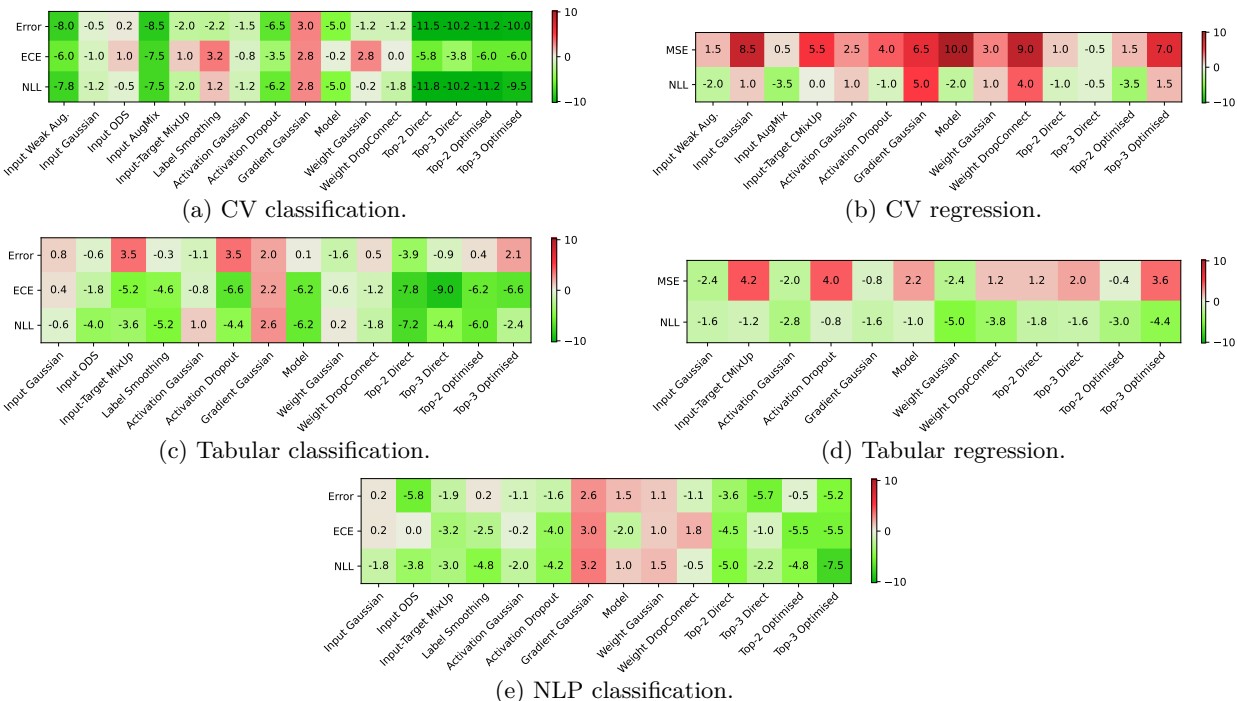

Figure 1: In-domain evaluation of the differences in rankings compared to not using any noise.

## 4.2 In-Domain Evaluation

In Figure 1, we show the in-domain (ID) performance of NNs trained with various noise injection methods across CV classification and regression, tabular data classification and regression, and NLP classification. Overall, we observe that the noise injection methods significantly improve the generalisation and calibration in many cases, but different noise types are needed for various tasks. In CV classification, almost all noises improve the error rate, with many simultaneously improving calibration. The most beneficial noises are AugMix, weak augmentation and Dropout. MixUp and label smoothing are a surprise to a certain extent as they improved generalisation but not calibration. In CV regression, improving generalisation was challenging, with no improvement. However, several noises have improved NLL, with AugMix, weak augmentation, and Dropout achieving the best balance. These results suggest that image augmentation broadly benefits CV, confirming expectations.

Several noises have improved the error rate to a lesser extent or kept it at a similar level in tabular data classification. In contrast, almost all noises have improved ECE and NLL. The improvements were particularly impactful in several cases, with model noise, label smoothing, and Dropout being the best. While ODS is designed to improve adversarial robustness, it improved ECE and NLL while slightly improving error rates. All noises improve NLL for tabular regression, and some significantly improve MSE. Gaussian noises applied to the weights, activations, or inputs are the most useful types of noise for improving the two metrics. In NLP classification, about half of the noises improve error, with some also improving calibration simultaneously. The best noises are Dropout, label smoothing and ODS, which differs from what was the best for CV. These noises significantly lowered error and NLL, while MixUp and model noise were particularly useful for reducing ECE. ODS was beneficial for improving error and calibration via NLL, which can be a surprise as this technique was not previously considered for improving generalisation or calibration.

In Figure 2, we show detailed results for selecting representative datasets across the 5 tasks. We see the improvements in error can be large for CIFAR-10, for example, halving it in some of the best cases – weak augmentation and AugMix, with Dropout also leading to a few percentage point improvements. The situation is similar for ECE, where weak augmentation and AugMix make the ECE one-half or one-third. Many errors are slightly better, but certain noises, such as MixUp, label smoothing, or Gaussian noise added

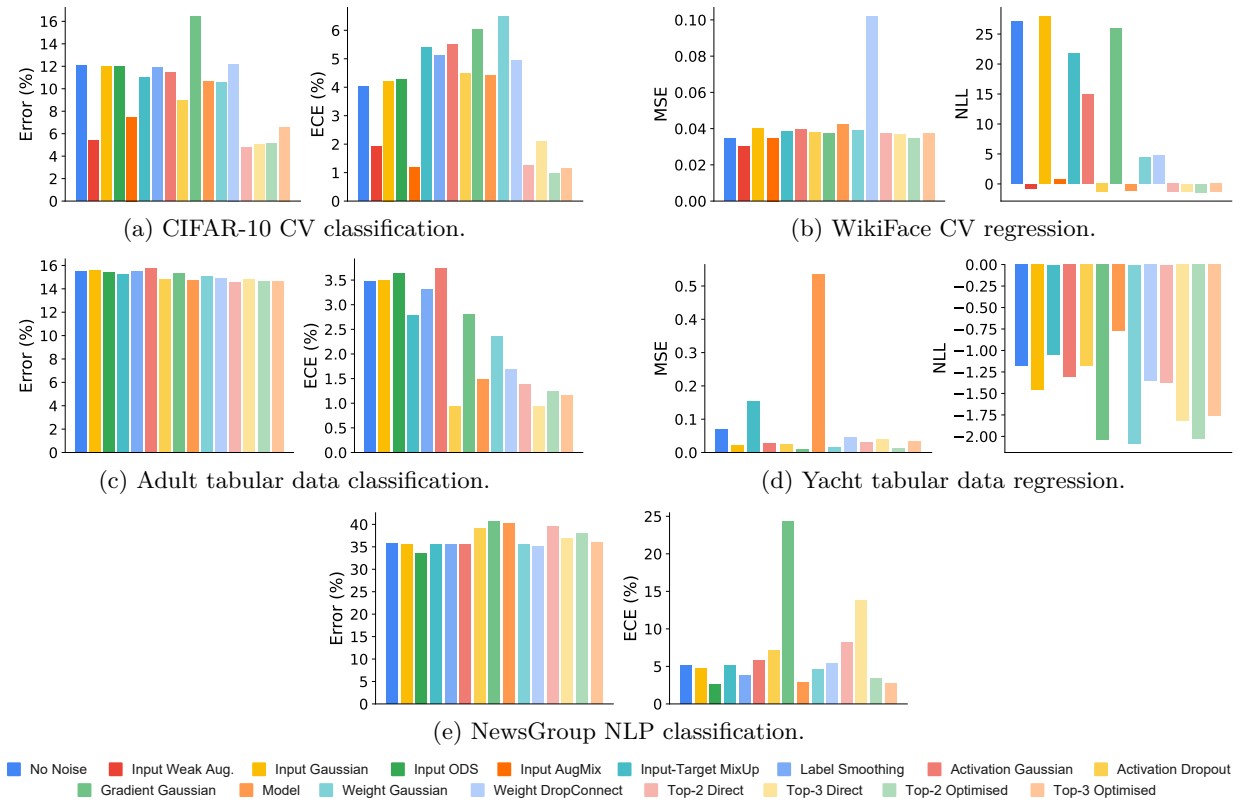

Figure 2: Detailed in-domain performance of NNs trained with various noises across the five tasks.

to the activations, worsen the calibration. For WikiFace, there are more minor improvements in error from weak augmentation and AugMix with overall similar MSE across different noises. Still, the differences in calibration as measured using NLL can be considerable, with most noises improving the NLL significantly.

Moving the focus to tabular data, most noises applied to the Adult classification dataset improve the error marginally. In contrast, many improve ECE significantly, with the best ones being Dropout, model noise and DropConnect. Most noises have significantly improved MSE for the Yacht regression dataset, but CMixUp and model noise led to significant increases. The best ones have been gradient Gaussian and Gaussian noise added to the weights. NLL has been improved in several cases, including gradient Gaussian and weight Gaussian, demonstrating solid improvements in MSE and NLL. The errors stay similar for NLP classification on NewsGroup using the global pooling CNN model. ODS leads to the best improvement, while several noises, specifically Dropout, gradient Gaussian, and model noise, lead to worse generalisation. ODS and label smoothing have also noticeably improved ECE.

**Main Observations:** The noises are effective across various tasks and datasets. The shortlist of the most effective methods is AugMix and weak augmentation in CV, model noise, and Gaussian noise added to weights for tabular data and dropout in NLP. Different task types benefit from different types of noise.

### 4.3 Out-of-Domain Evaluation

We evaluate the performance on the ID test set and an augmented OOD set, including an average over visual corruptions across 19 categories and 5 severities (Hendrycks & Dietterich, 2019). Likewise, we average the performance across 5 categories and 5 severities for tabular data. The summary of the results is in Figure 3, with analysis of correlations between ID and OOD rankings via Kendall Tau score in Table 1. For CV classification, we observe that the generalisation improvements also remain for OOD, but improving calibration in terms of ECE turns out to be much more challenging. The overall ranking of the best noises remains similar, with AugMix and weak augmentation remaining the best. MixUp rose to prominence thanks

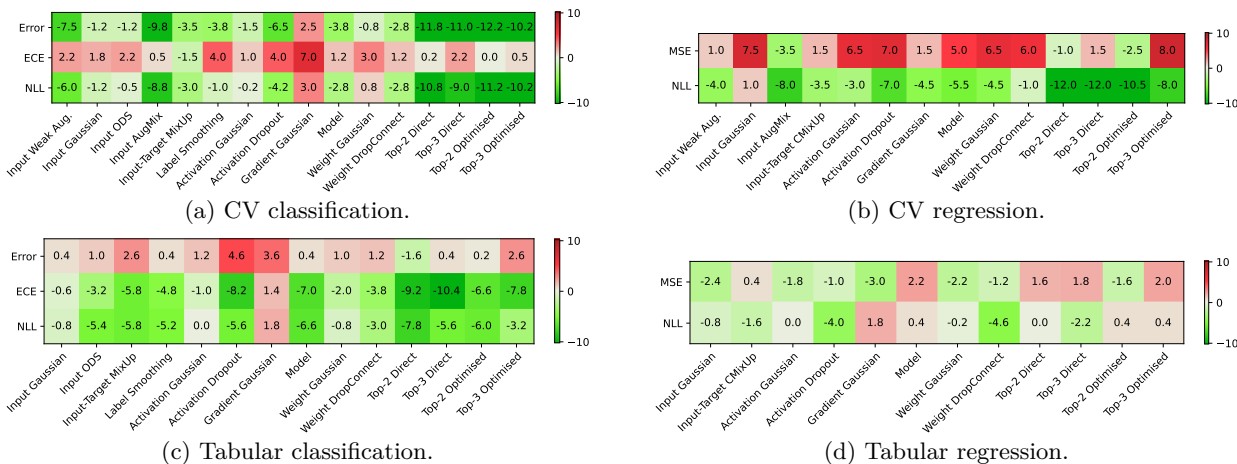

Figure 3: OOD evaluation of the differences in rankings compared to not using any noise.

| Metric | SVHN | CIFAR-10 | CIFAR-100 | TinyImageNet | Average |
|--------|------|----------|-----------|--------------|---------|
| ERROR | $0.912_{\pm\,0.000}$ | $0.706_{\pm\,0.000}$ | $0.897_{\pm\,0.000}$ | $0.912_{\pm\,0.000}$ | 0.857 |
| ECE | $1.000_{\pm\,0.000}$ | $0.618_{\pm\,0.000}$ | $0.250_{\pm\,0.177}$ | $0.118_{\pm\,0.542}$ | 0.496 |
| NLL | $0.971_{\pm\,0.000}$ | $0.515_{\pm\,0.003}$ | $0.868_{\pm\,0.000}$ | $0.882_{\pm\,0.000}$ | 0.809 |

(a) CV classification.

| Metric | Rotated CIFAR-100 | WikiFace | Average |
|--------|-------------------|----------|---------|
| MSE | $0.257_{\pm\,0.202}$ | $0.581_{\pm\,0.002}$ | 0.419 |
| NLL | $-0.238_{\pm\,0.239}$ | $0.810_{\pm\,0.000}$ | 0.286 |

(b) CV regression.

| Metric | Wine | Toxicity | Abalone | Students | Adult | Average |
|--------|------|----------|---------|----------|-------|---------|
| ERROR | $0.861_{\pm\,0.000}$ | $0.743_{\pm\,0.000}$ | $0.529_{\pm\,0.006}$ | $0.532_{\pm\,0.007}$ | $0.924_{\pm\,0.000}$ | 0.718 |
| ECE | $0.905_{\pm\,0.000}$ | $0.905_{\pm\,0.000}$ | $0.867_{\pm\,0.000}$ | $0.867_{\pm\,0.000}$ | $0.962_{\pm\,0.000}$ | 0.901 |
| NLL | $0.924_{\pm\,0.000}$ | $0.981_{\pm\,0.000}$ | $0.790_{\pm\,0.000}$ | $0.810_{\pm\,0.000}$ | $0.962_{\pm\,0.000}$ | 0.893 |

(c) Tabular data classification.

| Metric | Energy | Boston | Wine | Yacht | Concrete | Average |
|--------|--------|--------|------|-------|----------|---------|
| MSE | $0.667_{\pm\,0.001}$ | $0.923_{\pm\,0.000}$ | $-0.564_{\pm\,0.007}$ | $0.641_{\pm\,0.002}$ | $0.872_{\pm\,0.000}$ | 0.508 |
| NLL | $-0.615_{\pm\,0.003}$ | $0.974_{\pm\,0.000}$ | $0.154_{\pm\,0.510}$ | $0.590_{\pm\,0.004}$ | $0.436_{\pm\,0.042}$ | 0.308 |

(d) Tabular data regression.

Table 1: Kendall Tau correlation between ID and OOD rankings of different noise types for various tasks.

to the best OOD calibration and improved errors and NLL. Analysis of Kendall Tau correlation in Table 1a shows that ID and OOD rankings are strongly correlated for error and NLL, while only moderately for ECE. CV regression is similar to classification ranking the best noises, with only AugMix leading to improvements in OOD generalisation. However, calibration is improved by most noises, with AugMix excelling. Only a minor correlation exists between ID and OOD rankings for MSE and NLL metrics. For tabular classification, the noises generally improve ECE and NLL but not the error rate under OOD settings, with model noise, label smoothing, and Dropout being the best. This suggests all of these are among the best noises for both ID and OOD. ID and OOD rankings show a strong correlation overall. Several noises improve OOD generalisation and calibration for tabular regression, with DropConnect, Dropout and Gaussian noise added to the inputs, leading to the best overall improvements. The ID and OOD ranking Kendall Tau correlation is low in this case. MixUp and CMixUp have improved OOD calibration for both tabular classification and regression.

We study selected representative datasets regarding OOD performance in Figure 4. OOD results on CIFAR-10 show that AugMix significantly improves both error and ECE, making ECE one-third of the no noise equivalent. MixUp leads to similarly considerable improvements in ECE and more minor yet significant improvements in error. Several noises, e.g., Dropout and Gaussian noise, added to activations or weights lead to a few percentages worse ECE. On WikiFace, most OOD MSE values are similar, but OOD calibration in NLL is improved significantly for several noises, including AugMix, weak augmentation or Dropout. Improvements in generalisation for the Adult tabular classification dataset are minor, but the improvements in calibration can be significant, for example, Dropout and model noise halving the OOD ECE value. For the Yacht tabular regression dataset, the improvements in generalisation have been more critical, with the same being true for calibration measured in terms of OOD NLL.

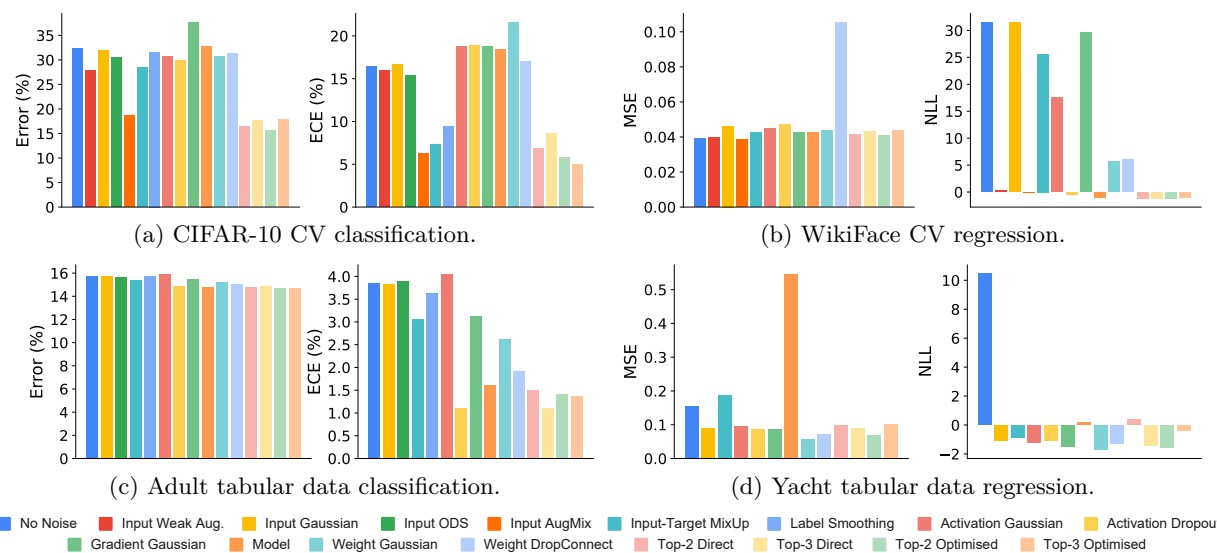

Figure 4: Detailed OOD performance of NNs trained with various noises across the four tasks.

We include an additional OOD investigation on TinyImageNet, where we study the performance on black and white sketches. We use the same classes as in TinyImageNet (Wang et al., 2019) and report the full details and results in the Appendix. The shift from natural images to sketches represents a more significant domain shift than our standard OOD shifts (Hendrycks & Dietterich, 2019), and hence also tests the generalisation of the noises to a larger degree. We do this to disambiguate the performance of AugMix, which uses augmentations that may be similar, but are still distinct (Hendrycks et al., 2020), to the ones used for our OOD evaluation. The results clearly show that AugMix obtains similar rankings for all three metrics on both the synthetic OOD and sketch domain evaluations. Overall we see strong Kendall Tau correlation in terms of the error across all noises, but smaller in terms of ECE and NLL.

**Main Observations:** We see consistent improvements in OOD generalisation and calibration for tabular data. Errors and NLL are improved for CV classification, but calibration is generally not improved when measured via ECE. CV regression usually sees improvements in OOD NLL only. The best ID noise types have often remained the best OOD, but overall, the correlations between ID and OOD rankings were not high in all cases. MixUp, or CMixUp for regression, showed surprising behaviour as it was much more helpful for improving OOD calibration than ID calibration.

### 4.4 Combination of Noises

Next, we evaluate the combination of noises. We construct them from empirical combinations of the Top 2 or 3 noises from the ID evaluation for each task, based on average rank across respective datasets and metrics. We consider two cases: *1.)* the found hyperparameters of the noises are directly applied, and *2.)* the hyperparameters of the noises are jointly tuned. We utilise the same 50-trial budget to tune the selected noises jointly. We restricted our experiments to combinations of two or three noises, as we empirically found that tuning all the noises jointly in our study is not feasible even with a larger computational budget. We consider the individual noise hyperparameter tuning and then the direct application or joint tuning of their combination to be a realistic scenario where a practitioner has a limited compute budget and wants to improve their model's performance (Godbole et al., 2023) iteratively.

The results are already in Figures 1, 2, 3 and 4 and denoted as Top-2 Direct, Top-3 Direct for *1.)*, Top-2 Optimised and Top-3 Optimised for *2.)*. The combinations for Top-2 and Top-3 are in Table 2. To simplify the analysis of how effective the different combinations of noises are, we compute their average rank improvement compared to no noise and report it in Table 3. Notice that when we choose a combination of noises to involve noises from the same category, for example, ODS and input Gaussian are both input noises, these are applied sequentially.

| Task | Top-2 | Third Method |
|---|---|---|
| CV classification | Input AugMix, Input Weak Augmentation | Activation Dropout |
| NLP classification | Activation Dropout, Target Label Smoothing | Input ODS |
| Tabular classification | Model, Target Label Smoothing | Activation Dropout |
| CV regression | Input AugMix, Input Weak Augmentation | Activation Dropout |
| Tabular regression | Weight Gaussian, Activation Gaussian | Input Gaussian |

Table 2: Top task and noise combinations. Underlined methods are from the same type.

| Scenario | Top-2 Direct | Top-3 Direct | Top-2 Optimised | Top-3 Optimised |
|---|---|---|---|---|
| ID | -4.75 | -3.69 | -4.34 | -3.30 |
| OOD | -5.24 | -4.43 | -5.00 | -2.59 |

Table 3: Average rank improvement over no noise for the different combination strategies.

We can draw several observations from Table 3. *1.)* Directly combining hyperparameters for the top two or three noises is a good strategy when considering the same budget for hyperparameter tuning as for one noise. A significantly larger budget is likely needed for jointly optimising hyperparameters of multiple noises. *2.)* A combination of two noises performs better than a combination of three noises, suggesting there may be negative interactions when too many noise sources are used without extensive hyperparameter tuning. *3.)* The behaviour of different combination strategies is consistent across ID and OOD settings.

Commenting on the overall performance of the combinations of noises, the combinations are typically better for classification tasks than the individual noises. Still, the opposite may be true for regression. As observed in Figures 1a, 1c and 1e, the combinations are consistently ranked lowest in comparison to using no noise for classification, showing the effectiveness of the combinations. However, Figures 1b and 1d show that regression can benefit from only using one noise at a time. OOD analysis in Figure 3 confirms the benefits of combinations of noises for classification tasks, and it also shows that it can be beneficial for regression, contrary to the ID behaviour. The combinations are generally ranked lower and can improve calibration and generalisation, as seen in lower MSE, NLL, or error and ECE simultaneously.

**Main Observations:** The combination of noises can improve both calibration and generalisation simultaneously. Directly combining two noises is better than three, as too many can lead to conflicts. Combining noises directly with their hyperparameters is generally reasonable and a significantly larger tuning budget for optimising hyperparameters would be needed for optimising their hyperparameters jointly.

### 4.5 Transferability of Hyperparameters Across Datasets and Models

Furthermore, we evaluate the transferability of the hyperparameters across datasets and models. We consider two cases: the transfer of hyperparameters to a new dataset and the transfer of hyperparameters to a new architecture. For the dataset transfer, we consider the following combinations: SVHN to CIFAR-10, CIFAR-10 to CIFAR-100, CIFAR-100 to TinyImageNet, and 3 tabular regression datasets combinations, Concrete to Energy, Boston to Wine, Yacht to Concrete. We consider the following combinations for the architecture transfer: FC to ResNet-18 for SVHN and ResNet-18 to ResNet-34 for CIFAR-10, CIFAR-100 and TinyImageNet. We use a NN with an additional layer for tabular data, i.e., five layers instead of four.

#### 4.5.1 Dataset Transfer

Figures 5a and 5c show the dataset transfer results for ID settings, with OOD settings shown in Figures 6a and 6c. We observe generally good transferability of hyperparameters across datasets for CV classification in ID and OOD settings. In particular, weak augmentation, AugMix and Dropout lead to solid improvements in the ID setting. AugMix also excels in OOD scenarios under dataset transfer, but weak augmentation and Dropout are not as strong in calibration measured using ECE. Certain noises are less transferable, including Gaussian noise added to the input and DropConnect. Hyperparameters for noise in tabular regression are less transferable because of worse generalisation measured using MSE.

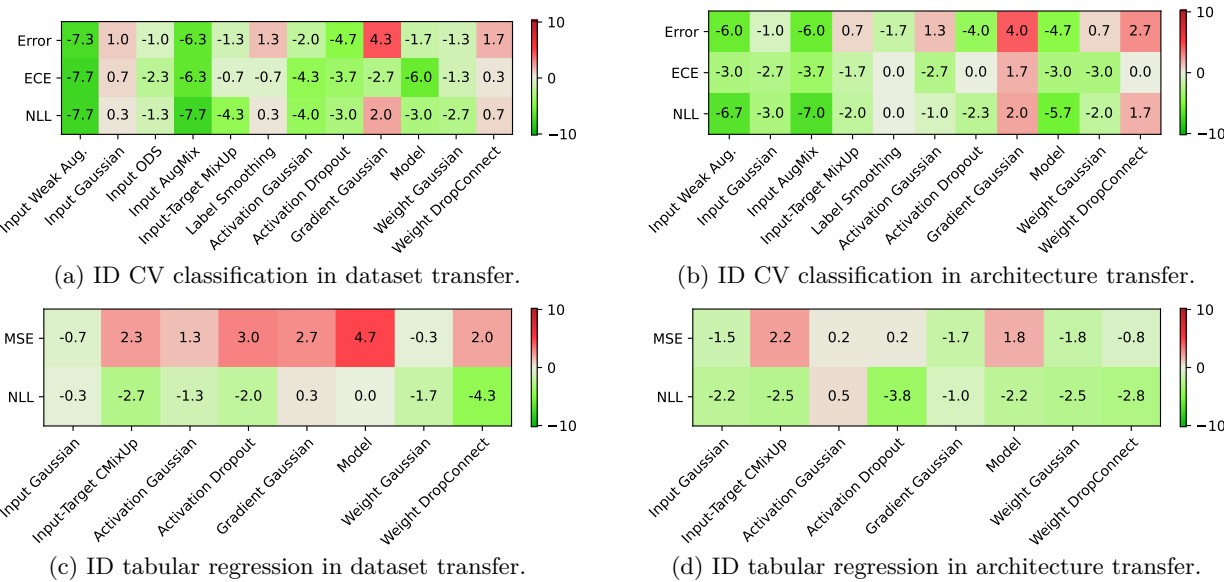

(a) ID CV classification in dataset transfer.

(b) ID CV classification in architecture transfer.

(c) ID tabular regression in dataset transfer.

(d) ID tabular regression in architecture transfer.

Figure 5: Transfer of hyperparameters on in-domain (ID) data.

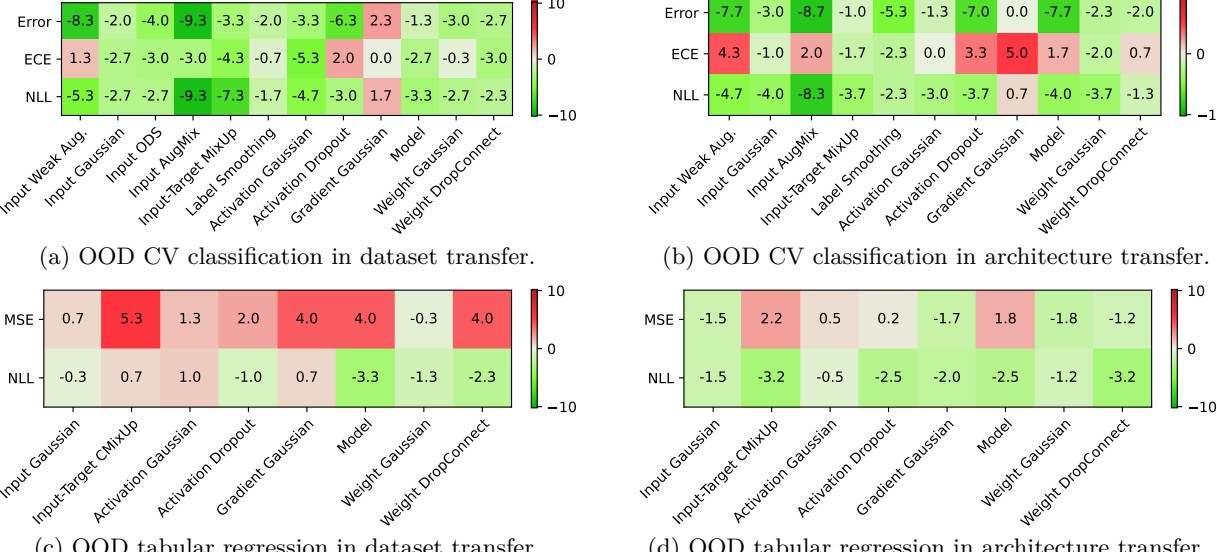

(a) OOD CV classification in dataset transfer.

(b) OOD CV classification in architecture transfer.

(c) OOD tabular regression in dataset transfer.

(d) OOD tabular regression in architecture transfer.

Figure 6: Transfer of hyperparameters on out-of-domain (OOD) data.

**Main Observations:** The transfer of hyperparameters from dataset to dataset generally works well for CV classification. However, caution is advised as it is not the case for all noise types. For tabular data regression, tuning of hyperparameters is recommended.

### 4.5.2 Architecture Transfer

In Figures 5b and 5d, we show the ID results for the architecture transfer, with Figures 6b and 6d reporting the OOD results. The transferability of noise hyperparameters is lower than across datasets for CV classification, but it is still successful, especially for weak augmentation and AugMix for ID settings. Transfer of hyperparameters for tabular data regression works for certain noise types in the ID setting, including adding Gaussian noise to the input or the weights, which are among the Top-3 noises for tabular data regression.

**Main Observations:** Transfer of hyperparameters across architectures appears more challenging than across datasets but can be successful in some instances. Caution is advised, and tuning is recommended.

### 4.6 Learnt Representation Landscapes

We study the learnt representation landscapes of NNs trained with various noises through the lenses of ID and OOD performance in terms of error, ECE, NLL or MSE. We consider the noises individually, with the ID-found hyperparameters starting from the same weight initialisation for fairness. We visualise linear interpolation modulated through an $\alpha$ parameter between the final, $\alpha = 0$ and initial model, $\alpha = 1$ (Goodfellow et al., 2014). The interpolation empirically investigates the smoothness of the training process. We also visualise the landscape in 2D (Li et al., 2018; Holbrook, 2020) by saving the network after each epoch and concatenating the weights. Instead of using random coordinates, we use the first two principal components of the weights as the coordinates. We normalise them based on the magnitude of the original weights, and we project all the weights onto these two components in the vicinity of $\alpha$ and $\beta$. The 2D visualisations show us the exploration and exploitation of the training process. In Figures 7 and 8, we compare the metric landscapes of no noise with AugMix and Dropout noises, respectively on CIFAR-10 and WikiFace datasets. We used 20 points for linear interpolation and 100 points for the 2D plots across five selected OOD augmentations and 1000 test data samples for compute efficiency. In red, we show the error or MSE; in green, we offer the NLL or ECE. In the 1D plots, ● and ▲ stand for ID and OOD error or MSE, and × and ■ stand for ID and OOD ECE or NLL. In the 2D plots, the darker combined contours signify worse performance than the lighter parts, and the ★ in blue or black denotes the start or end weights, respectively. The Appendix contains the metric landscapes for all other noises, tabular classification – Adult, and regression – Yacht datasets.

Observing Figures 7 and 8, we first notice the ID and OOD results are similar, with the OOD results being slightly worse across all metrics. This includes both the 1D plots and the 2D plots. Second, as seen in Figures 7a and 7b, the curves for error, representing generalisation, and ECE or NLL, representing confidence calibration, do not share the same shapes or curvatures. MSE and NLL curves in Figure 8a are more similar than error and ECE curves. Looking at the 1D plots, for example in Figures 7a 7b and 8a, the error or MSE can be more smoothly interpolated than ECE or NLL. Figure 8a shows models trained without noise can become overconfident, reflected in large NLL and small MSE. Adding noise such as Dropout can fix this, leading to low NLL for the final model in Figure 8d. Looking at the 2D plots in Figures 7c and 8b, the error or MSE valley is wider than the ECE or NLL valley, and they are not aligned. From a detailed comparison between no noise and AugMix or Dropout in Figures 7 and 8, we observe that AugMix and Dropout can smoothen the optimisation in the 1D plots, but not for ECE, and decrease the gap between ID and OOD performance. The 2D plots show that AugMix and Dropout can explore broader metric landscapes than no noise, shown in ranges of $\alpha$ and $\beta$ in the 2D plots, and marginally align the error or MSE with NLL. Seen in the lightness of the 2D contour plots, the noises navigate lower NLL or ECE landscapes than no noise.

Our general observations considering both CV and tabular datasets show that while noises such as AugMix, weak augmentation, MixUp or activation and weight noises based around Dropout can smoothen the optimisation regarding error or MSE, they rarely smoothen the optimisation regarding ECE. The metric landscapes often look similar to no noise, but the optimisation ends in more profound valleys. Across the datasets and tasks, label smoothing, input additive Gaussian and ODS have minimal effect on the 2D landscapes or 1D interpolation. The model shrink and perturb make the optimisation more "stairs-like", and the metric landscape explored is broader. Together with gradient Gaussian noise, the shrink and perturb noises explore broader metric landscapes than the others. No method drastically changes the metric landscape or the interpolation from the default, but they can make the optimisation smoother or broader.

**Main Observations:** The metric landscapes for error or MSE and ECE or NLL are different, and the noises can smoothen the optimisation in terms of error or MSE but not necessarily in terms of ECE or NLL. When a model trained without noise is overconfident, adding noise to the training can resolve it and lead to a significantly better-calibrated model at the end of training.

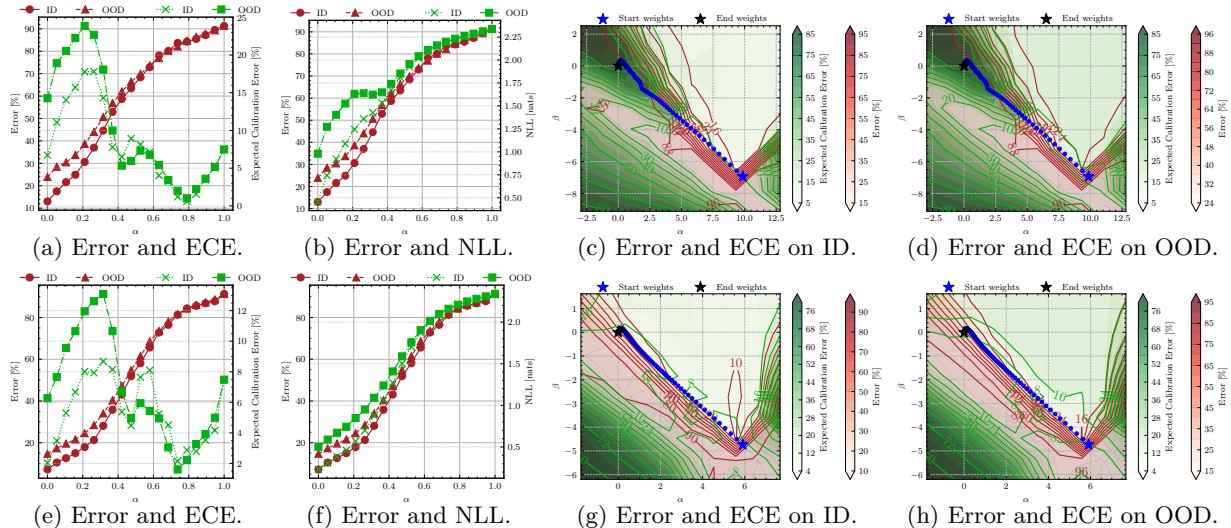

Figure 7: No noise (top) and Input AugMix (bottom) on CIFAR-10.

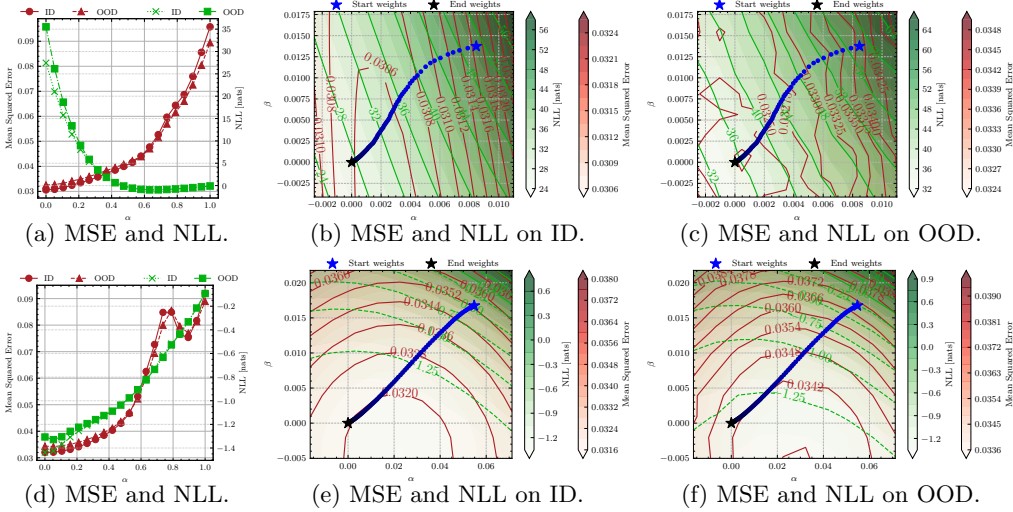

Figure 8: No noise (top) and Activation Dropout (bottom) on WikiFace.

## 5 Conclusion

**Key Takeaways**: Noise injection methods can improve NN performance across various tasks and datasets. This is despite the fact L2 regularisation was already tuned to prevent overfitting, indicating noise injection methods can provide additional benefits beyond standard regularisation. The methods were not equally efficient across all tasks and datasets, with significant differences in performance between regression and classification. Nevertheless, out of the considered noise types, the proposed methodology identified at least one noise or a combination of noises that demonstrated improvements in both calibration and generalisation over the no noise baseline for all tasks. The most effective noise for CV was AugMix, model shrink and perturb and Gaussian noise added to weights for tabular data classification and regression, respectively. At the same time, Dropout and label smoothing worked the best for NLP. Even though ODS was not designed to improve calibration and generalisation, it has shown promising performance in several cases. Combining noises outperformed individual noises in most classification cases, with regression often benefitting from using only one noise at a time. While directly combining hyperparameters of noises is a reasonable strategy, tuning them can still be valuable if a large budget is used. The noises improved ID and OOD performance, but

the ID rankings were sometimes inconsistent with the OOD evaluation. AugMix remained highly ranked for robustness, demonstrating that domain-specific inductive biases are beneficial when crafting noise injection methods as they can improve generalisation and calibration. Note that our aim was not to add new knowledge about AugMix's performance against image corruptions, given its well-documented performance (Hendrycks et al., 2020), but to demonstrate that inductive biases should be considered when generating noise for better performance. The visualisation showed noises can smoothen the optimisation in terms of error or MSE but not necessarily in terms of ECE or NLL. It also showed noise can help mitigate overconfidence. Overall, the results indicate practitioners should consider combining noises, e.g. AugMix and Dropout, and tuning hyperparameters for their specific problem.

**Limitations**: To conduct this study, we had to restrict the experiments' scope. Our scope was limited to experimental datasets, tasks such as classification and regression, and standard NN architectures. Testing on more complex data and downstream tasks such as object detection, segmentation, or reinforcement learning would reveal more profound impacts of noise injection. Furthermore, diving deeper into one particular domain, such as NLP, could provide more insights into the effectiveness of noise injection methods. Moreover, we also limited the optimisation to SGD with momentum and a cosine learning rate schedule, which were tuned beforehand to make the hyperparameter search tractable. To draw practical conclusions, we evaluated the noise performance by minimising the NLL rather than exploring all possible settings. The costs associated with adjusting and evaluating different noises limited the scope of the experiments. Consequently, certain noises might prove more effective with more thorough tuning and a larger budget. This is especially true for noise combinations, where the number of possible combinations grows exponentially with the number of noises. Nonetheless, the existing findings offer valuable insights for practitioners by giving a preliminary indication of the most promising noise sources. This enables users to concentrate their efforts and compute the budget required for tuning. For example, AugMix, incorporating domain-specific inductive biases, was transferable and effective even with a limited budget. Developing methods to choose hyperparameters without the need for extensive tuning would enhance the accessibility of these techniques. Lastly, the out-of-distribution evaluation was focused on synthetic augmentations, which may not fully capture the real-world distribution shift, as noticed in (Taori et al., 2020) for computer vision tasks.

**Future Directions**: The strong performance of AugMix highlights the potential for developing specialised, domain-specific noise techniques such as DeepAugment (Hendrycks et al., 2021). For example, tailored domain-specific noise methods could benefit tabular data-based problems and NLP. Future work should also explore specific data-architecture noise interactions, as the transferability of hyperparameters was limited. Inspired by the annealed gradient noise, annealing noise levels overtraining may also prove helpful, as early noise could encourage robustness. In contrast, low late-stage noise could enable convergence on a high-accuracy solution. The potential for combining noises from the same category should also be investigated further. The noises affected the entire architecture, but targeting noise injection methods that only affect specific layers or sections of the network may be possible, requiring more or less regularisation. Specific noise-based approaches for simultaneously exploring the generalisation and confidence calibration trade-off should be investigated further. As demonstrated across image and tabular domains, the empirical differences between many methods seem minor. This indicates that there are more fundamental determinants of performance. Therefore, theoretical analysis of noise injection methods would be beneficial in understanding the underlying mechanisms and providing guidance for future research. Lastly, testing on wider-scale out-of-distribution datasets and real-world distribution shifts would provide a more comprehensive evaluation of the noise injection methods' impact on robustness. We hope our study and framework, embedded in our codebase, will assist further research in this area.

## Acknowledgements

Martin Ferianc was sponsored through a scholarship from the Institute of Communications and Connected Systems at UCL. Ondrej Bohdal was supported by the EPSRC Centre for Doctoral Training in Data Science, funded by the UK Engineering and Physical Sciences Research Council (grant EP/L016427/1) and the University of Edinburgh.

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

## Appendix

In the Appendix, we first provide the experimental settings and the hyperparameter ranges for all the experiments in Section A. We then provide the full numerical results and visualisations for all the experiments in Section B.

## A   Settings

### A.1   General Settings

We used stochastic gradient descent with a momentum of 0.9 to train all the networks. The learning rate and L2 regularisation were tuned and reused for each noise injection method. We used a cosine annealing learning rate schedule without restarts (Loshchilov & Hutter, 2017) for all experiments. In most cases, we used gradient norm clipping of 20.0 to stabilise the training, with gradient clipping of 10.0 for tabular regression and 5.0 for WikiFace. The batch size was set to 256 for all experiments. The final results are the average of 3 runs with 3 different seeds. We used cross-entropy loss for all classification experiments. For regression, we used the Gaussian negative log-likelihood (NLL) loss, where we modelled the variance as an additional output passed through an exponential function to ensure positivity. We added a small $\epsilon$ of $1e^{-8}$ to the softmax probabilities to avoid NaNs. We clipped the variance between $1e^{-4}$ and $1e^4$ to avoid NaNs. The hyperparameter ranges, and the sampling scale for each dataset-architecture pair are in Table 4. The hyperparameters and implementations of all the noises and experiments can be found in the code, which will be open-sourced. We used the default PyTorch weight initialisation for all layers.

For the tabular OOD experiments, we constructed custom augmentations where we applied Gaussian or Uniform noise scaled by the magnitude of the input features across 5 severities for addition: [0.02, 0.04, 0.06, 0.08, 0.1] or multiplication [0.04, 0.08, 0.12, 0.16, 0.2] where the severity scaled the range or the standard deviation of the noise applied to the input. Additionally, we zeroed out some input features with probability [0.04, 0.08, 0.12, 0.16, 0.2], denoting 5 severities. In total, there were 5 different input shifts across 5 severities each. To avoid label-flipping in applying these augmentations, we have introduced an empirically determined scaling factor for the severity of all noises for a particular dataset. They multiply the [0.02, 0.04, 0.06, 0.08, 0.1] or [0.04, 0.08, 0.12, 0.16, 0.2] by a scaling factor to determine the severity of the noise applied to the input based on the dataset. We use a K-nearest neighbour (KNN) classifier, specifically a 1-neighbour KNN, trained on a dataset's original, unmodified data. This classifier then predicts labels for the augmented data. We adjust the scaling factor for each dataset so that the KNN classifier's accuracy on the augmented data exceeds 99% or the mean squared error is less than 0.01. This approach ensures that the augmentations are subtle enough to maintain the integrity of the data, meaning the nearest neighbour—the closest match in the original dataset—remains the same. However, this is not a perfect solution, with an empirical guarantee that the augmentations are not too severe, and the nearest neighbour's label remains the same. The found scaling factors are shown in Table 5.

Regarding noise implementation details, Dropout, DropConnect, additive weight or activation Gaussian noise, are applied to all linear and convolutional weights throughout the network, excluding the last layer and normalisation layers. Both model and Gaussian gradient noise are implemented on all weights within the network, encompassing affine parameters in normalisation layers. Rotation was omitted from AugMix, given that one of our tasks involved predicting the rotation angle.

### A.2   Vision Experiments

For SVHN, we used a fully connected network with 4 hidden layers of 150 units followed by ReLU activations. When we used ResNet-18 we used it with [64, 128, 256, 512] channels in 4 stages with [2, 2, 2, 2] blocks with strides [1, 2, 2, 2]. When we used ResNet-34, we used it with [64, 128, 256, 512] channels in 4 stages with [3, 4, 6, 3] blocks with strides [1, 2, 2, 2]. In all cases, we trained the networks for 200 epochs. We only used 0-1 truncation followed by normalisation for each dataset without further data augmentations for training, validation and test sets. For rotation experiments, we enabled uniform rotations between (0, 90°) degrees and rescaled the targets accordingly to [-1, 1]. Gaussian noise, motion blur, snow, elastic

transformation, and JPEG compression were selected as OOD augmentations for visualisation experiments across all 5 severities. For CIFAR-10, CIFAR-100, and SVHN we used the dedicated test sets as the test set, while for TinyImageNet we used the official validation set as the test set. We used 10% of the training data to construct the validation sets. For WikiFace, we used 10% of the data as the test set and 10% of the remaining data as the validation set and the rest as the training set.

For evaluation on the sketch domain, we utilise ImageNet-Sketch dataset from Wang et al. (2019) and derive our own TinyImageNet-Sketch dataset from it. As the ImageNet-Sketch images are not square, we crop the centre and then resize to $64 \times 64$, the same size as TinyImageNet. We only keep the images of the same 200 classes as used in TinyImageNet, allowing us to directly evaluate pre-trained models on the new TinyImageNet-Sketch dataset.

### A.3 NLP Classification Experiments

For our NLP experiments we used NewsGroup and binary SST datasets. In the NewsGroup dataset we aim to classify news texts into one of the 20 available categories based on the topic. The task in the binary SST dataset is to predict the sentiment of a movie review into 2 categories: positive or negative. For SST we only consider the text itself, rather than also considering the available parse trees, making the task more challenging.

Each dataset was first pre-processed with respect to glove embeddings (Pennington et al., 2014) into embeddings of dimension 100 and sequence length 100 and 50 for NewsGroup and SST respectively. In both cases, we trained the networks for 100 epochs. We used the global-pooling convolutional network architecture from Kim (2014) with planes [128, 128, 128] and a transformer decoder (Vaswani et al., 2017) with embedding dimensions 100, 6 layers, 8 heads, 1024 hidden dimensions, 64 dimensions per head and no dropout. For the NewsGroup experiments, we used about 5% of the data as the test set, 10% of the remaining data as the validation set and the rest as the training set. For the SST experiments, we used the original development set as the test set, for validation we took 10% of the original training set and used the remainder as the training set. No OOD test was set for the NLP task due to a lack of suitable perturbations to construct OOD data.

### A.4 Tabular Regression Experiments

For the tabular experiments, we used a fully connected network with [100, 100, 100, 100] hidden units and ReLU activations. In all cases, we trained the networks for 100 epochs. We normalised the input features and targets to zero mean and unit variance by using the training set statistics and applied the same normalisation to the validation and test sets. For the tabular experiments, we used 20% of the data as the test set and 10% of the remaining data as the validation set and the rest as the training set. The regression targets were normalised to zero mean and unit variance.

## B Full Results

We provide full results of all experiments in the paper, where the main reported value is the mean across 3 repetitions, followed by the standard deviation. The ranks presented in the main body of the paper can be obtained by ranking the results in each table by the metric of interest. Following the tables, there are the visualisations of metric landscapes for CIFAR-10, Adult, WikiFace and Yacht datasets. We encourage the reader to look at our code for other datasets to regenerate them from there.

| Hyperparameter ($\delta$) | Range | Scale |
|---|---|---|
| Learning rate (LR) | $[10^{-4}, 10^{-1}]$ | Log |
| L2 weight | $[10^{-7}, 10^{-1}]$ | Log |
| Input Gaussian noise std. | $[10^{-4}, 10^{-1}]$ | Log |
| Input AugMix alpha | $[0, 1]$ | Linear |
| Input AugMix severity | $[1, 10]$ | Linear |
| Input AugMix width | $[1, 5]$ | Linear |
| Input AugMix chain-depth | $[-1, 3]$ | Linear |
| Input ODS epsilon | $[10^{-4}, 10^{-1}]$ | Log |
| Input ODS temperature | $[0.5, 5.0]$ | Log |
| Input-Target MixUp alpha | $[0, 1]$ | Linear |
| Input-Target CMixUp alpha | $[0, 1]$ | Linear |
| Input-Target CMixUp sigma | $[10^{-4}, 10^{2}]$ | Log |
| Target Label Smoothing | $[0, 0.25]$ | Linear |
| Activation Gaussian noise std | $[10^{-4}, 10^{-1}]$ | Log |
| Activation Dropout rate | $[0, 1]$ | Linear |
| Gradient Gaussian noise $\eta$ | $[0, 1]$ | Linear |
| Gradient Gaussian noise $\gamma$ | $[0, 1]$ | Linear |
| Weight Gaussian noise std | $[10^{-4}, 10^{-1}]$ | Log |
| Weight DropConnect rate | $[0, 1]$ | Linear |
| Model noise shrink factor | $[0.0, 1.0]$ | Linear |
| Model noise std | $[10^{-7}, 10^{-3}]$ | Log |
| Model noise frequency | $[0, 20]$ | Linear |

Table 4: Hyperparameters (HPs) optimised for individual noises and their range.

| Dataset | Task | Scaling Factor |
|---|---|---|
| Adult | Classification | 0.1438 |
| Abalone | Classification | 0.0886 |
| Concrete | Regression | 0.0207 |
| Energy | Regression | 0.0546 |
| Wine | Classification | 0.0078 |
| Wine | Regression | 0.0048 |
| Yacht | Regression | 0.0886 |
| Toxicity | Classification | 0.3793 |
| Students | Classification | 0.2336 |
| Boston | Regression | 0.0886 |

Table 5: Scaling factors for tabular data.

| Noise Type | SVHN | | CIFAR-10 | | CIFAR-100 | | TinyImageNet | |
|---|---|---|---|---|---|---|---|---|
| | ID | OOD | ID | OOD | ID | OOD | ID | OOD |
| No Noise | $16.33_{\pm0.07}$ | $20.18_{\pm0.09}$ | $12.04_{\pm0.21}$ | $32.32_{\pm0.24}$ | $44.69_{\pm0.84}$ | $61.52_{\pm0.52}$ | $54.15_{\pm0.36}$ | $75.65_{\pm0.19}$ |
| Input Weak Aug. | $14.67_{\pm0.14}$ | $18.59_{\pm0.14}$ | $5.39_{\pm0.17}$ | $27.86_{\pm0.37}$ | $26.51_{\pm0.10}$ | $52.57_{\pm0.13}$ | $39.73_{\pm0.27}$ | $67.73_{\pm0.13}$ |
| Input Gaussian | $16.45_{\pm0.23}$ | $20.23_{\pm0.22}$ | $12.02_{\pm0.05}$ | $31.97_{\pm0.31}$ | $44.34_{\pm1.70}$ | $61.14_{\pm1.60}$ | $53.28_{\pm0.24}$ | $75.12_{\pm0.06}$ |
| Input ODS | $16.35_{\pm0.20}$ | $20.13_{\pm0.15}$ | $12.01_{\pm0.16}$ | $30.42_{\pm0.45}$ | $44.28_{\pm0.64}$ | $61.34_{\pm0.56}$ | $66.47_{\pm15.18}$ | $82.46_{\pm8.72}$ |
| Input AugMix | $12.28_{\pm0.07}$ | $15.67_{\pm0.07}$ | $7.48_{\pm0.06}$ | $18.75_{\pm0.25}$ | $30.09_{\pm0.25}$ | $46.05_{\pm0.13}$ | $42.49_{\pm0.24}$ | $61.58_{\pm0.16}$ |
| Input-Target MixUp | $13.95_{\pm0.01}$ | $17.71_{\pm0.11}$ | $10.97_{\pm0.13}$ | $28.38_{\pm0.70}$ | $46.15_{\pm0.05}$ | $63.00_{\pm0.02}$ | $54.02_{\pm0.53}$ | $75.33_{\pm0.44}$ |
| Label Smoothing | $16.35_{\pm0.17}$ | $20.10_{\pm0.12}$ | $11.88_{\pm0.39}$ | $31.63_{\pm0.29}$ | $42.48_{\pm0.47}$ | $58.41_{\pm0.55}$ | $52.47_{\pm0.21}$ | $73.96_{\pm0.15}$ |
| Activation Gaussian | $16.33_{\pm0.17}$ | $20.14_{\pm0.13}$ | $11.44_{\pm0.12}$ | $30.71_{\pm0.20}$ | $44.58_{\pm0.28}$ | $61.81_{\pm0.15}$ | $53.96_{\pm0.16}$ | $75.45_{\pm0.14}$ |
| Activation Dropout | $13.83_{\pm0.13}$ | $17.43_{\pm0.10}$ | $8.93_{\pm0.25}$ | $29.85_{\pm0.86}$ | $41.51_{\pm0.83}$ | $58.92_{\pm0.80}$ | $43.26_{\pm0.35}$ | $69.32_{\pm0.32}$ |
| Gradient Gaussian | $17.59_{\pm0.15}$ | $22.49_{\pm0.07}$ | $16.41_{\pm0.15}$ | $37.57_{\pm0.12}$ | $45.33_{\pm0.52}$ | $62.49_{\pm0.24}$ | $59.99_{\pm0.38}$ | $80.15_{\pm0.21}$ |
| Model | $16.17_{\pm0.25}$ | $20.08_{\pm0.17}$ | $10.65_{\pm0.19}$ | $32.83_{\pm0.60}$ | $35.86_{\pm0.19}$ | $56.73_{\pm0.32}$ | $49.66_{\pm0.34}$ | $72.44_{\pm0.15}$ |
| Weight Gaussian | $16.60_{\pm0.16}$ | $20.29_{\pm0.11}$ | $10.53_{\pm0.24}$ | $30.76_{\pm0.55}$ | $42.97_{\pm0.54}$ | $60.68_{\pm0.24}$ | $54.20_{\pm0.11}$ | $75.71_{\pm0.13}$ |
| Weight DropConnect | $15.82_{\pm0.06}$ | $19.53_{\pm0.04}$ | $12.20_{\pm0.20}$ | $31.29_{\pm0.63}$ | $42.08_{\pm0.75}$ | $59.16_{\pm0.30}$ | $54.33_{\pm0.70}$ | $75.70_{\pm0.43}$ |
| Top-2 Direct Combination | $12.57_{\pm0.18}$ | $16.04_{\pm0.19}$ | $4.78_{\pm0.12}$ | $16.47_{\pm0.24}$ | $24.69_{\pm0.22}$ | $43.04_{\pm0.24}$ | $37.18_{\pm0.04}$ | $58.16_{\pm0.12}$ |
| Top-3 Direct Combination | $13.49_{\pm0.19}$ | $16.95_{\pm0.10}$ | $5.07_{\pm0.07}$ | $17.72_{\pm0.07}$ | $24.88_{\pm0.13}$ | $42.99_{\pm0.21}$ | $37.43_{\pm0.12}$ | $58.73_{\pm0.19}$ |
| Top-2 Optimised Combination | $12.14_{\pm0.10}$ | $15.51_{\pm0.11}$ | $5.15_{\pm0.22}$ | $15.63_{\pm0.27}$ | $25.28_{\pm0.15}$ | $43.86_{\pm0.16}$ | $36.88_{\pm0.08}$ | $57.90_{\pm0.24}$ |
| Top-3 Optimised Combination | $12.66_{\pm0.04}$ | $16.08_{\pm0.04}$ | $6.60_{\pm0.16}$ | $17.91_{\pm0.63}$ | $26.19_{\pm0.23}$ | $43.84_{\pm0.30}$ | $36.59_{\pm0.06}$ | $60.36_{\pm0.17}$ |

Table 6: CV classification: Error ($\downarrow$, %) comparison on in-distribution (ID) and out-of-distribution (OOD) test sets and with tuned hyperparameters.

| Noise Type | SVHN | | CIFAR-10 | | CIFAR-100 | | TinyImageNet | |
|---|---|---|---|---|---|---|---|---|
| | ID | OOD | ID | OOD | ID | OOD | ID | OOD |
| No Noise | $13.10_{\pm0.05}$ | $15.59_{\pm0.06}$ | $4.02_{\pm0.15}$ | $16.40_{\pm0.36}$ | $5.76_{\pm0.25}$ | $7.43_{\pm0.09}$ | $16.43_{\pm0.27}$ | $10.11_{\pm0.11}$ |
| Input Weak Aug. | $6.35_{\pm0.19}$ | $7.85_{\pm0.24}$ | $1.92_{\pm0.23}$ | $15.91_{\pm0.67}$ | $4.83_{\pm0.19}$ | $13.28_{\pm0.14}$ | $6.49_{\pm0.28}$ | $11.46_{\pm0.25}$ |
| Input Gaussian | $13.24_{\pm0.17}$ | $15.67_{\pm0.15}$ | $4.22_{\pm0.38}$ | $16.65_{\pm0.59}$ | $5.42_{\pm0.46}$ | $7.70_{\pm0.48}$ | $15.96_{\pm1.37}$ | $10.15_{\pm0.45}$ |
| Input ODS | $13.03_{\pm0.13}$ | $15.43_{\pm0.12}$ | $4.26_{\pm0.13}$ | $15.38_{\pm0.27}$ | $5.86_{\pm0.30}$ | $7.48_{\pm0.17}$ | $26.97_{\pm19.11}$ | $28.20_{\pm26.53}$ |
| Input AugMix | $4.81_{\pm0.07}$ | $6.03_{\pm0.09}$ | $1.18_{\pm0.06}$ | $6.27_{\pm0.39}$ | $4.41_{\pm0.39}$ | $11.38_{\pm0.24}$ | $4.87_{\pm0.31}$ | $15.78_{\pm0.31}$ |
| Input-Target MixUp | $2.81_{\pm0.08}$ | $3.49_{\pm0.07}$ | $5.41_{\pm3.04}$ | $7.31_{\pm0.66}$ | $14.20_{\pm0.13}$ | $8.57_{\pm0.42}$ | $16.33_{\pm1.40}$ | $10.26_{\pm0.32}$ |
| Label Smoothing | $8.66_{\pm0.10}$ | $10.61_{\pm0.08}$ | $5.11_{\pm0.12}$ | $9.46_{\pm0.02}$ | $21.55_{\pm0.15}$ | $17.10_{\pm0.15}$ | $29.45_{\pm0.17}$ | $16.62_{\pm0.03}$ |
| Activation Gaussian | $13.09_{\pm0.14}$ | $15.56_{\pm0.09}$ | $5.50_{\pm0.13}$ | $18.80_{\pm0.25}$ | $5.12_{\pm0.32}$ | $7.58_{\pm0.25}$ | $14.77_{\pm0.76}$ | $9.83_{\pm0.29}$ |
| Activation Dropout | $5.33_{\pm0.19}$ | $6.61_{\pm0.17}$ | $4.48_{\pm0.15}$ | $18.85_{\pm0.62}$ | $5.48_{\pm0.91}$ | $8.64_{\pm0.85}$ | $10.31_{\pm0.66}$ | $20.13_{\pm0.86}$ |
| Gradient Gaussian | $14.84_{\pm0.09}$ | $18.39_{\pm0.04}$ | $6.03_{\pm0.27}$ | $18.79_{\pm0.14}$ | $5.54_{\pm0.42}$ | $9.70_{\pm0.27}$ | $23.56_{\pm0.24}$ | $34.33_{\pm0.22}$ |
| Model | $10.93_{\pm0.19}$ | $12.82_{\pm0.13}$ | $4.42_{\pm0.01}$ | $18.43_{\pm0.54}$ | $9.06_{\pm0.32}$ | $11.00_{\pm0.39}$ | $10.80_{\pm0.24}$ | $9.01_{\pm0.03}$ |
| Weight Gaussian | $13.38_{\pm0.12}$ | $15.72_{\pm0.08}$ | $6.48_{\pm0.24}$ | $21.54_{\pm0.66}$ | $5.99_{\pm0.26}$ | $7.79_{\pm0.36}$ | $14.95_{\pm1.03}$ | $9.98_{\pm0.33}$ |
| Weight DropConnect | $12.51_{\pm0.10}$ | $14.87_{\pm0.05}$ | $4.94_{\pm0.20}$ | $16.99_{\pm0.66}$ | $5.79_{\pm0.36}$ | $8.21_{\pm0.47}$ | $15.50_{\pm0.83}$ | $10.12_{\pm0.35}$ |
| Top-2 Direct Combination | $1.79_{\pm0.15}$ | $2.76_{\pm0.17}$ | $1.25_{\pm0.15}$ | $6.91_{\pm0.26}$ | $6.19_{\pm0.19}$ | $14.21_{\pm0.44}$ | $4.04_{\pm0.27}$ | $15.69_{\pm0.34}$ |
| Top-3 Direct Combination | $1.31_{\pm0.17}$ | $1.86_{\pm0.09}$ | $2.08_{\pm0.12}$ | $8.57_{\pm0.33}$ | $6.66_{\pm0.04}$ | $14.67_{\pm0.21}$ | $13.05_{\pm1.09}$ | $22.67_{\pm1.96}$ |
| Top-2 Optimised Combination | $2.55_{\pm0.07}$ | $3.48_{\pm0.10}$ | $0.96_{\pm0.13}$ | $5.80_{\pm0.16}$ | $6.68_{\pm0.22}$ | $14.54_{\pm0.46}$ | $3.21_{\pm0.39}$ | $15.33_{\pm0.05}$ |
| Top-3 Optimised Combination | $2.89_{\pm0.11}$ | $3.98_{\pm0.08}$ | $1.15_{\pm0.34}$ | $4.97_{\pm1.05}$ | $5.60_{\pm0.31}$ | $12.56_{\pm0.61}$ | $9.90_{\pm0.42}$ | $20.11_{\pm0.26}$ |

Table 7: CV classification: ECE ($\downarrow$, %) comparison on in-distribution (ID) and out-of-distribution (OOD) test sets and with tuned hyperparameters.

| Noise Type | SVHN | | CIFAR-10 | | CIFAR-100 | | TinyImageNet | |
|---|---|---|---|---|---|---|---|---|
| | ID | OOD | ID | OOD | ID | OOD | ID | OOD |
| No Noise | $1.43_{\pm0.00}$ | $1.64_{\pm0.01}$ | $0.42_{\pm0.01}$ | $1.21_{\pm0.02}$ | $1.85_{\pm0.03}$ | $2.69_{\pm0.03}$ | $2.80_{\pm0.02}$ | $3.90_{\pm0.01}$ |
| Input Weak Aug. | $0.62_{\pm0.01}$ | $0.75_{\pm0.01}$ | $0.20_{\pm0.00}$ | $1.10_{\pm0.03}$ | $1.07_{\pm0.01}$ | $2.44_{\pm0.00}$ | $1.84_{\pm0.02}$ | $3.46_{\pm0.02}$ |
| Input Gaussian | $1.43_{\pm0.01}$ | $1.65_{\pm0.01}$ | $0.42_{\pm0.01}$ | $1.21_{\pm0.03}$ | $1.84_{\pm0.08}$ | $2.67_{\pm0.08}$ | $2.75_{\pm0.06}$ | $3.86_{\pm0.02}$ |
| Input ODS | $1.40_{\pm0.02}$ | $1.60_{\pm0.02}$ | $0.42_{\pm0.01}$ | $1.14_{\pm0.02}$ | $1.84_{\pm0.03}$ | $2.69_{\pm0.03}$ | $4.79_{\pm2.79}$ | $6.23_{\pm3.27}$ |
| Input AugMix | $0.49_{\pm0.01}$ | $0.61_{\pm0.00}$ | $0.25_{\pm0.00}$ | $0.63_{\pm0.01}$ | $1.17_{\pm0.00}$ | $2.04_{\pm0.01}$ | $1.87_{\pm0.01}$ | $3.14_{\pm0.01}$ |
| Input-Target MixUp | $0.52_{\pm0.00}$ | $0.64_{\pm0.00}$ | $0.40_{\pm0.03}$ | $0.91_{\pm0.02}$ | $2.03_{\pm0.00}$ | $2.77_{\pm0.01}$ | $2.79_{\pm0.07}$ | $3.88_{\pm0.04}$ |
| Label Smoothing | $0.75_{\pm0.01}$ | $0.90_{\pm0.01}$ | $0.46_{\pm0.01}$ | $1.09_{\pm0.01}$ | $2.18_{\pm0.02}$ | $2.87_{\pm0.03}$ | $3.25_{\pm0.02}$ | $4.15_{\pm0.01}$ |
| Activation Gaussian | $1.41_{\pm0.00}$ | $1.63_{\pm0.00}$ | $0.43_{\pm0.00}$ | $1.27_{\pm0.01}$ | $1.84_{\pm0.01}$ | $2.71_{\pm0.01}$ | $2.74_{\pm0.02}$ | $3.87_{\pm0.01}$ |
| Activation Dropout | $0.51_{\pm0.01}$ | $0.63_{\pm0.01}$ | $0.33_{\pm0.01}$ | $1.26_{\pm0.04}$ | $1.71_{\pm0.03}$ | $2.58_{\pm0.04}$ | $1.97_{\pm0.02}$ | $3.69_{\pm0.05}$ |
| Gradient Gaussian | $1.76_{\pm0.01}$ | $2.12_{\pm0.01}$ | $0.56_{\pm0.01}$ | $1.39_{\pm0.01}$ | $1.86_{\pm0.02}$ | $2.76_{\pm0.01}$ | $3.20_{\pm0.02}$ | $5.36_{\pm0.03}$ |
| Model | $1.02_{\pm0.01}$ | $1.17_{\pm0.01}$ | $0.37_{\pm0.00}$ | $1.27_{\pm0.03}$ | $1.60_{\pm0.02}$ | $2.63_{\pm0.02}$ | $2.38_{\pm0.01}$ | $3.64_{\pm0.01}$ |
| Weight Gaussian | $1.44_{\pm0.02}$ | $1.65_{\pm0.02}$ | $0.44_{\pm0.01}$ | $1.41_{\pm0.04}$ | $1.79_{\pm0.03}$ | $2.66_{\pm0.02}$ | $2.76_{\pm0.03}$ | $3.89_{\pm0.02}$ |
| Weight DropConnect | $1.32_{\pm0.01}$ | $1.52_{\pm0.01}$ | $0.43_{\pm0.01}$ | $1.21_{\pm0.03}$ | $1.73_{\pm0.04}$ | $2.58_{\pm0.02}$ | $2.78_{\pm0.06}$ | $3.90_{\pm0.04}$ |
| Top-2 Direct Combination | $0.44_{\pm0.00}$ | $0.54_{\pm0.01}$ | $0.16_{\pm0.00}$ | $0.57_{\pm0.01}$ | $0.97_{\pm0.00}$ | $1.96_{\pm0.02}$ | $1.60_{\pm0.00}$ | $2.98_{\pm0.01}$ |
| Top-3 Direct Combination | $0.45_{\pm0.00}$ | $0.55_{\pm0.00}$ | $0.17_{\pm0.00}$ | $0.63_{\pm0.01}$ | $0.97_{\pm0.01}$ | $1.97_{\pm0.01}$ | $1.73_{\pm0.02}$ | $3.25_{\pm0.10}$ |
| Top-2 Optimised Combination | $0.43_{\pm0.00}$ | $0.54_{\pm0.00}$ | $0.17_{\pm0.01}$ | $0.53_{\pm0.01}$ | $0.98_{\pm0.01}$ | $2.00_{\pm0.02}$ | $1.58_{\pm0.01}$ | $2.96_{\pm0.02}$ |
| Top-3 Optimised Combination | $0.44_{\pm0.00}$ | $0.55_{\pm0.00}$ | $0.21_{\pm0.00}$ | $0.57_{\pm0.00}$ | $0.99_{\pm0.01}$ | $1.93_{\pm0.03}$ | $1.63_{\pm0.01}$ | $3.25_{\pm0.01}$ |

Table 8: CV classification: NLL ($\downarrow$) comparison on in-distribution (ID) and out-of-distribution (OOD) test sets and with tuned hyperparameters.

| Noise Type | Scores | | | Ranks | | |
|---|---|---|---|---|---|---|
| | ID | OOD | Sketch | ID | OOD | Sketch |
| No Noise | $54.15_{\pm0.36}$ | $75.65_{\pm0.19}$ | $89.61_{\pm0.35}$ | 13.0 | 13.0 | 16.0 |
| Input Weak Aug. | $39.73_{\pm0.27}$ | $67.73_{\pm0.13}$ | $85.90_{\pm0.56}$ | 5.0 | 6.0 | 6.0 |
| Input Gaussian | $53.28_{\pm0.24}$ | $75.12_{\pm0.06}$ | $89.29_{\pm0.07}$ | 10.0 | 10.0 | 13.0 |
| Input ODS | $66.47_{\pm15.18}$ | $82.46_{\pm8.72}$ | $89.54_{\pm0.61}$ | 17.0 | 17.0 | 15.0 |
| Input AugMix | $42.49_{\pm0.24}$ | $61.58_{\pm0.16}$ | $83.12_{\pm0.23}$ | 6.0 | 5.0 | 4.0 |
| Input-Target MixUp | $54.02_{\pm0.53}$ | $75.33_{\pm0.44}$ | $89.27_{\pm0.09}$ | 12.0 | 11.0 | 12.0 |
| Label Smoothing | $52.47_{\pm0.21}$ | $73.96_{\pm0.15}$ | $88.15_{\pm0.18}$ | 9.0 | 9.0 | 9.0 |
| Activation Gaussian | $53.96_{\pm0.16}$ | $75.45_{\pm0.14}$ | $89.01_{\pm0.35}$ | 11.0 | 12.0 | 10.0 |
| Activation Dropout | $43.26_{\pm0.35}$ | $69.32_{\pm0.32}$ | $87.60_{\pm0.61}$ | 7.0 | 7.0 | 8.0 |
| Gradient Gaussian | $59.99_{\pm0.38}$ | $80.15_{\pm0.21}$ | $92.16_{\pm0.16}$ | 16.0 | 16.0 | 17.0 |
| Model | $49.66_{\pm0.34}$ | $72.44_{\pm0.15}$ | $86.98_{\pm0.15}$ | 8.0 | 8.0 | 7.0 |
| Weight Gaussian | $54.20_{\pm0.11}$ | $75.71_{\pm0.13}$ | $89.46_{\pm0.32}$ | 14.0 | 15.0 | 14.0 |
| Weight DropConnect | $54.33_{\pm0.70}$ | $75.70_{\pm0.43}$ | $89.24_{\pm0.27}$ | 15.0 | 14.0 | 11.0 |
| Top-2 Direct Combination | $37.18_{\pm0.04}$ | $58.16_{\pm0.12}$ | $82.52_{\pm0.65}$ | 3.0 | 2.0 | 2.0 |
| Top-3 Direct Combination | $37.43_{\pm0.12}$ | $58.73_{\pm0.19}$ | $83.46_{\pm0.17}$ | 4.0 | 3.0 | 5.0 |
| Top-2 Optimised Combination | $36.88_{\pm0.08}$ | $57.90_{\pm0.24}$ | $82.30_{\pm0.15}$ | 2.0 | 1.0 | 1.0 |
| Top-3 Optimised Combination | $36.59_{\pm0.06}$ | $60.36_{\pm0.17}$ | $83.02_{\pm0.46}$ | 1.0 | 4.0 | 3.0 |

Table 9: TinyImageNet classification: Error ($\downarrow, \%$) comparison on in-distribution (ID) and out-of-distribution (OOD) and Sketch test sets and with tuned hyperparameters.

| Noise Type | Scores | | | Ranks | | |
|---|---|---|---|---|---|---|
| | ID | OOD | Sketch | ID | OOD | Sketch |
| No Noise | $16.43_{\pm0.27}$ | $10.11_{\pm0.11}$ | $8.67_{\pm0.46}$ | 14.0 | 4.0 | 5.0 |
| Input Weak Aug. | $6.49_{\pm0.28}$ | $11.46_{\pm0.25}$ | $18.10_{\pm0.52}$ | 4.0 | 8.0 | 10.0 |
| Input Gaussian | $15.96_{\pm1.37}$ | $10.15_{\pm0.45}$ | $7.53_{\pm1.13}$ | 12.0 | 6.0 | 3.0 |
| Input ODS | $26.97_{\pm19.11}$ | $28.20_{\pm26.53}$ | $11.31_{\pm1.74}$ | 16.0 | 16.0 | 9.0 |
| Input AugMix | $4.87_{\pm0.31}$ | $15.78_{\pm0.31}$ | $20.08_{\pm0.82}$ | 3.0 | 11.0 | 11.0 |
| Input-Target MixUp | $16.33_{\pm1.40}$ | $10.26_{\pm0.32}$ | $7.28_{\pm1.06}$ | 13.0 | 7.0 | 2.0 |
| Label Smoothing | $29.45_{\pm0.17}$ | $16.62_{\pm0.03}$ | $5.28_{\pm0.21}$ | 17.0 | 12.0 | 1.0 |
| Activation Gaussian | $14.77_{\pm0.76}$ | $9.83_{\pm0.29}$ | $9.04_{\pm0.99}$ | 9.0 | 2.0 | 7.0 |
| Activation Dropout | $10.31_{\pm0.66}$ | $20.13_{\pm0.86}$ | $34.71_{\pm2.38}$ | 6.0 | 14.0 | 15.0 |
| Gradient Gaussian | $23.56_{\pm0.24}$ | $34.33_{\pm0.22}$ | $47.82_{\pm1.24}$ | 15.0 | 17.0 | 17.0 |
| Model | $10.80_{\pm0.24}$ | $9.01_{\pm0.03}$ | $9.90_{\pm0.12}$ | 7.0 | 1.0 | 8.0 |
| Weight Gaussian | $14.95_{\pm1.03}$ | $9.98_{\pm0.33}$ | $8.99_{\pm1.99}$ | 10.0 | 3.0 | 6.0 |
| Weight DropConnect | $15.50_{\pm0.83}$ | $10.12_{\pm0.35}$ | $7.76_{\pm0.72}$ | 11.0 | 5.0 | 4.0 |
| Top-2 Direct Combination | $4.04_{\pm0.27}$ | $15.69_{\pm0.34}$ | $25.12_{\pm0.91}$ | 2.0 | 10.0 | 13.0 |
| Top-3 Direct Combination | $13.05_{\pm1.09}$ | $22.67_{\pm1.96}$ | $39.54_{\pm0.43}$ | 8.0 | 15.0 | 16.0 |
| Top-2 Optimised Combination | $3.21_{\pm0.39}$ | $15.33_{\pm0.05}$ | $24.93_{\pm1.23}$ | 1.0 | 9.0 | 12.0 |
| Top-3 Optimised Combination | $9.90_{\pm0.42}$ | $20.11_{\pm0.26}$ | $34.64_{\pm0.59}$ | 5.0 | 13.0 | 14.0 |

Table 10: TinyImageNet classification: ECE ($\downarrow, \%$) comparison on in-distribution (ID) and out-of-distribution (OOD) and Sketch test sets and with tuned hyperparameters.

| Noise Type | Scores | | | Ranks | | |
|---|---|---|---|---|---|---|
| | ID | OOD | Sketch | ID | OOD | Sketch |
| No Noise | $2.80_{\pm0.02}$ | $3.90_{\pm0.01}$ | $4.68_{\pm0.01}$ | 14.0 | 14.0 | 8.0 |
| Input Weak Aug. | $1.84_{\pm0.02}$ | $3.46_{\pm0.02}$ | $4.69_{\pm0.02}$ | 5.0 | 6.0 | 9.0 |
| Input Gaussian | $2.75_{\pm0.06}$ | $3.86_{\pm0.02}$ | $4.65_{\pm0.02}$ | 10.0 | 9.0 | 5.0 |
| Input ODS | $4.79_{\pm2.79}$ | $6.23_{\pm3.27}$ | $4.72_{\pm0.06}$ | 17.0 | 17.0 | 12.0 |
| Input AugMix | $1.87_{\pm0.01}$ | $3.14_{\pm0.01}$ | $4.52_{\pm0.01}$ | 6.0 | 3.0 | 1.0 |
| Input-Target MixUp | $2.79_{\pm0.07}$ | $3.88_{\pm0.04}$ | $4.66_{\pm0.01}$ | 13.0 | 11.0 | 6.0 |
| Label Smoothing | $3.25_{\pm0.02}$ | $4.15_{\pm0.01}$ | $4.70_{\pm0.00}$ | 16.0 | 15.0 | 10.0 |
| Activation Gaussian | $2.74_{\pm0.02}$ | $3.87_{\pm0.01}$ | $4.64_{\pm0.03}$ | 9.0 | 10.0 | 3.0 |
| Activation Dropout | $1.97_{\pm0.02}$ | $3.69_{\pm0.05}$ | $5.48_{\pm0.19}$ | 7.0 | 8.0 | 15.0 |
| Gradient Gaussian | $3.20_{\pm0.02}$ | $5.36_{\pm0.03}$ | $7.53_{\pm0.11}$ | 15.0 | 16.0 | 17.0 |
| Model | $2.38_{\pm0.01}$ | $3.64_{\pm0.01}$ | $4.54_{\pm0.01}$ | 8.0 | 7.0 | 2.0 |
| Weight Gaussian | $2.76_{\pm0.03}$ | $3.89_{\pm0.02}$ | $4.67_{\pm0.05}$ | 11.0 | 12.0 | 7.0 |
| Weight DropConnect | $2.78_{\pm0.06}$ | $3.90_{\pm0.04}$ | $4.64_{\pm0.02}$ | 12.0 | 13.0 | 4.0 |
| Top-2 Direct Combination | $1.60_{\pm0.00}$ | $2.98_{\pm0.01}$ | $4.72_{\pm0.09}$ | 2.0 | 2.0 | 13.0 |
| Top-3 Direct Combination | $1.73_{\pm0.02}$ | $3.25_{\pm0.10}$ | $5.75_{\pm0.03}$ | 4.0 | 5.0 | 16.0 |
| Top-2 Optimised Combination | $1.58_{\pm0.01}$ | $2.96_{\pm0.02}$ | $4.71_{\pm0.05}$ | 1.0 | 1.0 | 11.0 |
| Top-3 Optimised Combination | $1.63_{\pm0.01}$ | $3.25_{\pm0.01}$ | $5.22_{\pm0.07}$ | 3.0 | 4.0 | 14.0 |

Table 11: TinyImageNet classification: NLL ($\downarrow$) comparison on in-distribution (ID), out-of-distribution (OOD) and Sketch test sets and with tuned hyperparameters.

| Metric | ID vs OOD | ID vs Sketch | OOD vs Sketch |
|---|---|---|---|
| Error | $0.912_{\pm0.000}$ | $0.794_{\pm0.000}$ | $0.824_{\pm0.000}$ |
| ECE | $0.118_{\pm0.542}$ | $-0.397_{\pm0.027}$ | $0.397_{\pm0.027}$ |
| NLL | $0.878_{\pm0.000}$ | $0.029_{\pm0.903}$ | $0.081_{\pm0.650}$ |

Table 12: TinyImageNet classification: Kendall Tau correlation between ID, OOD and Sketch rankings of different noise types.

| Noise Type | CIFAR-10 | | CIFAR-100 | | TinyImageNet | |
|---|---|---|---|---|---|---|
| | ID | OOD | ID | OOD | ID | OOD |
| No Noise | $16.09_{\pm0.18}$ | $30.81_{\pm0.76}$ | $40.97_{\pm0.40}$ | $61.56_{\pm0.11}$ | $54.70_{\pm0.81}$ | $75.96_{\pm0.52}$ |
| Input Weak Aug. | $7.99_{\pm0.07}$ | $27.60_{\pm0.42}$ | $24.03_{\pm0.09}$ | $52.82_{\pm0.13}$ | $39.60_{\pm0.26}$ | $67.41_{\pm0.02}$ |
| Input Gaussian | $16.72_{\pm0.13}$ | $30.28_{\pm0.85}$ | $41.07_{\pm0.25}$ | $61.04_{\pm0.14}$ | $54.01_{\pm0.60}$ | $75.31_{\pm0.33}$ |
| Input ODS | $15.79_{\pm0.13}$ | $29.54_{\pm0.71}$ | $41.13_{\pm0.43}$ | $60.43_{\pm0.39}$ | $52.77_{\pm0.54}$ | $74.61_{\pm0.11}$ |
| Input AugMix | $10.26_{\pm0.04}$ | $20.95_{\pm0.06}$ | $30.18_{\pm0.46}$ | $45.94_{\pm0.29}$ | $40.04_{\pm0.29}$ | $60.67_{\pm0.15}$ |
| Input-Target MixUp | $16.90_{\pm0.17}$ | $32.62_{\pm0.36}$ | $39.04_{\pm0.30}$ | $58.48_{\pm0.03}$ | $51.49_{\pm0.27}$ | $72.87_{\pm0.12}$ |
| Label Smoothing | $17.18_{\pm0.13}$ | $31.10_{\pm0.68}$ | $42.00_{\pm0.09}$ | $61.49_{\pm0.30}$ | $52.33_{\pm0.14}$ | $73.95_{\pm0.08}$ |
| Activation Gaussian | $16.34_{\pm0.15}$ | $30.52_{\pm0.79}$ | $38.98_{\pm0.19}$ | $59.70_{\pm0.22}$ | $52.49_{\pm0.23}$ | $74.62_{\pm0.21}$ |
| Activation Dropout | $12.45_{\pm0.12}$ | $27.90_{\pm0.10}$ | $31.58_{\pm0.54}$ | $56.38_{\pm0.36}$ | $51.85_{\pm0.10}$ | $74.08_{\pm0.01}$ |
| Gradient Gaussian | $18.70_{\pm0.15}$ | $34.04_{\pm0.49}$ | $47.64_{\pm0.31}$ | $67.79_{\pm0.22}$ | $55.91_{\pm0.25}$ | $76.96_{\pm0.06}$ |
| Model | $13.11_{\pm0.25}$ | $32.40_{\pm0.27}$ | $79.31_{\pm27.85}$ | $87.90_{\pm15.70}$ | $48.86_{\pm0.13}$ | $72.07_{\pm0.12}$ |
| Weight Gaussian | $16.54_{\pm0.11}$ | $30.86_{\pm0.29}$ | $37.14_{\pm0.19}$ | $58.32_{\pm0.48}$ | $52.88_{\pm0.45}$ | $74.82_{\pm0.18}$ |
| Weight DropConnect | $16.79_{\pm0.28}$ | $28.80_{\pm0.80}$ | $41.18_{\pm0.56}$ | $61.39_{\pm0.02}$ | $53.67_{\pm0.19}$ | $75.28_{\pm0.14}$ |

Table 13: CV classification: Error ($\downarrow, \%$) comparison on in-distribution (ID) and out-of-distribution (OOD) test sets and with hyperparameters transferred across datasets.

| Noise Type | CIFAR-10 | | CIFAR-100 | | TinyImageNet | |
|---|---|---|---|---|---|---|
| | ID | OOD | ID | OOD | ID | OOD |
| No Noise | $9.76_{\pm0.07}$ | $20.73_{\pm0.67}$ | $12.77_{\pm0.86}$ | $10.73_{\pm0.38}$ | $20.11_{\pm2.25}$ | $11.66_{\pm0.63}$ |
| Input Weak Aug. | $5.40_{\pm0.09}$ | $20.98_{\pm0.62}$ | $3.65_{\pm0.38}$ | $11.12_{\pm0.59}$ | $5.97_{\pm0.40}$ | $11.75_{\pm0.48}$ |
| Input Gaussian | $10.39_{\pm0.15}$ | $20.21_{\pm0.76}$ | $12.81_{\pm0.25}$ | $10.66_{\pm0.16}$ | $16.74_{\pm0.67}$ | $10.72_{\pm0.16}$ |
| Input ODS | $9.89_{\pm0.12}$ | $20.40_{\pm0.75}$ | $11.18_{\pm2.18}$ | $10.28_{\pm1.07}$ | $15.98_{\pm0.37}$ | $10.52_{\pm0.22}$ |
| Input AugMix | $5.53_{\pm0.08}$ | $11.65_{\pm0.07}$ | $8.94_{\pm0.07}$ | $8.86_{\pm0.05}$ | $3.97_{\pm0.20}$ | $15.43_{\pm0.53}$ |
| Input-Target MixUp | $6.27_{\pm0.51}$ | $8.12_{\pm0.20}$ | $14.76_{\pm0.48}$ | $10.99_{\pm0.06}$ | $19.33_{\pm0.27}$ | $10.08_{\pm0.32}$ |
| Label Smoothing | $1.65_{\pm0.20}$ | $6.05_{\pm0.46}$ | $21.53_{\pm0.30}$ | $15.64_{\pm0.20}$ | $27.18_{\pm0.27}$ | $15.49_{\pm0.11}$ |
| Activation Gaussian | $10.03_{\pm0.06}$ | $20.36_{\pm0.69}$ | $5.07_{\pm0.19}$ | $9.29_{\pm0.35}$ | $14.04_{\pm0.72}$ | $9.88_{\pm0.11}$ |
| Activation Dropout | $9.41_{\pm0.17}$ | $21.79_{\pm0.11}$ | $7.94_{\pm0.39}$ | $19.81_{\pm0.47}$ | $16.67_{\pm0.07}$ | $11.00_{\pm0.12}$ |
| Gradient Gaussian | $12.77_{\pm0.20}$ | $24.59_{\pm0.50}$ | $8.38_{\pm0.33}$ | $18.02_{\pm0.19}$ | $13.41_{\pm0.73}$ | $10.07_{\pm0.14}$ |
| Model | $7.53_{\pm0.26}$ | $21.33_{\pm0.36}$ | $0.80_{\pm1.10}$ | $4.07_{\pm5.73}$ | $14.45_{\pm0.23}$ | $10.68_{\pm0.07}$ |
| Weight Gaussian | $10.05_{\pm0.11}$ | $20.70_{\pm0.23}$ | $11.79_{\pm1.29}$ | $22.28_{\pm1.77}$ | $15.55_{\pm0.55}$ | $10.38_{\pm0.30}$ |
| Weight DropConnect | $10.93_{\pm0.26}$ | $19.75_{\pm0.67}$ | $11.24_{\pm0.70}$ | $9.93_{\pm0.18}$ | $18.10_{\pm0.20}$ | $11.11_{\pm0.10}$ |

Table 14: CV classification: ECE ($\downarrow, \%$) comparison on in-distribution (ID) and out-of-distribution (OOD) test sets and with hyperparameters transferred across datasets.

| Noise Type | CIFAR-10 | | CIFAR-100 | | TinyImageNet | |
|---|---|---|---|---|---|---|
| | ID | OOD | ID | OOD | ID | OOD |
| No Noise | $0.71_{\pm 0.01}$ | $1.55_{\pm 0.05}$ | $1.88_{\pm 0.02}$ | $2.87_{\pm 0.01}$ | $2.96_{\pm 0.12}$ | $3.99_{\pm 0.06}$ |
| Input Weak Aug. | $0.40_{\pm 0.00}$ | $1.84_{\pm 0.09}$ | $1.02_{\pm 0.01}$ | $2.45_{\pm 0.02}$ | $1.80_{\pm 0.01}$ | $3.43_{\pm 0.02}$ |
| Input Gaussian | $0.72_{\pm 0.01}$ | $1.50_{\pm 0.06}$ | $1.89_{\pm 0.01}$ | $2.84_{\pm 0.01}$ | $2.81_{\pm 0.06}$ | $3.91_{\pm 0.03}$ |
| Input ODS | $0.71_{\pm 0.00}$ | $1.55_{\pm 0.07}$ | $1.87_{\pm 0.07}$ | $2.80_{\pm 0.05}$ | $2.72_{\pm 0.02}$ | $3.85_{\pm 0.00}$ |
| Input AugMix | $0.39_{\pm 0.00}$ | $0.84_{\pm 0.00}$ | $1.30_{\pm 0.01}$ | $2.02_{\pm 0.01}$ | $1.73_{\pm 0.01}$ | $3.10_{\pm 0.02}$ |
| Input-Target MixUp | $0.56_{\pm 0.01}$ | $1.04_{\pm 0.02}$ | $1.71_{\pm 0.01}$ | $2.59_{\pm 0.00}$ | $2.62_{\pm 0.02}$ | $3.65_{\pm 0.01}$ |
| Label Smoothing | $0.56_{\pm 0.00}$ | $1.01_{\pm 0.02}$ | $2.16_{\pm 0.01}$ | $3.04_{\pm 0.01}$ | $3.16_{\pm 0.00}$ | $4.10_{\pm 0.01}$ |
| Activation Gaussian | $0.70_{\pm 0.01}$ | $1.51_{\pm 0.06}$ | $1.66_{\pm 0.01}$ | $2.73_{\pm 0.02}$ | $2.64_{\pm 0.03}$ | $3.81_{\pm 0.02}$ |
| Activation Dropout | $0.74_{\pm 0.02}$ | $1.96_{\pm 0.02}$ | $1.24_{\pm 0.02}$ | $2.69_{\pm 0.03}$ | $2.70_{\pm 0.01}$ | $3.83_{\pm 0.00}$ |
| Gradient Gaussian | $0.93_{\pm 0.01}$ | $1.99_{\pm 0.04}$ | $1.93_{\pm 0.01}$ | $3.20_{\pm 0.02}$ | $2.83_{\pm 0.02}$ | $3.98_{\pm 0.01}$ |
| Model | $0.52_{\pm 0.00}$ | $1.43_{\pm 0.02}$ | $3.59_{\pm 1.44}$ | $4.07_{\pm 0.75}$ | $2.46_{\pm 0.01}$ | $3.69_{\pm 0.01}$ |
| Weight Gaussian | $0.72_{\pm 0.01}$ | $1.54_{\pm 0.03}$ | $1.54_{\pm 0.03}$ | $2.81_{\pm 0.07}$ | $2.72_{\pm 0.04}$ | $3.86_{\pm 0.02}$ |
| Weight DropConnect | $0.79_{\pm 0.01}$ | $1.48_{\pm 0.06}$ | $1.86_{\pm 0.01}$ | $2.85_{\pm 0.00}$ | $2.83_{\pm 0.01}$ | $3.92_{\pm 0.00}$ |

Table 15: CV classification: NLL ($\downarrow$) comparison on in-distribution (ID) and out-of-distribution (OOD) test sets and with hyperparameters transferred across datasets.

| Noise Type | SVHN | | CIFAR-100 | | TinyImageNet | |
|---|---|---|---|---|---|---|
| | ID | OOD | ID | OOD | ID | OOD |
| No Noise | $5.12_{\pm 0.13}$ | $9.20_{\pm 0.10}$ | $43.87_{\pm 0.48}$ | $61.25_{\pm 0.46}$ | $53.63_{\pm 0.17}$ | $75.20_{\pm 0.32}$ |
| Input Weak Aug. | $4.13_{\pm 0.10}$ | $8.51_{\pm 0.23}$ | $27.33_{\pm 0.35}$ | $51.09_{\pm 0.18}$ | $38.21_{\pm 0.24}$ | $64.75_{\pm 0.29}$ |
| Input Gaussian | $5.01_{\pm 0.08}$ | $9.09_{\pm 0.06}$ | $43.05_{\pm 0.35}$ | $58.71_{\pm 0.32}$ | $54.15_{\pm 1.59}$ | $74.61_{\pm 1.09}$ |
| Input AugMix | $3.51_{\pm 0.05}$ | $8.27_{\pm 0.04}$ | $30.23_{\pm 0.06}$ | $45.51_{\pm 0.17}$ | $42.05_{\pm 0.29}$ | $59.93_{\pm 0.10}$ |
| Input-Target MixUp | $5.58_{\pm 0.12}$ | $12.75_{\pm 0.11}$ | $43.93_{\pm 0.68}$ | $59.98_{\pm 0.23}$ | $52.95_{\pm 1.26}$ | $73.82_{\pm 0.54}$ |
| Label Smoothing | $5.04_{\pm 0.03}$ | $8.88_{\pm 0.01}$ | $42.99_{\pm 0.36}$ | $57.27_{\pm 0.43}$ | $53.77_{\pm 0.32}$ | $74.25_{\pm 0.11}$ |
| Activation Gaussian | $5.14_{\pm 0.08}$ | $9.17_{\pm 0.06}$ | $43.74_{\pm 0.31}$ | $59.29_{\pm 0.65}$ | $56.32_{\pm 2.03}$ | $76.49_{\pm 1.23}$ |
| Activation Dropout | $4.37_{\pm 0.02}$ | $8.16_{\pm 0.04}$ | $42.89_{\pm 0.90}$ | $58.26_{\pm 0.34}$ | $42.47_{\pm 0.35}$ | $67.78_{\pm 0.25}$ |
| Gradient Gaussian | $6.25_{\pm 0.10}$ | $11.43_{\pm 0.14}$ | $44.39_{\pm 0.73}$ | $59.56_{\pm 0.87}$ | $57.92_{\pm 0.75}$ | $77.81_{\pm 0.34}$ |
| Model | $3.98_{\pm 0.02}$ | $8.11_{\pm 0.04}$ | $37.14_{\pm 0.33}$ | $56.97_{\pm 0.16}$ | $47.11_{\pm 0.53}$ | $69.76_{\pm 0.09}$ |
| Weight Gaussian | $5.04_{\pm 0.04}$ | $9.08_{\pm 0.06}$ | $44.28_{\pm 0.31}$ | $59.84_{\pm 0.08}$ | $54.11_{\pm 2.41}$ | $74.52_{\pm 1.42}$ |
| Weight DropConnect | $5.11_{\pm 0.11}$ | $9.03_{\pm 0.11}$ | $45.75_{\pm 0.89}$ | $59.71_{\pm 1.15}$ | $56.44_{\pm 3.41}$ | $76.12_{\pm 2.55}$ |

Table 16: CV classification: Error ($\downarrow$, %) comparison on in-distribution (ID) and out-of-distribution (OOD) test sets and with hyperparameters transferred across architectures.

| Noise Type | SVHN | | CIFAR-100 | | TinyImageNet | |
|---|---|---|---|---|---|---|
| | ID | OOD | ID | OOD | ID | OOD |
| No Noise | $2.73_{\pm0.10}$ | $5.13_{\pm0.06}$ | $5.76_{\pm0.34}$ | $7.50_{\pm0.11}$ | $14.68_{\pm0.47}$ | $9.93_{\pm0.08}$ |
| Input Weak Aug. | $2.66_{\pm0.05}$ | $5.75_{\pm0.09}$ | $6.33_{\pm0.32}$ | $15.26_{\pm0.80}$ | $8.20_{\pm1.23}$ | $9.88_{\pm0.38}$ |
| Input Gaussian | $2.68_{\pm0.05}$ | $5.09_{\pm0.03}$ | $5.07_{\pm0.61}$ | $9.59_{\pm1.63}$ | $13.79_{\pm0.49}$ | $8.75_{\pm0.46}$ |
| Input AugMix | $1.46_{\pm0.04}$ | $3.04_{\pm0.07}$ | $6.56_{\pm0.71}$ | $14.17_{\pm0.59}$ | $6.23_{\pm0.63}$ | $16.60_{\pm0.23}$ |
| Input-Target MixUp | $5.61_{\pm1.19}$ | $5.27_{\pm0.90}$ | $2.68_{\pm0.26}$ | $6.77_{\pm0.43}$ | $12.24_{\pm0.82}$ | $8.32_{\pm0.10}$ |
| Label Smoothing | $0.95_{\pm0.07}$ | $1.16_{\pm0.03}$ | $10.18_{\pm2.84}$ | $8.54_{\pm1.72}$ | $17.47_{\pm0.22}$ | $9.76_{\pm0.12}$ |
| Activation Gaussian | $2.75_{\pm0.05}$ | $5.11_{\pm0.07}$ | $5.45_{\pm1.23}$ | $10.46_{\pm1.78}$ | $10.66_{\pm3.63}$ | $8.42_{\pm0.89}$ |
| Activation Dropout | $3.03_{\pm0.02}$ | $5.72_{\pm0.02}$ | $5.49_{\pm0.32}$ | $10.00_{\pm0.41}$ | $14.68_{\pm0.17}$ | $25.16_{\pm0.75}$ |
| Gradient Gaussian | $3.61_{\pm0.11}$ | $7.07_{\pm0.15}$ | $5.59_{\pm1.20}$ | $11.16_{\pm1.73}$ | $21.10_{\pm0.45}$ | $31.33_{\pm0.35}$ |
| Model | $2.38_{\pm0.05}$ | $4.78_{\pm0.04}$ | $7.59_{\pm0.52}$ | $13.21_{\pm0.40}$ | $5.63_{\pm0.31}$ | $10.34_{\pm0.44}$ |
| Weight Gaussian | $2.67_{\pm0.03}$ | $5.06_{\pm0.02}$ | $5.30_{\pm0.20}$ | $9.87_{\pm0.53}$ | $12.42_{\pm0.87}$ | $8.24_{\pm0.34}$ |
| Weight DropConnect | $2.91_{\pm0.08}$ | $5.38_{\pm0.07}$ | $6.98_{\pm1.36}$ | $11.51_{\pm1.69}$ | $11.42_{\pm2.14}$ | $7.99_{\pm0.82}$ |

Table 17: CV classification: ECE ($\downarrow$, %) comparison on in-distribution (ID) and out-of-distribution (OOD) test sets and with hyperparameters transferred across architectures.

| Noise Type | SVHN | | CIFAR-100 | | TinyImageNet | |
|---|---|---|---|---|---|---|
| | ID | OOD | ID | OOD | ID | OOD |
| No Noise | $0.23_{\pm0.01}$ | $0.41_{\pm0.00}$ | $1.83_{\pm0.03}$ | $2.69_{\pm0.02}$ | $2.72_{\pm0.03}$ | $3.86_{\pm0.02}$ |
| Input Weak Aug. | $0.23_{\pm0.00}$ | $0.48_{\pm0.00}$ | $1.09_{\pm0.01}$ | $2.36_{\pm0.02}$ | $1.84_{\pm0.05}$ | $3.28_{\pm0.04}$ |
| Input Gaussian | $0.23_{\pm0.00}$ | $0.41_{\pm0.00}$ | $1.75_{\pm0.03}$ | $2.56_{\pm0.02}$ | $2.67_{\pm0.08}$ | $3.76_{\pm0.06}$ |
| Input AugMix | $0.16_{\pm0.00}$ | $0.31_{\pm0.00}$ | $1.19_{\pm0.01}$ | $2.05_{\pm0.00}$ | $1.90_{\pm0.01}$ | $3.05_{\pm0.01}$ |
| Input-Target MixUp | $0.24_{\pm0.01}$ | $0.46_{\pm0.01}$ | $1.76_{\pm0.03}$ | $2.58_{\pm0.01}$ | $2.56_{\pm0.03}$ | $3.68_{\pm0.03}$ |
| Label Smoothing | $0.19_{\pm0.00}$ | $0.31_{\pm0.00}$ | $1.99_{\pm0.06}$ | $2.65_{\pm0.06}$ | $2.82_{\pm0.02}$ | $3.86_{\pm0.01}$ |
| Activation Gaussian | $0.23_{\pm0.00}$ | $0.41_{\pm0.00}$ | $1.78_{\pm0.02}$ | $2.59_{\pm0.04}$ | $2.72_{\pm0.19}$ | $3.84_{\pm0.09}$ |
| Activation Dropout | $0.26_{\pm0.00}$ | $0.48_{\pm0.00}$ | $1.76_{\pm0.04}$ | $2.55_{\pm0.01}$ | $2.03_{\pm0.00}$ | $3.70_{\pm0.04}$ |
| Gradient Gaussian | $0.30_{\pm0.01}$ | $0.57_{\pm0.02}$ | $1.79_{\pm0.03}$ | $2.60_{\pm0.04}$ | $2.94_{\pm0.05}$ | $4.79_{\pm0.04}$ |
| Model | $0.18_{\pm0.00}$ | $0.35_{\pm0.00}$ | $1.59_{\pm0.01}$ | $2.67_{\pm0.01}$ | $2.10_{\pm0.02}$ | $3.46_{\pm0.00}$ |
| Weight Gaussian | $0.23_{\pm0.00}$ | $0.40_{\pm0.00}$ | $1.81_{\pm0.02}$ | $2.61_{\pm0.01}$ | $2.63_{\pm0.13}$ | $3.74_{\pm0.09}$ |
| Weight DropConnect | $0.24_{\pm0.01}$ | $0.43_{\pm0.01}$ | $1.87_{\pm0.04}$ | $2.61_{\pm0.04}$ | $2.73_{\pm0.16}$ | $3.82_{\pm0.15}$ |

Table 18: CV classification: NLL ($\downarrow$) comparison on in-distribution (ID) and out-of-distribution (OOD) test set and with hyperparameters transferred across architectures.

| Noise Type | Wine | | Toxicity | | Abalone | | Students | | Adult | |
|---|---|---|---|---|---|---|---|---|---|---|
| | ID | OOD | ID | OOD | ID | OOD | ID | OOD | ID | OOD |
| No Noise | $35.73_{\pm2.20}$ | $36.37_{\pm2.07}$ | $47.06_{\pm8.32}$ | $44.98_{\pm7.10}$ | $43.30_{\pm0.29}$ | $43.34_{\pm0.20}$ | $65.40_{\pm4.88}$ | $65.89_{\pm3.72}$ | $15.54_{\pm0.23}$ | $15.73_{\pm0.19}$ |
| Input Gaussian | $35.94_{\pm2.43}$ | $36.28_{\pm2.15}$ | $47.06_{\pm8.32}$ | $44.78_{\pm7.26}$ | $43.30_{\pm0.43}$ | $43.53_{\pm0.26}$ | $65.40_{\pm4.88}$ | $65.81_{\pm3.62}$ | $15.58_{\pm0.42}$ | $15.74_{\pm0.37}$ |
| Input ODS | $35.83_{\pm1.64}$ | $36.17_{\pm1.53}$ | $48.04_{\pm5.55}$ | $45.65_{\pm5.41}$ | $43.38_{\pm0.73}$ | $43.61_{\pm0.60}$ | $64.56_{\pm3.73}$ | $65.74_{\pm3.30}$ | $15.47_{\pm0.39}$ | $15.67_{\pm0.34}$ |
| Input-Target MixUp | $38.02_{\pm0.82}$ | $38.12_{\pm0.80}$ | $50.98_{\pm5.00}$ | $46.47_{\pm4.29}$ | $43.46_{\pm0.45}$ | $43.55_{\pm0.40}$ | $65.40_{\pm2.60}$ | $66.13_{\pm2.87}$ | $15.28_{\pm0.33}$ | $15.44_{\pm0.33}$ |
| Label Smoothing | $35.52_{\pm2.37}$ | $36.05_{\pm2.30}$ | $50.00_{\pm4.16}$ | $46.59_{\pm4.20}$ | $43.22_{\pm0.20}$ | $43.41_{\pm0.24}$ | $64.98_{\pm4.30}$ | $65.69_{\pm3.52}$ | $15.55_{\pm0.35}$ | $15.74_{\pm0.31}$ |
| Activation Gaussian | $35.83_{\pm1.74}$ | $36.12_{\pm1.49}$ | $47.06_{\pm8.32}$ | $44.82_{\pm7.43}$ | $43.22_{\pm0.54}$ | $43.41_{\pm0.29}$ | $64.56_{\pm4.51}$ | $66.46_{\pm4.38}$ | $15.77_{\pm0.40}$ | $15.91_{\pm0.42}$ |
| Activation Dropout | $36.04_{\pm2.23}$ | $36.40_{\pm1.84}$ | $56.86_{\pm15.62}$ | $56.27_{\pm15.36}$ | $43.54_{\pm1.28}$ | $43.69_{\pm1.06}$ | $71.73_{\pm3.32}$ | $72.44_{\pm3.51}$ | $14.82_{\pm0.32}$ | $14.92_{\pm0.31}$ |
| Gradient Gaussian | $32.60_{\pm0.53}$ | $33.97_{\pm1.19}$ | $50.00_{\pm4.16}$ | $49.10_{\pm1.22}$ | $43.74_{\pm0.11}$ | $43.86_{\pm0.11}$ | $68.78_{\pm5.69}$ | $68.56_{\pm4.60}$ | $15.38_{\pm0.31}$ | $15.52_{\pm0.29}$ |
| Model | $37.92_{\pm1.28}$ | $38.15_{\pm1.31}$ | $45.10_{\pm6.04}$ | $43.49_{\pm6.16}$ | $43.42_{\pm0.20}$ | $43.66_{\pm0.26}$ | $65.82_{\pm4.51}$ | $65.86_{\pm3.57}$ | $14.72_{\pm0.46}$ | $14.81_{\pm0.38}$ |
| Weight Gaussian | $35.52_{\pm1.95}$ | $36.16_{\pm1.75}$ | $50.00_{\pm6.35}$ | $49.06_{\pm5.41}$ | $43.10_{\pm0.39}$ | $43.36_{\pm0.29}$ | $64.98_{\pm4.88}$ | $66.80_{\pm4.25}$ | $15.13_{\pm0.31}$ | $15.23_{\pm0.24}$ |
| Weight DropConnect | $38.85_{\pm0.53}$ | $38.99_{\pm0.77}$ | $47.06_{\pm8.32}$ | $44.90_{\pm6.87}$ | $43.42_{\pm0.78}$ | $43.36_{\pm0.59}$ | $64.56_{\pm4.51}$ | $66.58_{\pm4.40}$ | $14.93_{\pm0.32}$ | $15.04_{\pm0.29}$ |
| Top-2 Direct Combination | $37.50_{\pm1.53}$ | $37.88_{\pm1.28}$ | $46.08_{\pm5.00}$ | $44.00_{\pm5.17}$ | $43.18_{\pm0.68}$ | $43.41_{\pm0.67}$ | $64.56_{\pm4.51}$ | $65.82_{\pm3.69}$ | $14.61_{\pm0.30}$ | $14.77_{\pm0.27}$ |
| Top-3 Direct Combination | $38.23_{\pm1.03}$ | $38.46_{\pm0.94}$ | $31.37_{\pm7.72}$ | $31.37_{\pm7.72}$ | $43.06_{\pm0.88}$ | $43.39_{\pm0.82}$ | $69.20_{\pm2.60}$ | $69.35_{\pm2.21}$ | $14.83_{\pm0.31}$ | $14.91_{\pm0.29}$ |
| Top-2 Optimised Combination | $37.60_{\pm2.12}$ | $37.79_{\pm2.36}$ | $50.00_{\pm6.35}$ | $47.02_{\pm5.47}$ | $43.22_{\pm0.56}$ | $43.40_{\pm0.32}$ | $65.82_{\pm4.51}$ | $66.23_{\pm3.77}$ | $14.64_{\pm0.35}$ | $14.74_{\pm0.33}$ |
| Top-3 Optimised Combination | $39.90_{\pm2.08}$ | $39.96_{\pm1.80}$ | $49.02_{\pm6.93}$ | $44.47_{\pm7.36}$ | $43.26_{\pm0.66}$ | $43.56_{\pm0.41}$ | $82.70_{\pm5.69}$ | $82.75_{\pm5.58}$ | $14.64_{\pm0.32}$ | $14.74_{\pm0.33}$ |

Table 19: Tabular data classification: error ($\downarrow$, %) comparison on in-distribution (ID) and out-of-distribution (OOD) test sets and with tuned hyperparameters.

| Noise Type | Wine | | Toxicity | | Abalone | | Students | | Adult | |
|---|---|---|---|---|---|---|---|---|---|---|
| | ID | OOD | ID | OOD | ID | OOD | ID | OOD | ID | OOD |
| No Noise | $8.75_{\pm1.34}$ | $9.07_{\pm0.96}$ | $45.78_{\pm8.12}$ | $44.18_{\pm6.80}$ | $3.94_{\pm0.59}$ | $4.19_{\pm0.61}$ | $12.46_{\pm2.92}$ | $13.81_{\pm2.17}$ | $3.47_{\pm0.29}$ | $3.85_{\pm0.34}$ |
| Input Gaussian | $8.56_{\pm0.60}$ | $8.82_{\pm1.13}$ | $45.84_{\pm8.40}$ | $43.93_{\pm7.07}$ | $3.60_{\pm0.41}$ | $4.05_{\pm0.67}$ | $13.36_{\pm0.44}$ | $14.37_{\pm1.68}$ | $3.50_{\pm0.44}$ | $3.84_{\pm0.42}$ |
| Input ODS | $5.75_{\pm0.71}$ | $6.02_{\pm0.52}$ | $42.82_{\pm4.17}$ | $39.93_{\pm3.67}$ | $3.04_{\pm0.75}$ | $3.55_{\pm1.01}$ | $14.04_{\pm2.38}$ | $14.07_{\pm1.77}$ | $3.64_{\pm0.46}$ | $3.89_{\pm0.41}$ |
| Input-Target MixUp | $4.43_{\pm0.77}$ | $4.76_{\pm0.07}$ | $43.18_{\pm5.18}$ | $41.22_{\pm3.89}$ | $3.41_{\pm1.00}$ | $3.86_{\pm0.92}$ | $11.07_{\pm3.74}$ | $11.74_{\pm3.19}$ | $2.79_{\pm0.36}$ | $3.06_{\pm0.43}$ |
| Label Smoothing | $4.82_{\pm0.61}$ | $5.47_{\pm1.60}$ | $42.08_{\pm4.15}$ | $38.75_{\pm4.52}$ | $2.77_{\pm0.74}$ | $3.42_{\pm0.88}$ | $12.69_{\pm2.59}$ | $13.81_{\pm1.98}$ | $3.32_{\pm0.32}$ | $3.62_{\pm0.36}$ |
| Activation Gaussian | $9.27_{\pm1.24}$ | $9.52_{\pm1.12}$ | $46.46_{\pm8.05}$ | $44.11_{\pm7.04}$ | $3.17_{\pm0.96}$ | $3.79_{\pm0.87}$ | $11.91_{\pm0.96}$ | $13.62_{\pm0.49}$ | $3.74_{\pm0.57}$ | $4.05_{\pm0.56}$ |
| Activation Dropout | $6.93_{\pm1.24}$ | $7.26_{\pm1.51}$ | $23.26_{\pm16.83}$ | $23.81_{\pm16.23}$ | $3.48_{\pm1.96}$ | $3.75_{\pm1.55}$ | $9.75_{\pm2.95}$ | $9.25_{\pm3.16}$ | $0.95_{\pm0.06}$ | $1.10_{\pm0.10}$ |
| Gradient Gaussian | $16.96_{\pm1.93}$ | $18.15_{\pm1.56}$ | $48.32_{\pm3.18}$ | $48.00_{\pm0.99}$ | $5.30_{\pm1.57}$ | $5.87_{\pm1.71}$ | $17.58_{\pm4.99}$ | $18.03_{\pm4.63}$ | $2.81_{\pm0.53}$ | $3.13_{\pm0.52}$ |
| Model | $4.73_{\pm1.73}$ | $5.30_{\pm1.36}$ | $39.66_{\pm2.95}$ | $37.93_{\pm3.67}$ | $2.80_{\pm1.09}$ | $3.40_{\pm0.90}$ | $12.35_{\pm2.63}$ | $13.65_{\pm1.99}$ | $1.49_{\pm0.51}$ | $1.62_{\pm0.43}$ |
| Weight Gaussian | $9.56_{\pm1.17}$ | $9.36_{\pm1.35}$ | $48.77_{\pm4.79}$ | $46.12_{\pm4.70}$ | $3.48_{\pm0.55}$ | $3.90_{\pm0.54}$ | $12.27_{\pm1.18}$ | $13.49_{\pm0.76}$ | $2.36_{\pm0.25}$ | $2.61_{\pm0.22}$ |
| Weight DropConnect | $5.65_{\pm3.34}$ | $5.26_{\pm2.58}$ | $46.47_{\pm7.40}$ | $44.01_{\pm6.62}$ | $3.72_{\pm1.47}$ | $3.96_{\pm1.12}$ | $12.73_{\pm1.63}$ | $13.75_{\pm0.73}$ | $1.70_{\pm0.40}$ | $1.92_{\pm0.41}$ |
| Top-2 Direct Combination | $4.58_{\pm0.47}$ | $4.19_{\pm0.42}$ | $32.77_{\pm1.78}$ | $32.93_{\pm2.41}$ | $2.80_{\pm0.81}$ | $3.07_{\pm0.86}$ | $11.78_{\pm0.40}$ | $11.99_{\pm0.40}$ | $1.39_{\pm0.33}$ | $1.50_{\pm0.32}$ |
| Top-3 Direct Combination | $3.77_{\pm0.90}$ | $3.95_{\pm0.62}$ | $15.99_{\pm9.87}$ | $16.13_{\pm9.60}$ | $2.65_{\pm1.51}$ | $2.88_{\pm1.24}$ | $12.44_{\pm1.68}$ | $12.01_{\pm1.35}$ | $0.95_{\pm0.21}$ | $1.10_{\pm0.19}$ |
| Top-2 Optimised Combination | $5.01_{\pm1.16}$ | $5.51_{\pm1.07}$ | $45.30_{\pm7.41}$ | $43.41_{\pm6.21}$ | $2.98_{\pm0.83}$ | $3.69_{\pm0.87}$ | $10.68_{\pm0.62}$ | $11.72_{\pm0.08}$ | $1.25_{\pm0.38}$ | $1.43_{\pm0.35}$ |
| Top-3 Optimised Combination | $7.39_{\pm1.06}$ | $7.26_{\pm1.48}$ | $38.88_{\pm6.65}$ | $36.78_{\pm6.77}$ | $3.08_{\pm1.34}$ | $3.62_{\pm0.92}$ | $2.56_{\pm1.82}$ | $2.70_{\pm1.82}$ | $1.17_{\pm0.19}$ | $1.38_{\pm0.23}$ |

Table 20: Tabular data classification: ECE ($\downarrow$, %) comparison on in-distribution (ID) and out-of-distribution (OOD) test sets and with tuned hyperparameters.

| Noise Type | Wine | | Toxicity | | Abalone | | Students | | Adult | |
|---|---|---|---|---|---|---|---|---|---|---|
| | ID | OOD | ID | OOD | ID | OOD | ID | OOD | ID | OOD |
| No Noise | $0.94_{\pm0.05}$ | $0.95_{\pm0.04}$ | $4.85_{\pm0.23}$ | $4.87_{\pm0.27}$ | $0.84_{\pm0.02}$ | $0.86_{\pm0.02}$ | $1.87_{\pm0.08}$ | $1.98_{\pm0.07}$ | $0.35_{\pm0.01}$ | $0.36_{\pm0.01}$ |
| Input Gaussian | $0.94_{\pm0.05}$ | $0.95_{\pm0.05}$ | $4.85_{\pm0.27}$ | $4.87_{\pm0.28}$ | $0.84_{\pm0.02}$ | $0.86_{\pm0.02}$ | $1.87_{\pm0.08}$ | $1.97_{\pm0.07}$ | $0.34_{\pm0.01}$ | $0.36_{\pm0.01}$ |
| Input ODS | $0.91_{\pm0.05}$ | $0.92_{\pm0.05}$ | $2.20_{\pm0.33}$ | $2.35_{\pm0.39}$ | $0.84_{\pm0.02}$ | $0.85_{\pm0.02}$ | $1.87_{\pm0.08}$ | $1.98_{\pm0.07}$ | $0.35_{\pm0.01}$ | $0.36_{\pm0.01}$ |
| Input-Target MixUp | $0.91_{\pm0.04}$ | $0.91_{\pm0.04}$ | $2.39_{\pm0.27}$ | $2.44_{\pm0.19}$ | $0.84_{\pm0.02}$ | $0.86_{\pm0.02}$ | $1.88_{\pm0.04}$ | $1.96_{\pm0.04}$ | $0.34_{\pm0.01}$ | $0.35_{\pm0.01}$ |
| Label Smoothing | $0.93_{\pm0.04}$ | $0.94_{\pm0.04}$ | $1.76_{\pm0.22}$ | $1.78_{\pm0.31}$ | $0.84_{\pm0.02}$ | $0.85_{\pm0.02}$ | $1.87_{\pm0.08}$ | $1.97_{\pm0.07}$ | $0.34_{\pm0.01}$ | $0.35_{\pm0.01}$ |
| Activation Gaussian | $0.94_{\pm0.05}$ | $0.96_{\pm0.05}$ | $4.78_{\pm0.26}$ | $4.82_{\pm0.30}$ | $0.84_{\pm0.02}$ | $0.86_{\pm0.02}$ | $1.90_{\pm0.09}$ | $2.00_{\pm0.08}$ | $0.35_{\pm0.01}$ | $0.36_{\pm0.01}$ |
| Activation Dropout | $0.92_{\pm0.05}$ | $0.93_{\pm0.05}$ | $1.03_{\pm0.24}$ | $1.03_{\pm0.23}$ | $0.84_{\pm0.02}$ | $0.85_{\pm0.02}$ | $2.23_{\pm0.04}$ | $2.25_{\pm0.05}$ | $0.32_{\pm0.01}$ | $0.32_{\pm0.01}$ |
| Gradient Gaussian | $1.26_{\pm0.09}$ | $1.30_{\pm0.08}$ | $5.41_{\pm0.16}$ | $5.65_{\pm0.06}$ | $0.86_{\pm0.03}$ | $0.88_{\pm0.04}$ | $1.95_{\pm0.06}$ | $2.05_{\pm0.05}$ | $0.34_{\pm0.01}$ | $0.35_{\pm0.01}$ |
| Model | $0.92_{\pm0.04}$ | $0.93_{\pm0.04}$ | $2.06_{\pm0.37}$ | $2.14_{\pm0.43}$ | $0.84_{\pm0.02}$ | $0.85_{\pm0.02}$ | $1.87_{\pm0.08}$ | $1.98_{\pm0.07}$ | $0.31_{\pm0.01}$ | $0.32_{\pm0.01}$ |
| Weight Gaussian | $0.95_{\pm0.05}$ | $0.96_{\pm0.05}$ | $3.02_{\pm0.20}$ | $3.15_{\pm0.25}$ | $0.84_{\pm0.02}$ | $0.86_{\pm0.02}$ | $1.91_{\pm0.08}$ | $2.01_{\pm0.08}$ | $0.33_{\pm0.01}$ | $0.34_{\pm0.01}$ |
| Weight DropConnect | $0.94_{\pm0.04}$ | $0.94_{\pm0.04}$ | $4.82_{\pm0.17}$ | $4.83_{\pm0.23}$ | $0.84_{\pm0.02}$ | $0.85_{\pm0.02}$ | $1.91_{\pm0.09}$ | $2.01_{\pm0.08}$ | $0.32_{\pm0.01}$ | $0.33_{\pm0.01}$ |
| Top-2 Direct Combination | $0.94_{\pm0.05}$ | $0.95_{\pm0.03}$ | $1.57_{\pm0.24}$ | $1.62_{\pm0.18}$ | $0.84_{\pm0.02}$ | $0.85_{\pm0.02}$ | $1.87_{\pm0.06}$ | $1.97_{\pm0.05}$ | $0.31_{\pm0.01}$ | $0.32_{\pm0.01}$ |
| Top-3 Direct Combination | $0.94_{\pm0.03}$ | $0.95_{\pm0.03}$ | $0.93_{\pm0.22}$ | $0.93_{\pm0.22}$ | $0.84_{\pm0.02}$ | $0.84_{\pm0.02}$ | $2.20_{\pm0.05}$ | $2.23_{\pm0.05}$ | $0.32_{\pm0.01}$ | $0.32_{\pm0.01}$ |
| Top-2 Optimised Combination | $0.92_{\pm0.05}$ | $0.93_{\pm0.04}$ | $2.47_{\pm0.41}$ | $2.50_{\pm0.45}$ | $0.84_{\pm0.02}$ | $0.85_{\pm0.02}$ | $1.86_{\pm0.05}$ | $1.96_{\pm0.05}$ | $0.31_{\pm0.01}$ | $0.32_{\pm0.01}$ |
| Top-3 Optimised Combination | $0.96_{\pm0.04}$ | $0.96_{\pm0.04}$ | $2.05_{\pm0.27}$ | $2.25_{\pm0.34}$ | $0.84_{\pm0.02}$ | $0.85_{\pm0.02}$ | $2.54_{\pm0.15}$ | $2.54_{\pm0.14}$ | $0.31_{\pm0.01}$ | $0.32_{\pm0.01}$ |

Table 21: Tabular data classification: NLL ($\downarrow$) comparison on in-distribution (ID) and out-of-distribution (OOD) test sets and with tuned hyperparameters.

| Noise Type | NewsGroup | | SST | |
|---|---|---|---|---|
| | GP-CNN | Transformer | GP-CNN | Transformer |
| No Noise | $35.67_{\pm1.13}$ | $36.56_{\pm0.83}$ | $19.15_{\pm0.75}$ | $21.60_{\pm0.24}$ |
| Input Gaussian | $35.44_{\pm0.87}$ | $36.59_{\pm0.84}$ | $21.14_{\pm0.33}$ | $21.52_{\pm0.24}$ |
| Input ODS | $33.56_{\pm0.33}$ | $34.44_{\pm0.74}$ | $18.46_{\pm1.03}$ | $21.52_{\pm1.22}$ |
| Input-Target MixUp | $35.44_{\pm0.78}$ | $36.70_{\pm0.92}$ | $18.85_{\pm0.11}$ | $21.33_{\pm0.41}$ |
| Label Smoothing | $35.59_{\pm0.69}$ | $36.56_{\pm1.10}$ | $21.75_{\pm0.53}$ | $21.29_{\pm0.38}$ |
| Activation Gaussian | $35.56_{\pm1.16}$ | $36.44_{\pm0.87}$ | $19.61_{\pm1.06}$ | $21.64_{\pm0.24}$ |
| Activation Dropout | $39.19_{\pm0.92}$ | $36.48_{\pm0.29}$ | $19.61_{\pm0.25}$ | $21.02_{\pm0.11}$ |
| Gradient Gaussian | $40.56_{\pm0.18}$ | $36.52_{\pm0.89}$ | $21.90_{\pm0.82}$ | $21.41_{\pm0.91}$ |
| Model | $40.22_{\pm0.64}$ | $36.52_{\pm0.76}$ | $19.57_{\pm0.59}$ | $21.67_{\pm0.25}$ |
| Weight Gaussian | $35.48_{\pm0.53}$ | $36.89_{\pm0.79}$ | $20.18_{\pm0.19}$ | $21.56_{\pm0.32}$ |
| Weight DropConnect | $35.19_{\pm0.69}$ | $37.04_{\pm0.68}$ | $20.15_{\pm0.81}$ | $21.18_{\pm0.39}$ |
| Top-2 Direct Combination | $39.52_{\pm0.69}$ | $36.41_{\pm0.52}$ | $18.77_{\pm1.00}$ | $20.64_{\pm0.50}$ |
| Top-3 Direct Combination | $36.96_{\pm0.73}$ | $34.67_{\pm0.74}$ | $17.58_{\pm0.44}$ | $20.22_{\pm0.61}$ |
| Top-2 Optimised Combination | $38.04_{\pm1.12}$ | $36.52_{\pm1.30}$ | $20.41_{\pm0.50}$ | $21.06_{\pm0.33}$ |
| Top-3 Optimised Combination | $36.00_{\pm0.48}$ | $35.19_{\pm0.37}$ | $18.85_{\pm0.78}$ | $19.07_{\pm0.61}$ |

Table 22: NewsGroup NLP classification: Error ($\downarrow$, %) comparison on in-distribution (ID) test set and with tuned hyperparameters.

| Noise Type | NewsGroup | | SST | |
|---|---|---|---|---|
| | **GP-CNN** | **Transformer** | **GP-CNN** | **Transformer** |
| NO NOISE | $5.12_{\pm 0.53}$ | $3.47_{\pm 0.98}$ | $13.99_{\pm 0.47}$ | $11.61_{\pm 3.97}$ |
| INPUT GAUSSIAN | $4.78_{\pm 1.30}$ | $3.59_{\pm 0.83}$ | $15.14_{\pm 0.17}$ | $11.32_{\pm 3.72}$ |
| INPUT ODS | $2.57_{\pm 0.81}$ | $7.99_{\pm 0.59}$ | $11.80_{\pm 0.93}$ | $14.70_{\pm 2.64}$ |
| INPUT-TARGET MIXUP | $5.07_{\pm 0.59}$ | $2.54_{\pm 1.27}$ | $7.76_{\pm 0.32}$ | $11.75_{\pm 3.55}$ |
| LABEL SMOOTHING | $3.78_{\pm 0.50}$ | $4.13_{\pm 0.94}$ | $9.10_{\pm 0.87}$ | $10.26_{\pm 3.30}$ |
| ACTIVATION GAUSSIAN | $5.75_{\pm 0.50}$ | $3.42_{\pm 1.05}$ | $13.99_{\pm 1.22}$ | $11.39_{\pm 3.71}$ |
| ACTIVATION DROPOUT | $7.19_{\pm 1.29}$ | $2.26_{\pm 0.20}$ | $11.50_{\pm 0.11}$ | $7.42_{\pm 1.24}$ |
| GRADIENT GAUSSIAN | $24.26_{\pm 1.01}$ | $3.24_{\pm 1.07}$ | $17.40_{\pm 0.93}$ | $12.20_{\pm 3.46}$ |
| MODEL | $2.91_{\pm 0.90}$ | $3.55_{\pm 0.84}$ | $14.00_{\pm 0.50}$ | $11.16_{\pm 3.62}$ |
| WEIGHT GAUSSIAN | $4.57_{\pm 0.48}$ | $4.01_{\pm 0.79}$ | $14.77_{\pm 0.04}$ | $12.09_{\pm 3.68}$ |
| WEIGHT DROPCONNECT | $5.41_{\pm 0.76}$ | $4.43_{\pm 0.72}$ | $16.08_{\pm 1.02}$ | $11.27_{\pm 3.01}$ |
| TOP-2 DIRECT COMBINATION | $8.25_{\pm 0.67}$ | $2.91_{\pm 0.33}$ | $4.55_{\pm 1.11}$ | $6.40_{\pm 0.93}$ |
| TOP-3 DIRECT COMBINATION | $13.76_{\pm 0.67}$ | $11.19_{\pm 0.88}$ | $2.12_{\pm 0.78}$ | $8.70_{\pm 2.85}$ |
| TOP-2 OPTIMISED COMBINATION | $3.44_{\pm 0.51}$ | $3.07_{\pm 0.52}$ | $8.86_{\pm 0.38}$ | $8.02_{\pm 2.12}$ |
| TOP-3 OPTIMISED COMBINATION | $2.75_{\pm 0.05}$ | $2.89_{\pm 0.42}$ | $9.33_{\pm 0.90}$ | $9.18_{\pm 0.61}$ |

Table 23: NewsGroup NLP classification: ECE ($\downarrow, \%$) comparison on in-distribution (ID) test set and with tuned hyperparameters.

| Noise Type | NewsGroup | | SST | |
|---|---|---|---|---|
| | **GP-CNN** | **Transformer** | **GP-CNN** | **Transformer** |
| NO NOISE | $1.14_{\pm 0.01}$ | $1.13_{\pm 0.02}$ | $0.81_{\pm 0.04}$ | $0.61_{\pm 0.10}$ |
| INPUT GAUSSIAN | $1.12_{\pm 0.01}$ | $1.13_{\pm 0.02}$ | $0.85_{\pm 0.05}$ | $0.59_{\pm 0.09}$ |
| INPUT ODS | $1.03_{\pm 0.00}$ | $1.10_{\pm 0.01}$ | $0.62_{\pm 0.02}$ | $0.73_{\pm 0.12}$ |
| INPUT-TARGET MIXUP | $1.13_{\pm 0.01}$ | $1.13_{\pm 0.02}$ | $0.48_{\pm 0.00}$ | $0.59_{\pm 0.08}$ |
| LABEL SMOOTHING | $1.12_{\pm 0.01}$ | $1.13_{\pm 0.02}$ | $0.52_{\pm 0.01}$ | $0.54_{\pm 0.05}$ |
| ACTIVATION GAUSSIAN | $1.14_{\pm 0.01}$ | $1.13_{\pm 0.02}$ | $0.83_{\pm 0.03}$ | $0.59_{\pm 0.09}$ |
| ACTIVATION DROPOUT | $1.18_{\pm 0.01}$ | $1.11_{\pm 0.02}$ | $0.55_{\pm 0.01}$ | $0.48_{\pm 0.01}$ |
| GRADIENT GAUSSIAN | $1.87_{\pm 0.07}$ | $1.13_{\pm 0.02}$ | $1.11_{\pm 0.06}$ | $0.60_{\pm 0.08}$ |
| MODEL | $1.21_{\pm 0.01}$ | $1.13_{\pm 0.02}$ | $0.85_{\pm 0.02}$ | $0.59_{\pm 0.09}$ |
| WEIGHT GAUSSIAN | $1.14_{\pm 0.01}$ | $1.13_{\pm 0.02}$ | $0.84_{\pm 0.02}$ | $0.62_{\pm 0.11}$ |
| WEIGHT DROPCONNECT | $1.13_{\pm 0.01}$ | $1.13_{\pm 0.02}$ | $1.05_{\pm 0.07}$ | $0.58_{\pm 0.07}$ |
| TOP-2 DIRECT COMBINATION | $1.19_{\pm 0.01}$ | $1.12_{\pm 0.01}$ | $0.43_{\pm 0.02}$ | $0.46_{\pm 0.01}$ |
| TOP-3 DIRECT COMBINATION | $1.18_{\pm 0.01}$ | $1.15_{\pm 0.01}$ | $0.40_{\pm 0.01}$ | $0.51_{\pm 0.04}$ |
| TOP-2 OPTIMISED COMBINATION | $1.14_{\pm 0.02}$ | $1.12_{\pm 0.01}$ | $0.51_{\pm 0.01}$ | $0.49_{\pm 0.02}$ |
| TOP-3 OPTIMISED COMBINATION | $1.09_{\pm 0.01}$ | $1.10_{\pm 0.01}$ | $0.48_{\pm 0.02}$ | $0.50_{\pm 0.01}$ |

Table 24: NewsGroup NLP classification: NLL ($\downarrow$) comparison on in-distribution (ID) test set and with tuned hyperparameters.

| Noise Type | Rotated CIFAR-100 | | WikiFace | |
|---|---|---|---|---|
| | ID | OOD | ID | OOD |
| No Noise | $0.03_{\pm0.00}$ | $0.13_{\pm0.01}$ | $0.03_{\pm0.00}$ | $0.04_{\pm0.00}$ |
| Input Weak Aug. | $0.03_{\pm0.00}$ | $0.13_{\pm0.01}$ | $0.03_{\pm0.00}$ | $0.04_{\pm0.00}$ |
| Input Gaussian | $0.03_{\pm0.00}$ | $0.14_{\pm0.01}$ | $0.04_{\pm0.00}$ | $0.05_{\pm0.00}$ |
| Input AugMix | $0.03_{\pm0.00}$ | $0.07_{\pm0.00}$ | $0.03_{\pm0.00}$ | $0.04_{\pm0.00}$ |
| Input-Target CMixUp | $0.03_{\pm0.00}$ | $0.09_{\pm0.00}$ | $0.04_{\pm0.00}$ | $0.04_{\pm0.00}$ |
| Activation Gaussian | $0.03_{\pm0.00}$ | $0.14_{\pm0.01}$ | $0.04_{\pm0.00}$ | $0.04_{\pm0.00}$ |
| Activation Dropout | $0.03_{\pm0.00}$ | $0.14_{\pm0.00}$ | $0.04_{\pm0.00}$ | $0.05_{\pm0.00}$ |
| Gradient Gaussian | $0.04_{\pm0.00}$ | $0.11_{\pm0.00}$ | $0.04_{\pm0.00}$ | $0.04_{\pm0.00}$ |
| Model | $0.04_{\pm0.00}$ | $0.16_{\pm0.01}$ | $0.04_{\pm0.00}$ | $0.04_{\pm0.00}$ |
| Weight Gaussian | $0.03_{\pm0.00}$ | $0.15_{\pm0.00}$ | $0.04_{\pm0.00}$ | $0.04_{\pm0.00}$ |
| Weight DropConnect | $0.03_{\pm0.00}$ | $0.12_{\pm0.01}$ | $0.10_{\pm0.04}$ | $0.11_{\pm0.04}$ |
| Top-2 Direct Combination | $0.03_{\pm0.00}$ | $0.08_{\pm0.00}$ | $0.04_{\pm0.00}$ | $0.04_{\pm0.00}$ |
| Top-3 Direct Combination | $0.03_{\pm0.00}$ | $0.08_{\pm0.01}$ | $0.04_{\pm0.00}$ | $0.04_{\pm0.00}$ |
| Top-2 Optimised Combination | $0.03_{\pm0.00}$ | $0.06_{\pm0.00}$ | $0.03_{\pm0.00}$ | $0.04_{\pm0.00}$ |
| Top-3 Optimised Combination | $0.24_{\pm0.15}$ | $0.29_{\pm0.10}$ | $0.04_{\pm0.00}$ | $0.04_{\pm0.00}$ |

Table 25: Rotated CV regression: MSE (↓) comparison on in-distribution (ID) and out-of-distribution (OOD) test sets and with tuned hyperparameters.

| Noise Type | Rotated CIFAR-100 | | WikiFace | |
|---|---|---|---|---|
| | ID | OOD | ID | OOD |
| No Noise | $-4.81_{\pm0.00}$ | $6.90_{\pm1.52}$ | $27.03_{\pm2.94}$ | $31.43_{\pm3.80}$ |
| Input Weak Aug. | $-4.60_{\pm0.06}$ | $3.41_{\pm0.53}$ | $-0.82_{\pm0.24}$ | $0.31_{\pm0.66}$ |
| Input Gaussian | $-4.67_{\pm0.11}$ | $7.57_{\pm2.14}$ | $27.96_{\pm2.66}$ | $31.50_{\pm2.15}$ |
| Input AugMix | $-4.82_{\pm0.01}$ | $-1.70_{\pm0.04}$ | $0.78_{\pm0.27}$ | $-0.12_{\pm0.11}$ |
| Input-Target CMixUp | $-4.62_{\pm0.04}$ | $1.86_{\pm0.35}$ | $21.83_{\pm2.35}$ | $25.62_{\pm0.93}$ |
| Activation Gaussian | $-4.27_{\pm0.07}$ | $2.94_{\pm0.18}$ | $14.93_{\pm1.09}$ | $17.61_{\pm1.01}$ |
| Activation Dropout | $-3.81_{\pm0.55}$ | $1.04_{\pm0.59}$ | $-1.35_{\pm0.02}$ | $-0.53_{\pm0.36}$ |
| Gradient Gaussian | $-3.70_{\pm0.00}$ | $-0.44_{\pm0.00}$ | $25.87_{\pm2.19}$ | $29.69_{\pm3.75}$ |
| Model | $-4.36_{\pm0.05}$ | $3.44_{\pm0.76}$ | $-1.08_{\pm0.03}$ | $-1.08_{\pm0.03}$ |
| Weight Gaussian | $-4.23_{\pm0.12}$ | $2.53_{\pm0.36}$ | $4.42_{\pm0.10}$ | $5.65_{\pm0.60}$ |
| Weight DropConnect | $-2.28_{\pm1.93}$ | $39.88_{\pm24.73}$ | $4.83_{\pm3.37}$ | $6.00_{\pm4.13}$ |
| Top-2 Direct Combination | $-4.14_{\pm0.23}$ | $-1.93_{\pm0.13}$ | $-1.34_{\pm0.01}$ | $-1.15_{\pm0.02}$ |
| Top-3 Direct Combination | $-4.21_{\pm0.07}$ | $-1.76_{\pm0.01}$ | $-1.32_{\pm0.01}$ | $-1.16_{\pm0.02}$ |
| Top-2 Optimised Combination | $-4.27_{\pm0.03}$ | $-1.71_{\pm0.05}$ | $-1.39_{\pm0.01}$ | $-1.12_{\pm0.02}$ |
| Top-3 Optimised Combination | $1.84_{\pm2.73}$ | $0.57_{\pm0.85}$ | $-1.34_{\pm0.03}$ | $-1.07_{\pm0.10}$ |

Table 26: Rotated CV regression: NLL (↓) comparison on in-distribution (ID) and out-of-distribution (OOD) test sets and with tuned hyperparameters.

| Noise Type | Energy | | Boston | | Wine | | Yacht | | Concrete | |
|---|---|---|---|---|---|---|---|---|---|---|
| | ID | OOD | ID | OOD | ID | OOD | ID | OOD | ID | OOD |
| No Noise | $0.04_{\pm0.00}$ | $0.05_{\pm0.00}$ | $0.13_{\pm0.02}$ | $0.15_{\pm0.02}$ | $0.25_{\pm0.04}$ | $29.42_{\pm25.71}$ | $0.07_{\pm0.07}$ | $0.16_{\pm0.05}$ | $0.10_{\pm0.01}$ | $0.10_{\pm0.01}$ |
| Input Gaussian | $0.04_{\pm0.00}$ | $0.05_{\pm0.00}$ | $0.11_{\pm0.02}$ | $0.13_{\pm0.02}$ | $0.25_{\pm0.04}$ | $29.39_{\pm25.66}$ | $0.02_{\pm0.02}$ | $0.09_{\pm0.07}$ | $0.10_{\pm0.01}$ | $0.10_{\pm0.01}$ |
| Input-Target CMixUp | $0.04_{\pm0.00}$ | $0.05_{\pm0.00}$ | $0.13_{\pm0.01}$ | $0.15_{\pm0.01}$ | $0.25_{\pm0.04}$ | $28.95_{\pm25.23}$ | $0.15_{\pm0.17}$ | $0.19_{\pm0.16}$ | $0.10_{\pm0.00}$ | $0.11_{\pm0.00}$ |
| Activation Gaussian | $0.04_{\pm0.00}$ | $0.05_{\pm0.00}$ | $0.13_{\pm0.03}$ | $0.15_{\pm0.02}$ | $0.25_{\pm0.04}$ | $29.57_{\pm25.82}$ | $0.03_{\pm0.02}$ | $0.09_{\pm0.07}$ | $0.10_{\pm0.01}$ | $0.10_{\pm0.01}$ |
| Activation Dropout | $0.04_{\pm0.00}$ | $0.05_{\pm0.00}$ | $0.16_{\pm0.05}$ | $0.18_{\pm0.04}$ | $0.38_{\pm0.06}$ | $17.02_{\pm13.89}$ | $0.03_{\pm0.01}$ | $0.09_{\pm0.07}$ | $0.10_{\pm0.01}$ | $0.11_{\pm0.01}$ |
| Gradient Gaussian | $0.04_{\pm0.00}$ | $0.04_{\pm0.00}$ | $0.24_{\pm0.13}$ | $0.29_{\pm0.11}$ | $0.25_{\pm0.04}$ | $28.34_{\pm24.70}$ | $0.01_{\pm0.01}$ | $0.09_{\pm0.08}$ | $0.10_{\pm0.01}$ | $0.10_{\pm0.01}$ |
| Model | $0.04_{\pm0.00}$ | $0.05_{\pm0.00}$ | $0.13_{\pm0.02}$ | $0.15_{\pm0.02}$ | $0.25_{\pm0.04}$ | $29.41_{\pm25.69}$ | $0.54_{\pm0.70}$ | $0.55_{\pm0.69}$ | $0.10_{\pm0.01}$ | $0.11_{\pm0.01}$ |
| Weight Gaussian | $0.04_{\pm0.00}$ | $0.05_{\pm0.00}$ | $0.12_{\pm0.04}$ | $0.15_{\pm0.03}$ | $0.25_{\pm0.04}$ | $29.51_{\pm25.77}$ | $0.02_{\pm0.01}$ | $0.06_{\pm0.05}$ | $0.10_{\pm0.01}$ | $0.10_{\pm0.01}$ |
| Weight DropConnect | $0.04_{\pm0.00}$ | $0.05_{\pm0.00}$ | $0.13_{\pm0.04}$ | $0.16_{\pm0.04}$ | $0.25_{\pm0.04}$ | $29.53_{\pm25.81}$ | $0.05_{\pm0.05}$ | $0.07_{\pm0.04}$ | $0.10_{\pm0.01}$ | $0.10_{\pm0.01}$ |
| Top-2 Direct | $0.04_{\pm0.00}$ | $0.05_{\pm0.00}$ | $0.15_{\pm0.02}$ | $0.18_{\pm0.02}$ | $0.25_{\pm0.04}$ | $29.43_{\pm25.69}$ | $0.03_{\pm0.03}$ | $0.10_{\pm0.07}$ | $0.10_{\pm0.01}$ | $0.10_{\pm0.01}$ |
| Top-3 Direct | $0.04_{\pm0.00}$ | $0.05_{\pm0.00}$ | $0.13_{\pm0.02}$ | $0.15_{\pm0.02}$ | $0.25_{\pm0.04}$ | $29.51_{\pm25.69}$ | $0.04_{\pm0.03}$ | $0.09_{\pm0.06}$ | $0.10_{\pm0.01}$ | $0.11_{\pm0.01}$ |
| Top-2 Optimised | $0.04_{\pm0.00}$ | $0.05_{\pm0.00}$ | $0.13_{\pm0.02}$ | $0.16_{\pm0.03}$ | $0.25_{\pm0.04}$ | $29.35_{\pm25.55}$ | $0.01_{\pm0.01}$ | $0.07_{\pm0.06}$ | $0.10_{\pm0.01}$ | $0.10_{\pm0.01}$ |
| Top-3 Optimised | $0.04_{\pm0.00}$ | $0.05_{\pm0.00}$ | $0.14_{\pm0.01}$ | $0.17_{\pm0.01}$ | $0.25_{\pm0.04}$ | $29.31_{\pm25.46}$ | $0.04_{\pm0.03}$ | $0.10_{\pm0.07}$ | $0.10_{\pm0.01}$ | $0.11_{\pm0.01}$ |

Table 27: Tabular regression: MSE (↓) comparison on in-distribution (ID) and out-of-distribution (OOD) test sets and with tuned hyperparameters.

| Noise Type | Energy | | Boston | | Wine | | Yacht | | Concrete | |
|---|---|---|---|---|---|---|---|---|---|---|
| | ID | OOD | ID | OOD | ID | OOD | ID | OOD | ID | OOD |
| No Noise | $-1.54_{\pm0.05}$ | $6.64_{\pm3.38}$ | $-0.19_{\pm0.12}$ | $0.04_{\pm0.12}$ | $-0.22_{\pm0.08}$ | $0.03_{\pm0.12}$ | $-1.18_{\pm0.22}$ | $10.45_{\pm16.46}$ | $-0.56_{\pm0.10}$ | $-0.01_{\pm0.24}$ |
| Input Gaussian | $-1.55_{\pm0.03}$ | $6.55_{\pm1.53}$ | $-0.54_{\pm0.14}$ | $-0.40_{\pm0.18}$ | $-0.22_{\pm0.08}$ | $0.03_{\pm0.12}$ | $-1.46_{\pm0.19}$ | $-1.04_{\pm0.30}$ | $-0.53_{\pm0.11}$ | $0.21_{\pm0.41}$ |
| Input-Target CMixUp | $-1.53_{\pm0.03}$ | $6.03_{\pm2.17}$ | $-0.42_{\pm0.12}$ | $-0.29_{\pm0.14}$ | $-0.19_{\pm0.09}$ | $81.98_{\pm58.22}$ | $-1.05_{\pm0.20}$ | $-0.87_{\pm0.31}$ | $-0.63_{\pm0.01}$ | $-0.26_{\pm0.26}$ |
| Activation Gaussian | $-1.56_{\pm0.03}$ | $6.64_{\pm0.32}$ | $-0.20_{\pm0.17}$ | $0.04_{\pm0.11}$ | $-0.22_{\pm0.08}$ | $0.04_{\pm0.10}$ | $-1.31_{\pm0.15}$ | $-1.21_{\pm0.14}$ | $-0.56_{\pm0.09}$ | $0.08_{\pm0.44}$ |
| Activation Dropout | $-1.53_{\pm0.05}$ | $4.44_{\pm2.01}$ | $-0.62_{\pm0.06}$ | $-0.59_{\pm0.06}$ | $0.01_{\pm0.05}$ | $0.03_{\pm0.04}$ | $-1.18_{\pm0.38}$ | $-1.07_{\pm0.39}$ | $-0.58_{\pm0.08}$ | $0.07_{\pm0.30}$ |
| Gradient Gaussian | $-1.56_{\pm0.07}$ | $8.61_{\pm3.39}$ | $0.55_{\pm0.79}$ | $0.80_{\pm0.98}$ | $-0.22_{\pm0.08}$ | $0.04_{\pm0.09}$ | $-2.04_{\pm0.30}$ | $-1.49_{\pm0.32}$ | $-0.55_{\pm0.09}$ | $0.13_{\pm0.43}$ |
| Model | $-1.55_{\pm0.04}$ | $7.20_{\pm3.32}$ | $-0.16_{\pm0.18}$ | $0.06_{\pm0.19}$ | $-0.22_{\pm0.08}$ | $0.03_{\pm0.11}$ | $-0.77_{\pm1.13}$ | $0.15_{\pm0.80}$ | $-0.59_{\pm0.06}$ | $-0.05_{\pm0.45}$ |
| Weight Gaussian | $-1.56_{\pm0.04}$ | $6.82_{\pm2.72}$ | $-0.36_{\pm0.18}$ | $-0.11_{\pm0.21}$ | $-0.22_{\pm0.08}$ | $0.04_{\pm0.11}$ | $-2.08_{\pm0.51}$ | $-1.71_{\pm0.23}$ | $-0.54_{\pm0.08}$ | $0.14_{\pm0.39}$ |
| Weight DropConnect | $-1.55_{\pm0.03}$ | $5.53_{\pm1.27}$ | $-0.62_{\pm0.05}$ | $-0.49_{\pm0.06}$ | $-0.22_{\pm0.08}$ | $0.02_{\pm0.12}$ | $-1.36_{\pm0.43}$ | $-1.25_{\pm0.33}$ | $-0.57_{\pm0.08}$ | $0.07_{\pm0.35}$ |
| Top-2 Direct | $-1.55_{\pm0.04}$ | $6.09_{\pm2.35}$ | $0.32_{\pm0.19}$ | $0.63_{\pm0.30}$ | $-0.22_{\pm0.08}$ | $0.02_{\pm0.11}$ | $-1.37_{\pm0.34}$ | $0.35_{\pm2.06}$ | $-0.57_{\pm0.10}$ | $0.11_{\pm0.32}$ |
| Top-3 Direct | $-1.53_{\pm0.03}$ | $4.53_{\pm2.51}$ | $-0.18_{\pm0.22}$ | $0.06_{\pm0.24}$ | $-0.22_{\pm0.08}$ | $0.03_{\pm0.12}$ | $-1.82_{\pm0.67}$ | $-1.39_{\pm0.22}$ | $-0.59_{\pm0.09}$ | $0.04_{\pm0.40}$ |
| Top-2 Optimised | $-1.56_{\pm0.03}$ | $8.60_{\pm3.54}$ | $0.16_{\pm0.18}$ | $0.46_{\pm0.29}$ | $-0.22_{\pm0.08}$ | $0.03_{\pm0.11}$ | $-2.03_{\pm0.14}$ | $-1.55_{\pm0.21}$ | $-0.57_{\pm0.09}$ | $0.22_{\pm0.55}$ |
| Top-3 Optimised | $-1.55_{\pm0.02}$ | $6.68_{\pm1.97}$ | $-0.27_{\pm0.46}$ | $-0.04_{\pm0.53}$ | $-0.22_{\pm0.08}$ | $0.03_{\pm0.12}$ | $-1.77_{\pm0.38}$ | $-0.40_{\pm1.62}$ | $-0.60_{\pm0.07}$ | $0.17_{\pm0.61}$ |

Table 28: Tabular regression: NLL (↓) comparison on in-distribution (ID) and out-of-distribution (OOD) test sets and with tuned hyperparameters.

| Noise Type | Energy | | Wine | | Concrete | |
|---|---|---|---|---|---|---|
| | ID | OOD | ID | OOD | ID | OOD |
| No Noise | $0.03_{\pm0.01}$ | $0.03_{\pm0.01}$ | $0.21_{\pm0.01}$ | $37.20_{\pm33.29}$ | $0.09_{\pm0.01}$ | $0.10_{\pm0.01}$ |
| Input Gaussian | $0.03_{\pm0.01}$ | $0.03_{\pm0.01}$ | $0.21_{\pm0.02}$ | $45.36_{\pm43.45}$ | $0.08_{\pm0.01}$ | $0.09_{\pm0.01}$ |
| Input-Target CMixUp | $0.03_{\pm0.00}$ | $0.04_{\pm0.00}$ | $0.18_{\pm0.01}$ | $49.23_{\pm37.16}$ | $0.10_{\pm0.01}$ | $0.11_{\pm0.00}$ |
| Activation Gaussian | $0.03_{\pm0.01}$ | $0.03_{\pm0.01}$ | $0.22_{\pm0.01}$ | $43.40_{\pm45.61}$ | $0.09_{\pm0.02}$ | $0.10_{\pm0.01}$ |
| Activation Dropout | $0.03_{\pm0.01}$ | $0.03_{\pm0.00}$ | $0.24_{\pm0.02}$ | $42.16_{\pm33.28}$ | $0.09_{\pm0.02}$ | $0.10_{\pm0.01}$ |
| Gradient Gaussian | $0.03_{\pm0.01}$ | $0.04_{\pm0.01}$ | $311.38_{\pm405.17}$ | $1858.44_{\pm2585.50}$ | $0.09_{\pm0.02}$ | $0.10_{\pm0.02}$ |
| Model | $0.03_{\pm0.00}$ | $0.04_{\pm0.00}$ | $0.21_{\pm0.01}$ | $37.68_{\pm32.86}$ | $0.11_{\pm0.01}$ | $0.12_{\pm0.01}$ |
| Weight Gaussian | $0.03_{\pm0.01}$ | $0.03_{\pm0.01}$ | $0.20_{\pm0.02}$ | $41.60_{\pm40.06}$ | $0.09_{\pm0.02}$ | $0.10_{\pm0.01}$ |
| Weight DropConnect | $0.03_{\pm0.01}$ | $0.03_{\pm0.01}$ | $0.20_{\pm0.02}$ | $46.44_{\pm50.40}$ | $0.09_{\pm0.02}$ | $0.10_{\pm0.01}$ |

Table 29: Tabular regression: MSE (↓) comparison on in-distribution (ID) and out-of-distribution (OOD) test sets and with hyperparameters transferred across datasets.

| Noise Type | Energy | | Wine | | Concrete | |
|---|---|---|---|---|---|---|
| | ID | OOD | ID | OOD | ID | OOD |
| NO NOISE | $-1.70_{\pm0.11}$ | $1.24_{\pm0.61}$ | $4.78_{\pm1.31}$ | $6.66_{\pm1.82}$ | $-0.54_{\pm0.15}$ | $0.32_{\pm0.69}$ |
| INPUT GAUSSIAN | $-1.74_{\pm0.07}$ | $1.06_{\pm0.88}$ | $5.24_{\pm1.58}$ | $19.29_{\pm19.08}$ | $-0.40_{\pm0.15}$ | $0.23_{\pm0.47}$ |
| INPUT-TARGET CMIXUP | $-1.66_{\pm0.10}$ | $0.42_{\pm1.30}$ | $-0.06_{\pm0.15}$ | $95014.61_{\pm134370.19}$ | $-0.65_{\pm0.05}$ | $0.53_{\pm1.27}$ |
| ACTIVATION GAUSSIAN | $-1.71_{\pm0.09}$ | $0.51_{\pm1.20}$ | $1.84_{\pm1.25}$ | $177076.49_{\pm250421.18}$ | $-0.50_{\pm0.14}$ | $0.46_{\pm0.74}$ |
| ACTIVATION DROPOUT | $-1.66_{\pm0.12}$ | $0.02_{\pm1.35}$ | $-0.27_{\pm0.05}$ | $-0.19_{\pm0.05}$ | $-0.56_{\pm0.11}$ | $1.19_{\pm2.07}$ |
| GRADIENT GAUSSIAN | $-1.70_{\pm0.11}$ | $5.71_{\pm6.29}$ | $1.97_{\pm1.94}$ | $2.06_{\pm2.03}$ | $-0.11_{\pm0.10}$ | $0.63_{\pm0.34}$ |
| MODEL | $-1.64_{\pm0.02}$ | $-0.16_{\pm0.64}$ | $4.68_{\pm1.33}$ | $6.22_{\pm1.31}$ | $-0.64_{\pm0.03}$ | $-0.49_{\pm0.17}$ |
| WEIGHT GAUSSIAN | $-1.71_{\pm0.06}$ | $-0.26_{\pm0.33}$ | $4.51_{\pm1.62}$ | $206785.40_{\pm200338.85}$ | $-0.55_{\pm0.07}$ | $-0.03_{\pm0.47}$ |
| WEIGHT DROPCONNECT | $-1.73_{\pm0.12}$ | $0.00_{\pm0.39}$ | $0.37_{\pm0.43}$ | $37100.06_{\pm52466.28}$ | $-0.67_{\pm0.13}$ | $-0.56_{\pm0.11}$ |

Table 30: Tabular regression: NLL ($\downarrow$) comparison on in-distribution (ID) and out-of-distribution (OOD) test sets and with hyperparameters transferred across datasets.

| Noise Type | Boston | | Yacht | | Concrete | |
|---|---|---|---|---|---|---|
| | ID | OOD | ID | OOD | ID | OOD |
| NO NOISE | $0.13_{\pm0.04}$ | $0.16_{\pm0.04}$ | $0.56_{\pm0.74}$ | $0.59_{\pm0.72}$ | $0.11_{\pm0.01}$ | $0.11_{\pm0.00}$ |
| INPUT GAUSSIAN | $0.13_{\pm0.03}$ | $0.15_{\pm0.03}$ | $0.55_{\pm0.75}$ | $0.59_{\pm0.72}$ | $0.10_{\pm0.01}$ | $0.11_{\pm0.01}$ |
| INPUT-TARGET CMIXUP | $0.50_{\pm0.47}$ | $0.52_{\pm0.46}$ | $0.05_{\pm0.02}$ | $0.11_{\pm0.06}$ | $0.11_{\pm0.01}$ | $0.12_{\pm0.01}$ |
| ACTIVATION GAUSSIAN | $0.11_{\pm0.03}$ | $0.13_{\pm0.03}$ | $0.56_{\pm0.75}$ | $0.62_{\pm0.71}$ | $0.11_{\pm0.01}$ | $0.11_{\pm0.01}$ |
| ACTIVATION DROPOUT | $0.17_{\pm0.07}$ | $0.18_{\pm0.06}$ | $0.05_{\pm0.02}$ | $0.08_{\pm0.01}$ | $0.11_{\pm0.01}$ | $0.11_{\pm0.01}$ |
| GRADIENT GAUSSIAN | $0.16_{\pm0.04}$ | $0.18_{\pm0.04}$ | $0.02_{\pm0.01}$ | $0.06_{\pm0.04}$ | $0.11_{\pm0.01}$ | $0.11_{\pm0.00}$ |
| MODEL | $0.15_{\pm0.06}$ | $0.17_{\pm0.06}$ | $0.11_{\pm0.04}$ | $0.13_{\pm0.04}$ | $0.12_{\pm0.01}$ | $0.12_{\pm0.01}$ |
| WEIGHT GAUSSIAN | $0.12_{\pm0.04}$ | $0.14_{\pm0.03}$ | $0.03_{\pm0.00}$ | $0.07_{\pm0.02}$ | $0.11_{\pm0.01}$ | $0.11_{\pm0.00}$ |
| WEIGHT DROPCONNECT | $0.13_{\pm0.03}$ | $0.15_{\pm0.03}$ | $0.04_{\pm0.02}$ | $0.10_{\pm0.07}$ | $0.11_{\pm0.01}$ | $0.11_{\pm0.01}$ |

Table 31: Tabular regression: MSE ($\downarrow$) comparison on in-distribution (ID) and out-of-distribution (OOD) test sets and with hyperparameters transferred across architectures.

| Noise Type | Boston | | Yacht | | Concrete | |
|---|---|---|---|---|---|---|
| | ID | OOD | ID | OOD | ID | OOD |
| NO NOISE | $-0.25_{\pm0.21}$ | $-0.01_{\pm0.17}$ | $-0.42_{\pm0.88}$ | $-0.02_{\pm0.80}$ | $-0.57_{\pm0.06}$ | $-0.02_{\pm0.54}$ |
| INPUT GAUSSIAN | $-0.35_{\pm0.10}$ | $-0.17_{\pm0.09}$ | $-0.94_{\pm1.25}$ | $-0.72_{\pm1.09}$ | $-0.58_{\pm0.08}$ | $0.06_{\pm0.63}$ |
| INPUT-TARGET CMIXUP | $-0.09_{\pm0.29}$ | $-0.02_{\pm0.23}$ | $-1.20_{\pm0.07}$ | $-1.06_{\pm0.11}$ | $-0.60_{\pm0.04}$ | $-0.25_{\pm0.32}$ |
| ACTIVATION GAUSSIAN | $-0.18_{\pm0.66}$ | $0.02_{\pm0.72}$ | $-0.44_{\pm0.93}$ | $-0.09_{\pm0.65}$ | $-0.56_{\pm0.07}$ | $-0.09_{\pm0.36}$ |
| ACTIVATION DROPOUT | $-0.56_{\pm0.04}$ | $-0.54_{\pm0.04}$ | $-1.11_{\pm0.31}$ | $-0.81_{\pm0.11}$ | $-0.58_{\pm0.07}$ | $0.08_{\pm0.66}$ |
| GRADIENT GAUSSIAN | $0.97_{\pm1.13}$ | $1.22_{\pm1.16}$ | $-1.52_{\pm0.43}$ | $-1.25_{\pm0.17}$ | $-0.57_{\pm0.06}$ | $-0.02_{\pm0.54}$ |
| MODEL | $-0.31_{\pm0.25}$ | $-0.10_{\pm0.19}$ | $-0.57_{\pm0.10}$ | $-0.56_{\pm0.11}$ | $-0.58_{\pm0.03}$ | $-0.33_{\pm0.15}$ |
| WEIGHT GAUSSIAN | $-0.42_{\pm0.18}$ | $-0.28_{\pm0.18}$ | $-1.62_{\pm0.34}$ | $-0.77_{\pm1.00}$ | $-0.56_{\pm0.07}$ | $0.11_{\pm0.69}$ |
| WEIGHT DROPCONNECT | $-0.46_{\pm0.09}$ | $-0.34_{\pm0.10}$ | $-1.72_{\pm0.31}$ | $-1.20_{\pm0.26}$ | $-0.55_{\pm0.06}$ | $0.07_{\pm0.60}$ |

Table 32: Tabular regression: NLL ($\downarrow$) comparison on in-distribution (ID) and out-of-distribution (OOD) test sets and with hyperparameters transferred across architectures.

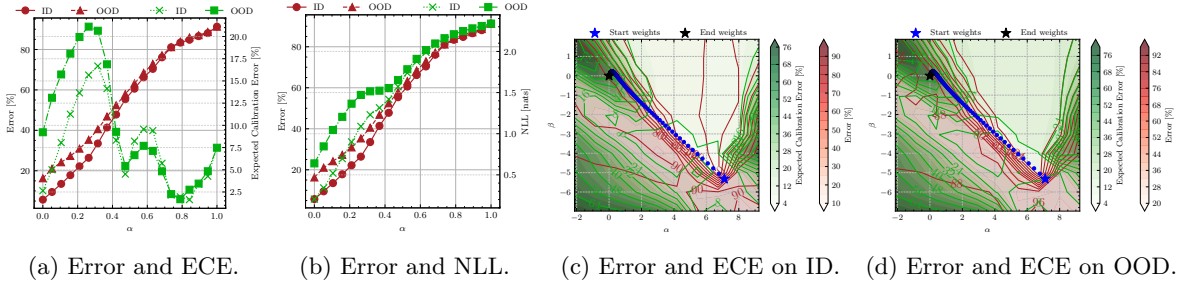

(a) Error and ECE.  (b) Error and NLL.  (c) Error and ECE on ID.  (d) Error and ECE on OOD.

Figure 9: Input Random Crop, Horizontal Flip on CIFAR-10. *Observations*: Did not change the smoothness of the 1D curves or the 2D metric landscape trajectory compared to no noise.

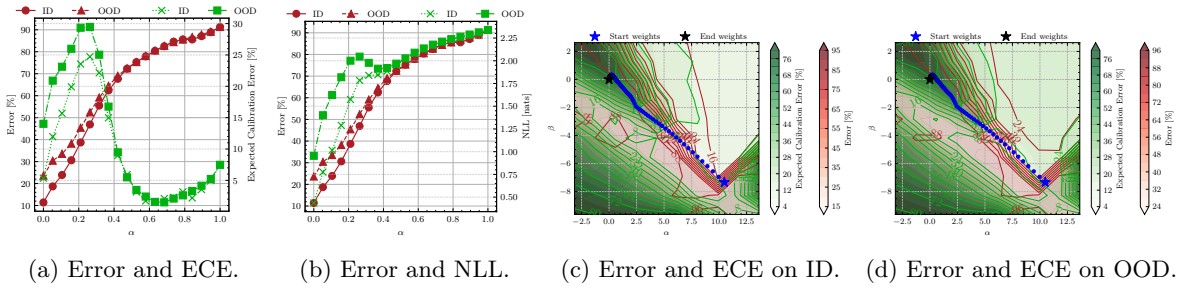

(a) Error and ECE.  (b) Error and NLL.  (c) Error and ECE on ID.  (d) Error and ECE on OOD.

Figure 10: Input Additive Gaussian on CIFAR-10. *Observations*: Changed the smoothness of the 1D curves where NLL became less smooth and removed the bumps in ECE for $\alpha$ approaching the initial model. The 2D metric landscape trajectory did not change in comparison to no noise.

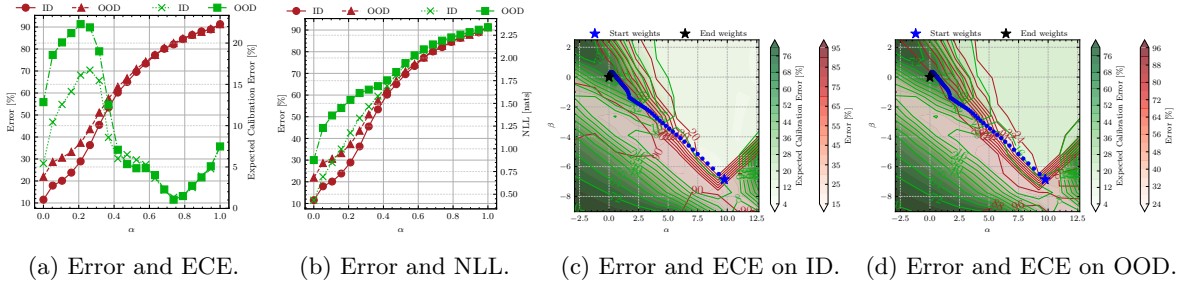

(a) Error and ECE.  (b) Error and NLL.  (c) Error and ECE on ID.  (d) Error and ECE on OOD.

Figure 11: Input ODS on CIFAR-10. *Observations*: Marginally changed the smoothness of the 1D curves. The 2D metric landscape trajectory did not change in comparison to no noise.

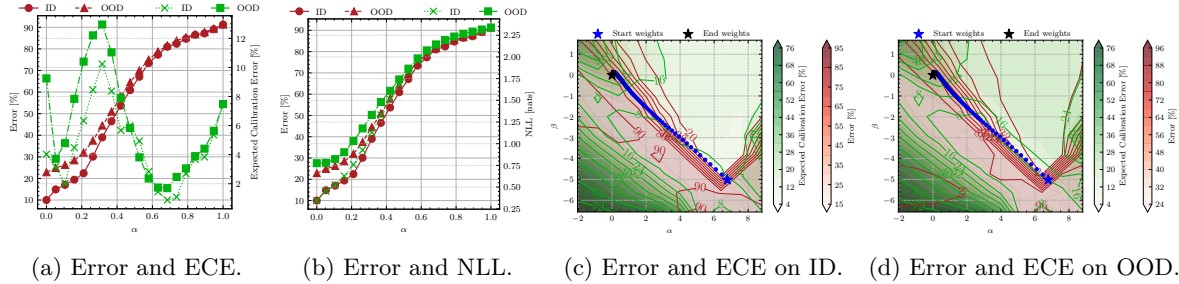

(a) Error and ECE.  (b) Error and NLL.  (c) Error and ECE on ID.  (d) Error and ECE on OOD.

Figure 12: Input-Target MixUp on CIFAR-10. *Observations*: Both the NLL and ECE 1D curves changed in comparison to no noise, and the 2D plots seem to explore wider valleys compared to no noise.

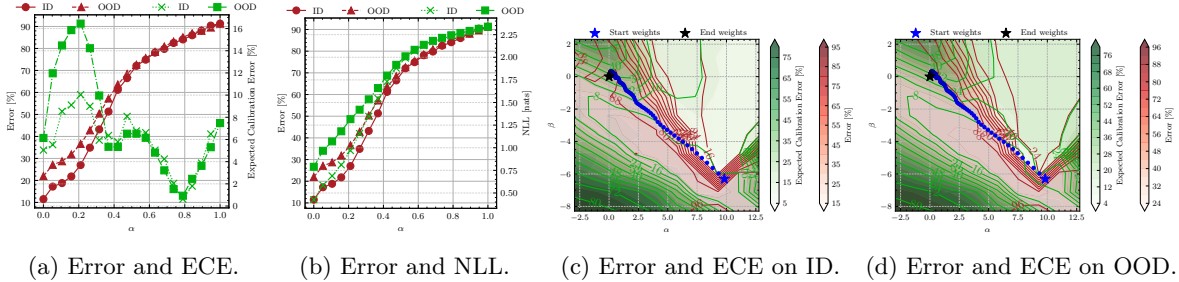

(a) Error and ECE.     (b) Error and NLL.     (c) Error and ECE on ID.     (d) Error and ECE on OOD.

Figure 13: Target Smoothing on CIFAR-10. *Observations*: The NLL became more aligned with the error, not the ECE. The 2D plots show slightly more variation in the trajectory than no noise.

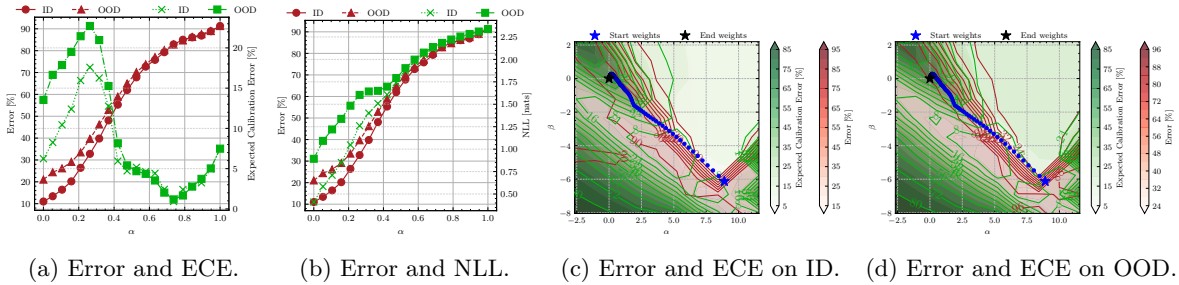

(a) Error and ECE.     (b) Error and NLL.     (c) Error and ECE on ID.     (d) Error and ECE on OOD.

Figure 14: Activation Additive Gaussian on CIFAR-10. *Observations*: Did not change the smoothness of the 1D curves or the 2D metric landscape trajectory compared to no noise.

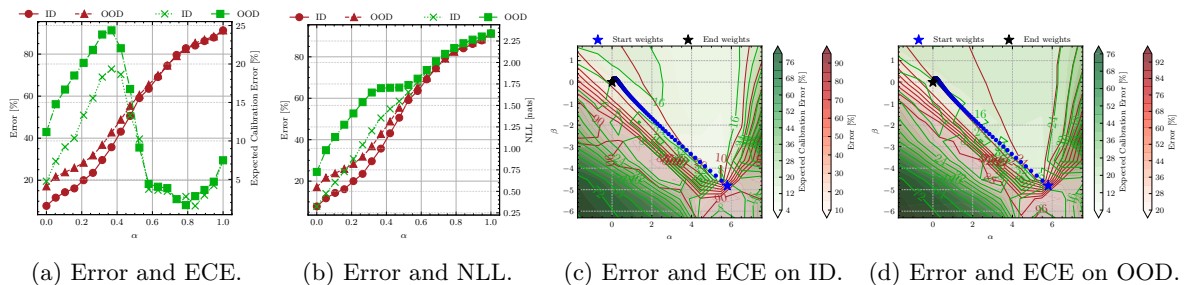

(a) Error and ECE.     (b) Error and NLL.     (c) Error and ECE on ID.     (d) Error and ECE on OOD.

Figure 15: Activation Dropout on CIFAR-10. *Observations*: Dropout narrowed the gap between ID and OOD results; nevertheless, the shape of the 1D curves is similar to no noise. The trajectories in 2D plots did not seem to converge into a narrow local minimum.

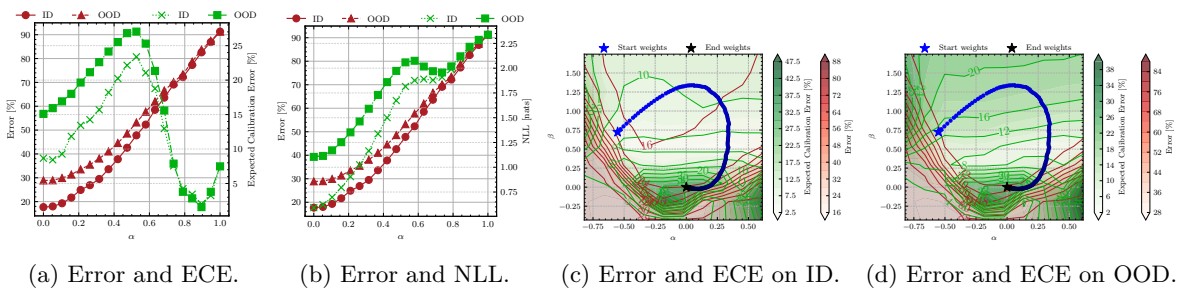

(a) Error and ECE.     (b) Error and NLL.     (c) Error and ECE on ID.     (d) Error and ECE on OOD.

Figure 16: Gradient Gaussian on CIFAR-10. *Observations*: The 1D and 2D figures changed curvature and shape drastically, and NLL and ECE follow a non-linear pattern. The 2D plots show a circular curvature, perhaps suggesting difficulty in convergence.

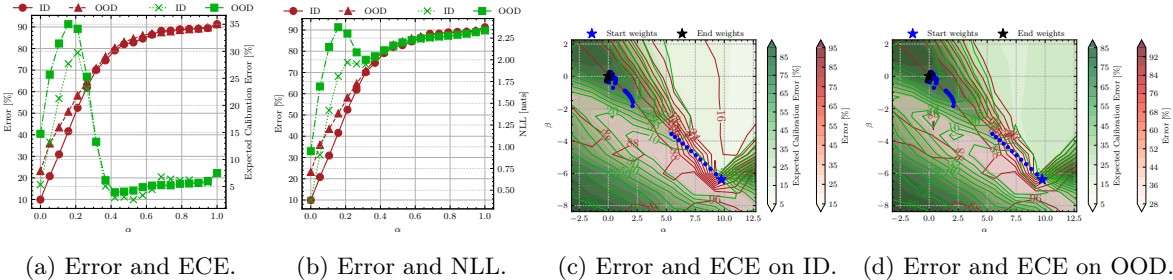

(a) Error and ECE.     (b) Error and NLL.     (c) Error and ECE on ID.     (d) Error and ECE on OOD.

Figure 17: Model Shrink and Perturb on CIFAR-10. *Observations*: The 1D and 2D figures changed curvature and shape drastically, and all metrics show a non-linear optimisation path as hypothesised. The point cluster around centres created by shrinking and perturbing the weights.

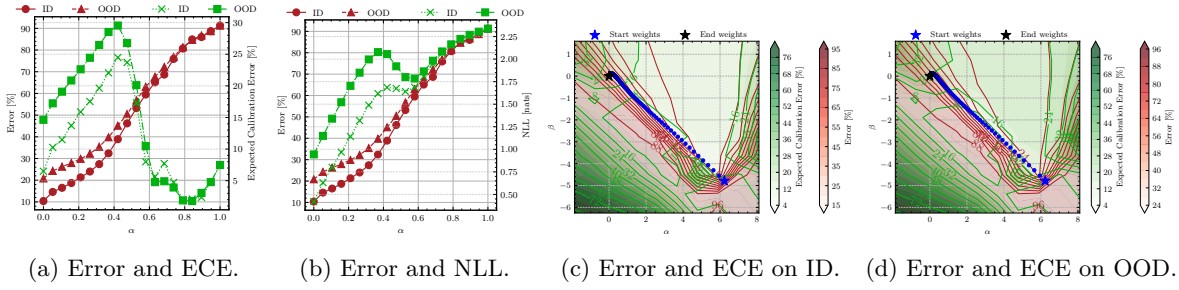

(a) Error and ECE.     (b) Error and NLL.     (c) Error and ECE on ID.     (d) Error and ECE on OOD.

Figure 18: Weight Additive Gaussian on CIFAR-10. *Observations*: The 1D curves marginally changed their shape. However, the difference between ID and OOD metrics became more profound. The 2D plots suggest that the optimisation was not able to converge.

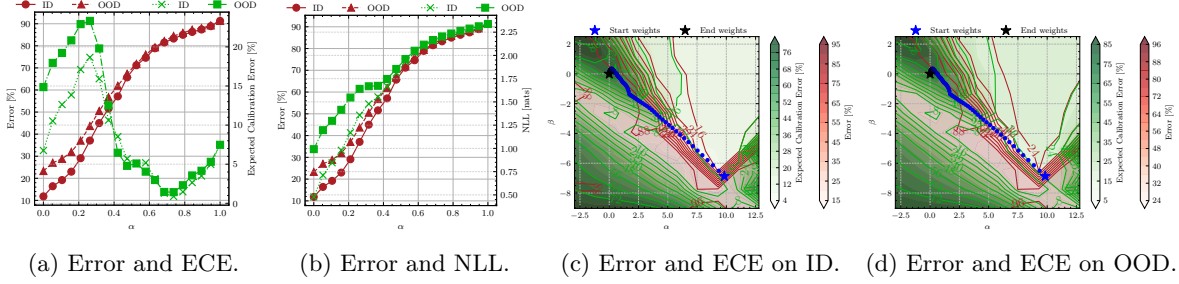

(a) Error and ECE.     (b) Error and NLL.     (c) Error and ECE on ID.     (d) Error and ECE on OOD.

Figure 19: Weight DropConnect on CIFAR-10. *Observations*: Did not change the smoothness of the 1D curves or the 2D metric landscape trajectory compared to no noise.

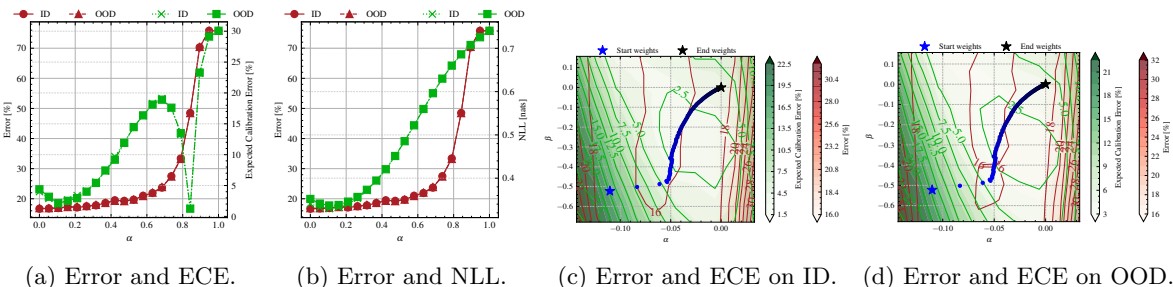

(a) Error and ECE.     (b) Error and NLL.     (c) Error and ECE on ID.     (d) Error and ECE on OOD.

Figure 20: No noise on Adult.

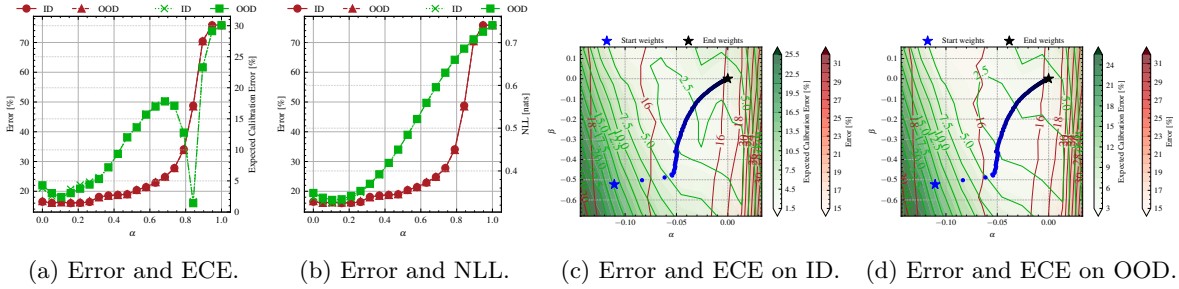

(a) Error and ECE.     (b) Error and NLL.     (c) Error and ECE on ID.     (d) Error and ECE on OOD.

Figure 21: Input Additive Gaussian on Adult. *Observations*: Did not change the smoothness of the 1D curves or the 2D metric landscape trajectory compared to no noise.

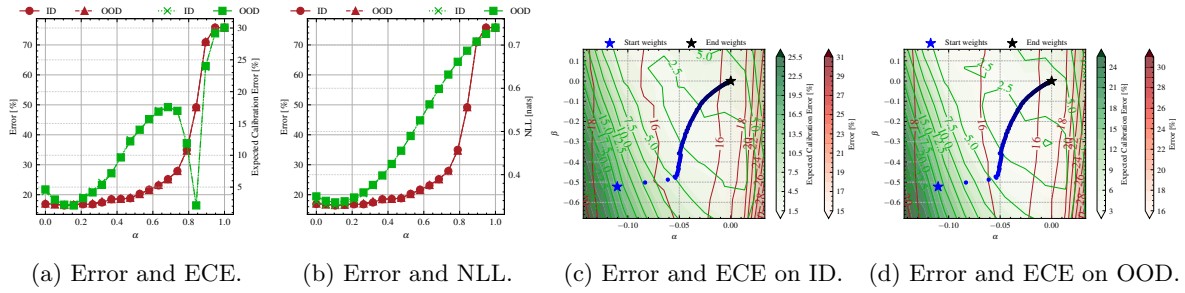

(a) Error and ECE.     (b) Error and NLL.     (c) Error and ECE on ID.     (d) Error and ECE on OOD.

Figure 22: Input ODS on Adult. *Observations*: Did not change the smoothness of the 1D curves or the 2D metric landscape trajectory compared to no noise.

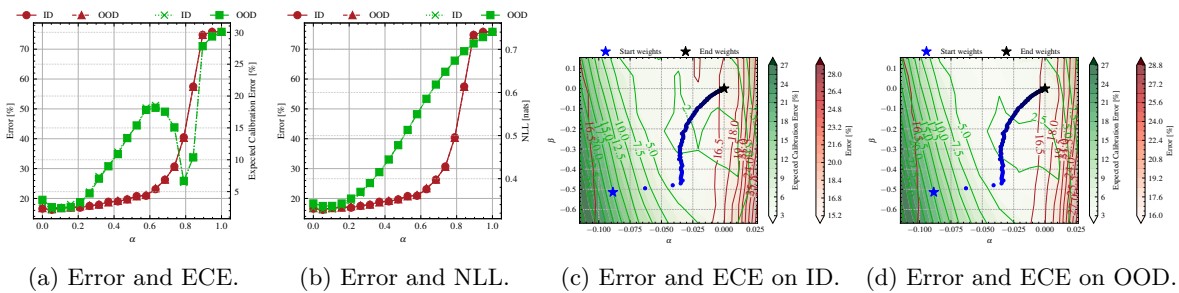

(a) Error and ECE.     (b) Error and NLL.     (c) Error and ECE on ID.     (d) Error and ECE on OOD.

Figure 23: Input-Target MixUp on Adult. *Observations*: Did not change the smoothness of the 1D curves or the 2D metric landscape trajectory compared to no noise.

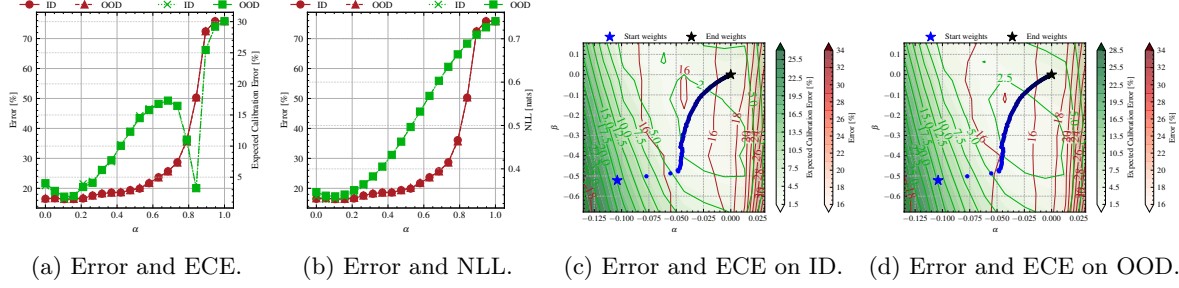

(a) Error and ECE.     (b) Error and NLL.     (c) Error and ECE on ID.     (d) Error and ECE on OOD.

Figure 24: Target Smoothing on Adult. *Observations*: Did not change the smoothness of the 1D curves or the 2D metric landscape trajectory compared to no noise.

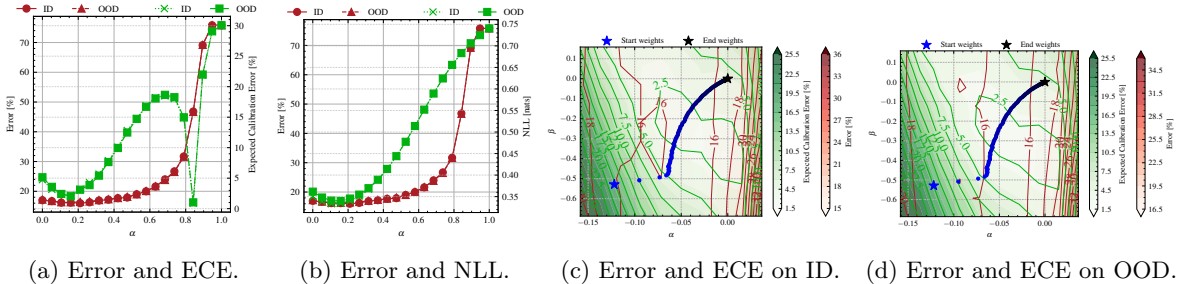

(a) Error and ECE.  (b) Error and NLL.  (c) Error and ECE on ID.  (d) Error and ECE on OOD.

Figure 25: Activation Additive Gaussian on Adult. *Observations*: Did not change the smoothness of the 1D curves or the 2D metric landscape trajectory compared to no noise.

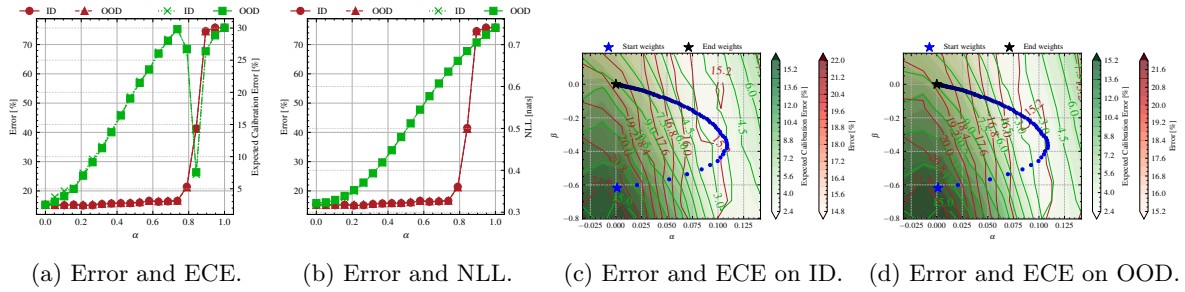

(a) Error and ECE.  (b) Error and NLL.  (c) Error and ECE on ID.  (d) Error and ECE on OOD.

Figure 26: Activation Dropout on Adult. *Observations*: Changed the ECE curvature and made the NLL plots smoother in the 1D case. In the 2D plots, the ECE and error appear aligned during optimisation. The curvature of the 2D plots has changed and there is a higher alignment between the ECE and error.

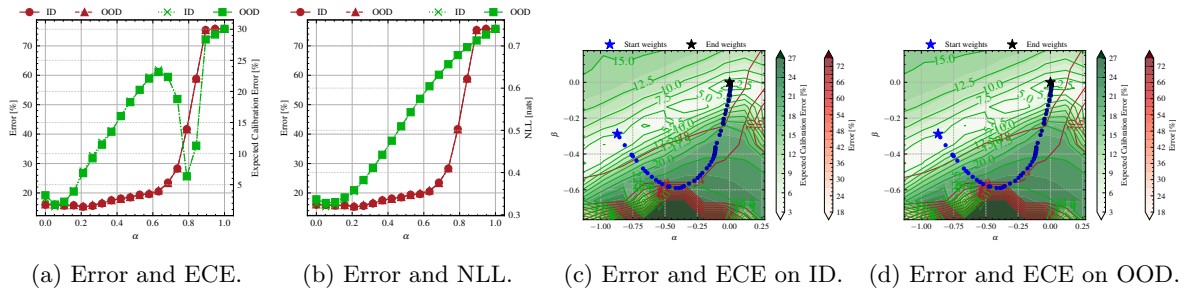

(a) Error and ECE.  (b) Error and NLL.  (c) Error and ECE on ID.  (d) Error and ECE on OOD.

Figure 27: Gradient Gaussian on Adult. *Observations*: Did not change the smoothness of the 1D curves, but the 2D trajectory appears more exploratory compared to no noise.

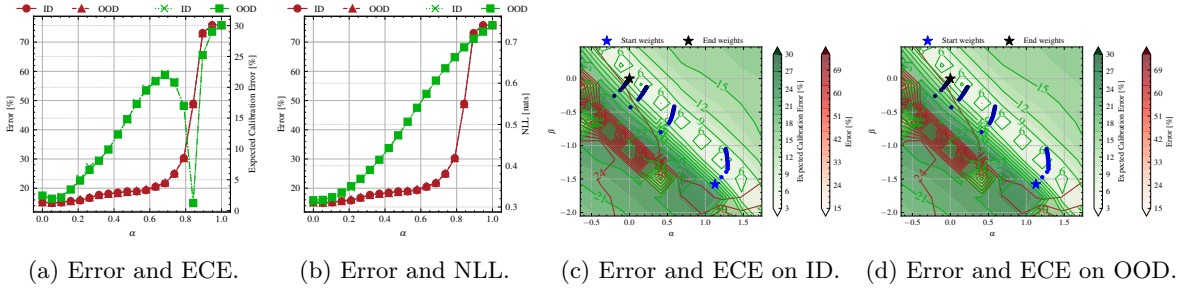

(a) Error and ECE.  (b) Error and NLL.  (c) Error and ECE on ID.  (d) Error and ECE on OOD.

Figure 28: Model Shrink and Perturb on Adult. *Observations*: Did not change the smoothness of the 1D curves, but the 2D trajectory appears more exploratory compared to no noise.

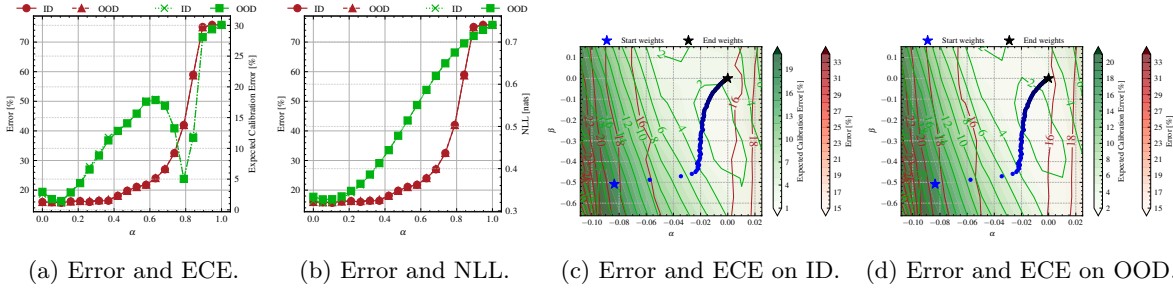

(a) Error and ECE.  (b) Error and NLL.  (c) Error and ECE on ID.  (d) Error and ECE on OOD.

Figure 29: Weight Additive Gaussian on Adult. *Observations*: Did not change the smoothness of the 1D curves or the 2D metric landscape trajectory compared to no noise.

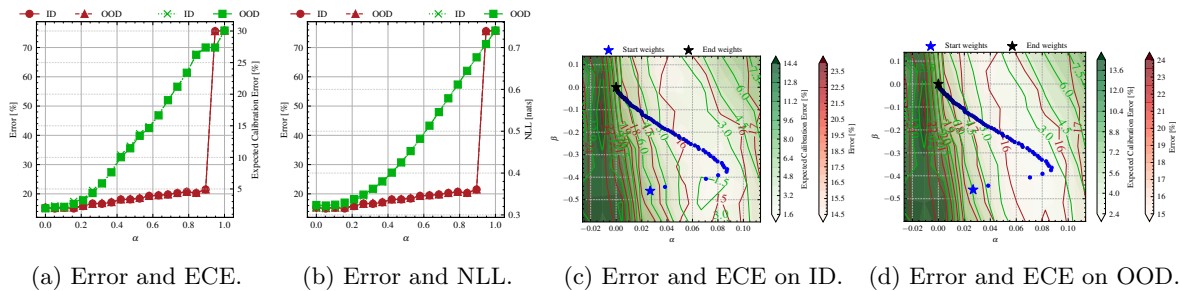

(a) Error and ECE.  (b) Error and NLL.  (c) Error and ECE on ID.  (d) Error and ECE on OOD.

Figure 30: Weight DropConnect on Adult. *Observations*: Changed the ECE curvature and made the NLL and ECE plots smoother in the 1D case. In the 2D plots, the ECE and error appear aligned during optimisation.

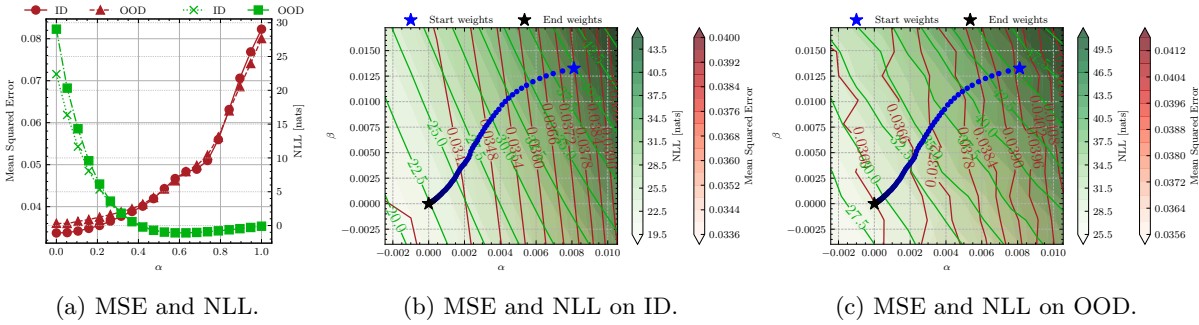

(a) MSE and NLL.  (b) MSE and NLL on ID.  (c) MSE and NLL on OOD.

Figure 31: Input Additive Gaussian on WikiFace. *Observations*: Did not change the smoothness of the 1D curves, or the 2D trajectory.

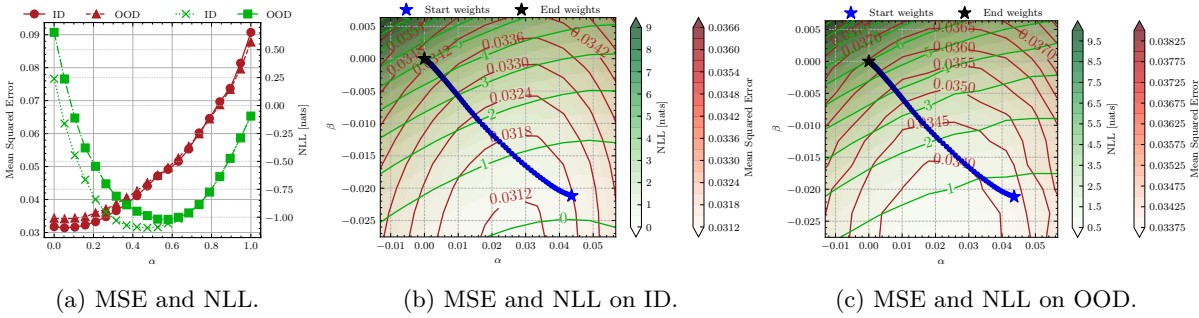

(a) MSE and NLL.  (b) MSE and NLL on ID.  (c) MSE and NLL on OOD.

Figure 32: Input Random Crop, Horizontal Flip on WikiFace. *Observations*: Surprisingly, the NLL starts decreasing compared to MSE as the model is interpolated between the final and the initial model in the 1D plots. The 2D plots demonstrate that the model was able to explore a deeper optimal from the start where NLL was slower to converge than MSE.

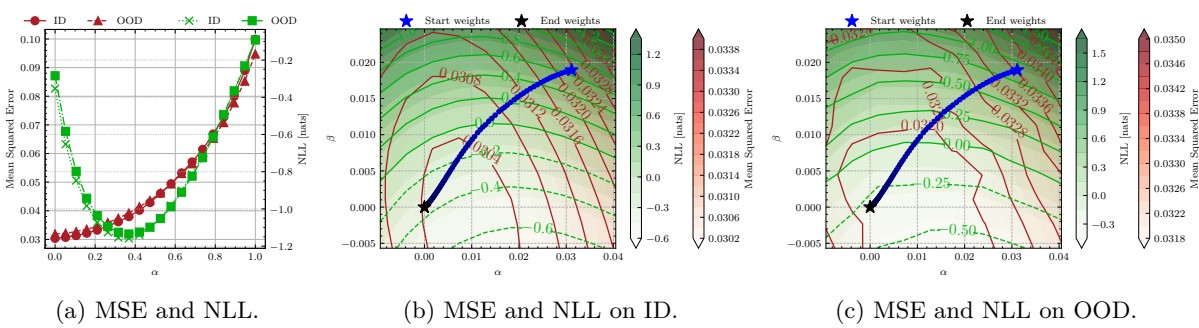

(a) MSE and NLL.  (b) MSE and NLL on ID.  (c) MSE and NLL on OOD.

Figure 33: Input AugMix on WikiFace. *Observations*: Surprisingly, the NLL starts decreasing compared to MSE as the model is interpolated between the final and the initial model in the 1D plots. The 2D plots demonstrate that the model was able to explore a deeper optimal from the start where NLL was slower to converge than MSE, and it did not converge in the optima from the perspective of NLL.

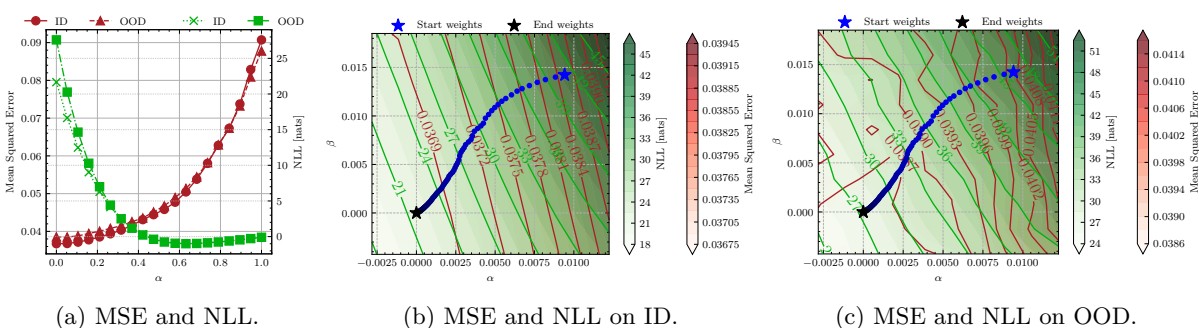

(a) MSE and NLL.  (b) MSE and NLL on ID.  (c) MSE and NLL on OOD.

Figure 34: Input-Target CMixUp on WikiFace. *Observations*: Did not change the smoothness of the 1D curves, or the 2D trajectory appears more exploratory compared to no noise.

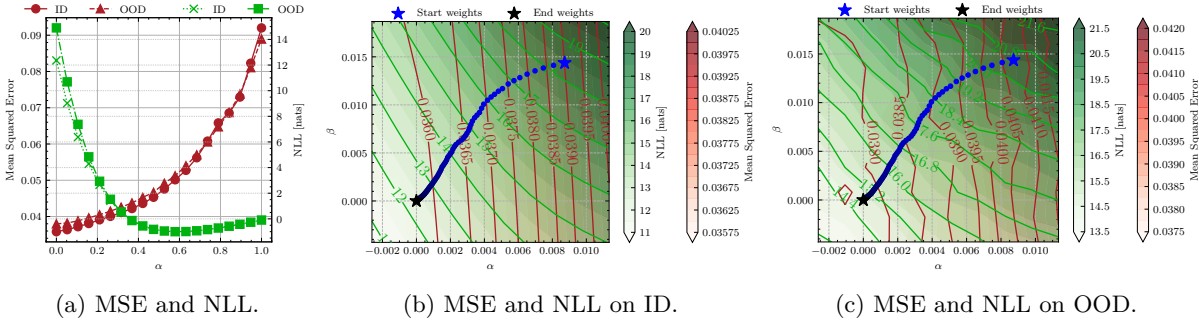

(a) MSE and NLL.  (b) MSE and NLL on ID.  (c) MSE and NLL on OOD.

Figure 35: Activation Additive Gaussian on WikiFace. *Observations*: Did not change the smoothness of the 1D curves, but the 2D trajectory appears more exploratory compared to no noise.

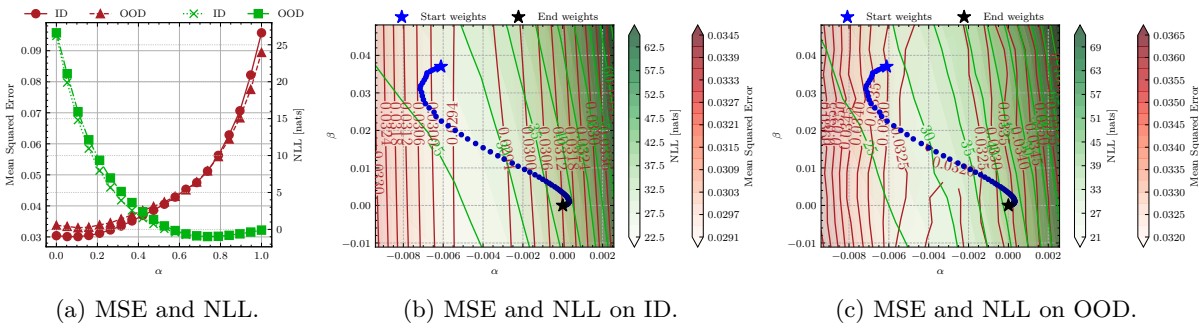

(a) MSE and NLL.  (b) MSE and NLL on ID.  (c) MSE and NLL on OOD.

Figure 36: Gradient Gaussian on WikiFace. *Observations*: Did not change the smoothness of the 1D curves, but the 2D trajectory appears to align MSE and NLL. However, it seems that the optimisation missed a local minimum during training.

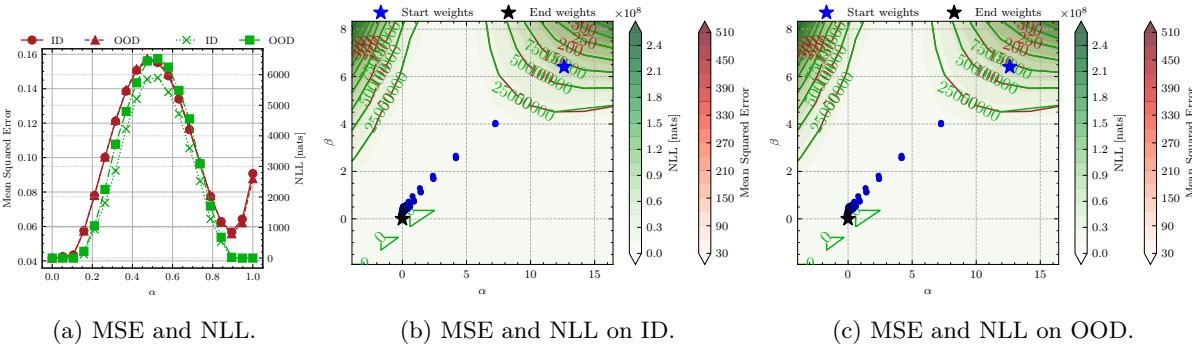

(a) MSE and NLL.  (b) MSE and NLL on ID.  (c) MSE and NLL on OOD.

Figure 37: Model Shrink and Perturb on WikiFace. *Observations*: Due to shrinking and perturbation, the experiment appears to converge in a narrow basin and as seed in the 1D plots, the optimisation was completely non-linear and unrecoverable.

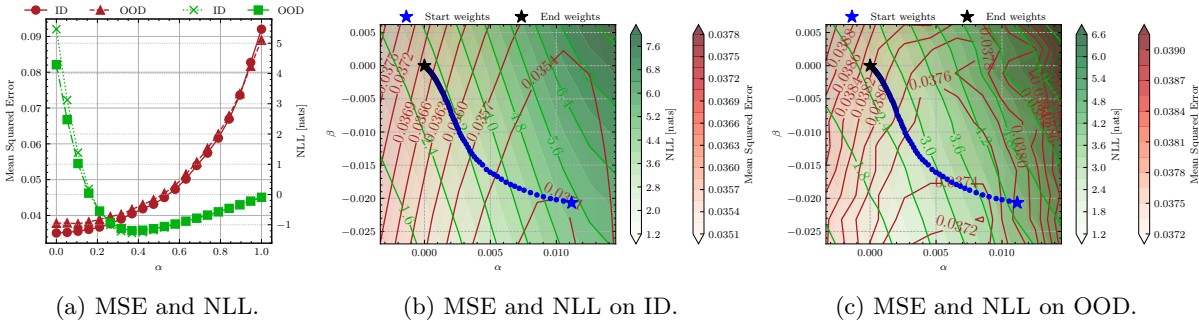

(a) MSE and NLL.     (b) MSE and NLL on ID.     (c) MSE and NLL on OOD.

Figure 38: Weight Additive Gaussian on WikiFace. *Observations*: The 1D curves look similar to no noise, although with respect to a different scale for NLL. The 2D plots explore a similar trajectory to no noise; however, the 2D landscape appears more distorted.

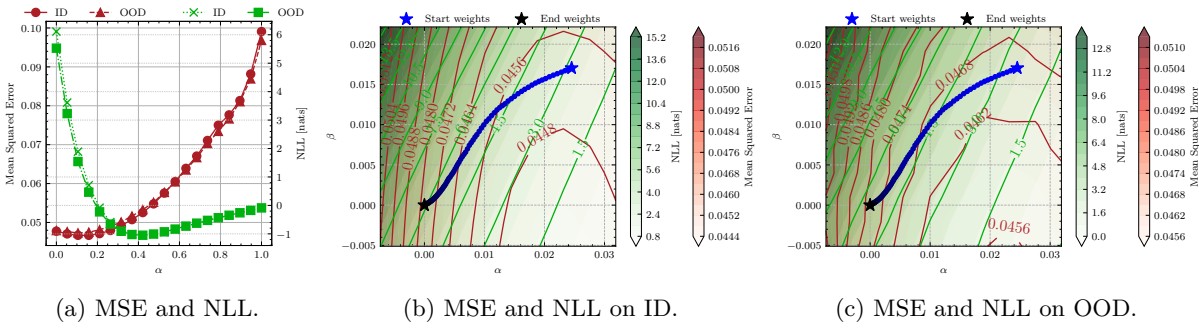

(a) MSE and NLL.     (b) MSE and NLL on ID.     (c) MSE and NLL on OOD.

Figure 39: Weight DropConnect on WikiFace. *Observations*: The 1D curves look similar to no noise, although with respect to a different scale for NLL. The 2D plots explore a similar trajectory to no noise.

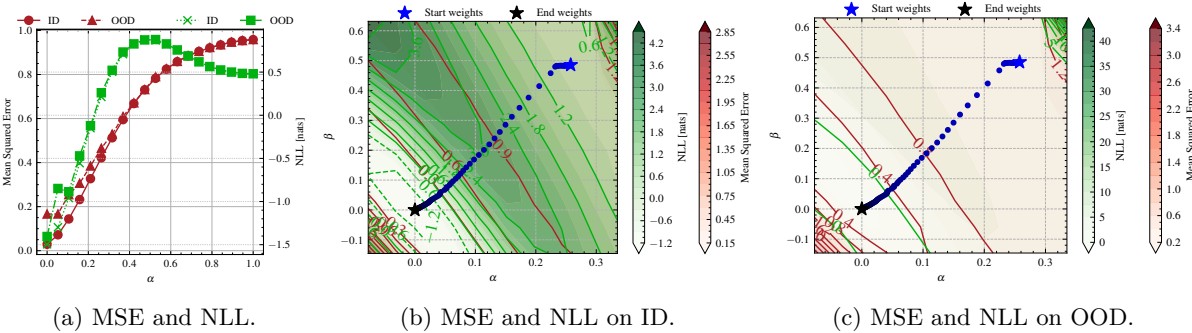

(a) MSE and NLL.     (b) MSE and NLL on ID.     (c) MSE and NLL on OOD.

Figure 40: No noise on Yacht.

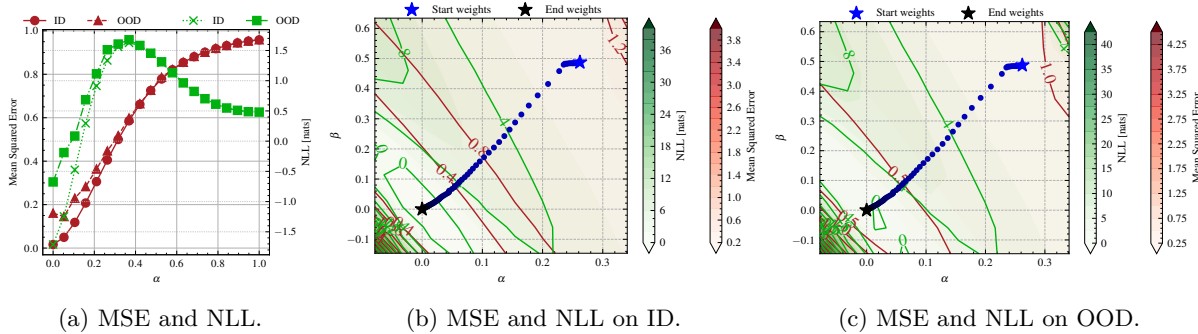

(a) MSE and NLL.

(b) MSE and NLL on ID.

(c) MSE and NLL on OOD.

Figure 41: Input Additive Gaussian on Yacht. *Observations*: While the shape of the 1D curves looks similar to no noise, the MSE and NLL magnitudes are different. The OOD NLL is substantially higher than the OOD NLL for no noise. The 2D plots demonstrate a wider landscape of feasible solutions than no noise.

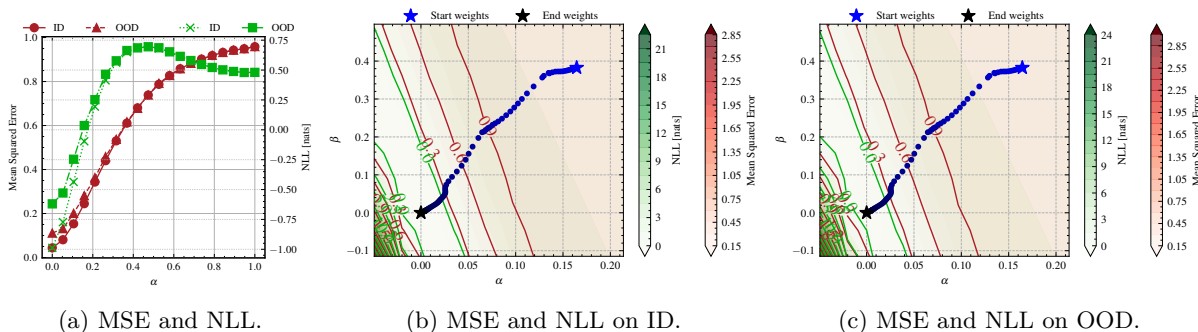

(a) MSE and NLL.

(b) MSE and NLL on ID.

(c) MSE and NLL on OOD.

Figure 42: Input-Target CMixUp on Yacht. *Observations*: While the shape of the 1D curves looks similar to no noise, the MSE and NLL magnitudes are different. The 2D plots demonstrate a wider landscape of feasible solutions than no noise.

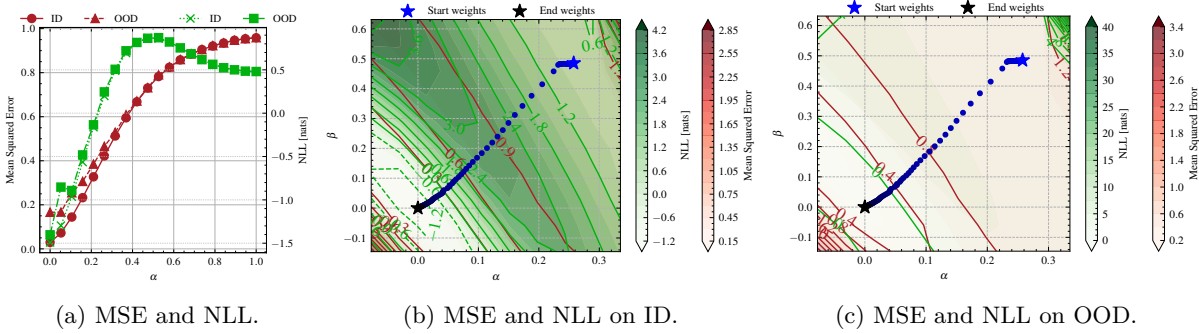

(a) MSE and NLL.

(b) MSE and NLL on ID.

(c) MSE and NLL on OOD.

Figure 43: Activation Additive Gaussian on Yacht. *Observations*: While the shape of the 1D curves looks similar to no noise, the MSE and NLL magnitudes are different. The 2D plots are close to the no-noise ones, showing marginal differences.

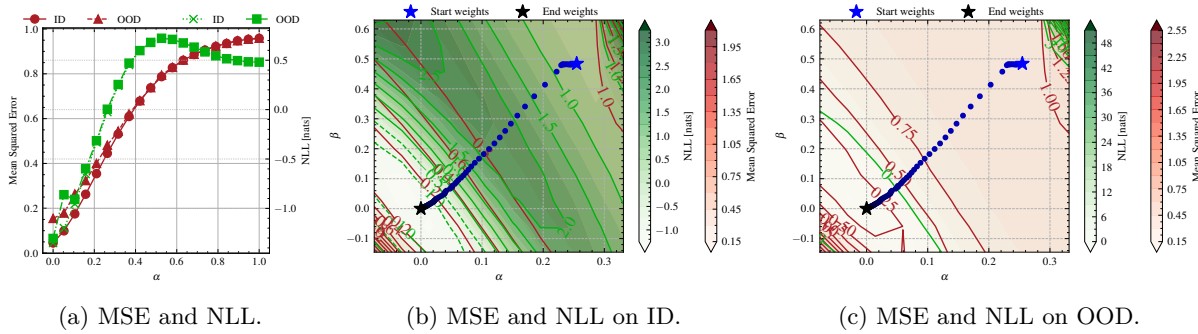

| (a) MSE and NLL. | (b) MSE and NLL on ID. | (c) MSE and NLL on OOD. |

Figure 44: Activation Dropout on Yacht. *Observations*: The 1D curves look similar to no noise but Dropout converged in a narrow valley, as demonstrated in the 2D plots.

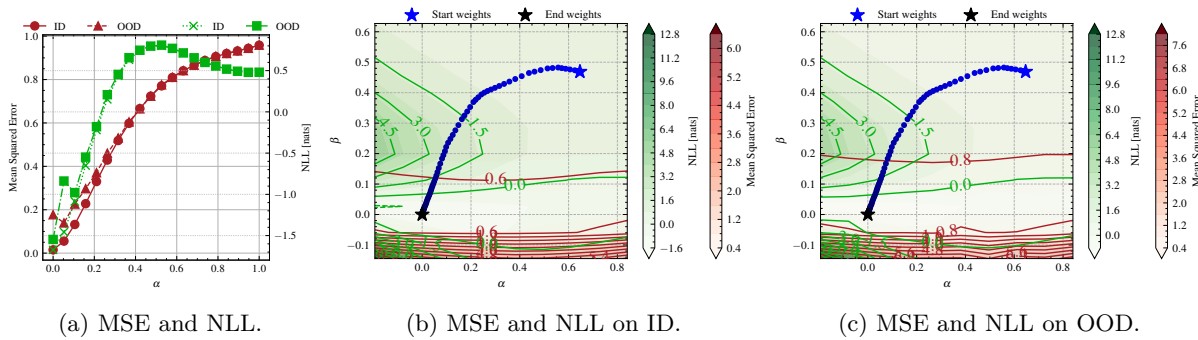

| (a) MSE and NLL. | (b) MSE and NLL on ID. | (c) MSE and NLL on OOD. |

Figure 45: Gradient Gaussian on Yacht. *Observations*: The 1D curves remained unchanged except for the magnitude of NLL or MSE. Nevertheless, the 2D plots show us that the optimisation trajectory significantly differed from no noise where the landscape of potential optimal solutions was wider.

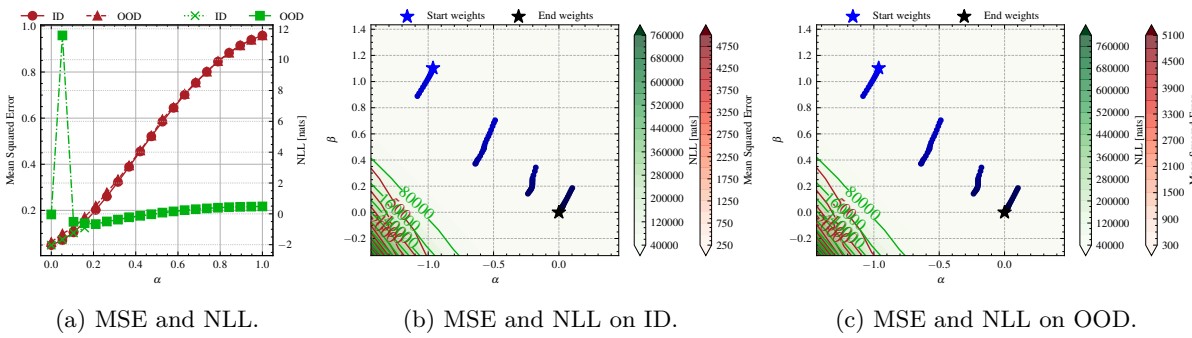

| (a) MSE and NLL. | (b) MSE and NLL on ID. | (c) MSE and NLL on OOD. |

Figure 46: Model Shrink and Perturb on Yacht. *Observations*: The model jumped between narrow valleys as seed in the 2D plots and the 1D plots show smoother behaviour from the OOD perspective for MSE but not NLL.

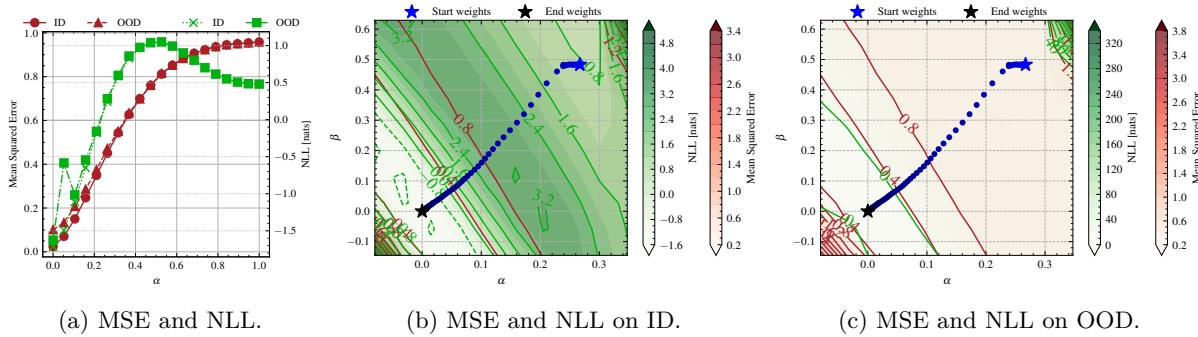

(a) MSE and NLL.  (b) MSE and NLL on ID.  (c) MSE and NLL on OOD.

Figure 47: Weight Additive Gaussian on Yacht. *Observations*: Did not change the smoothness of the 1D curves or the 2D trajectory.

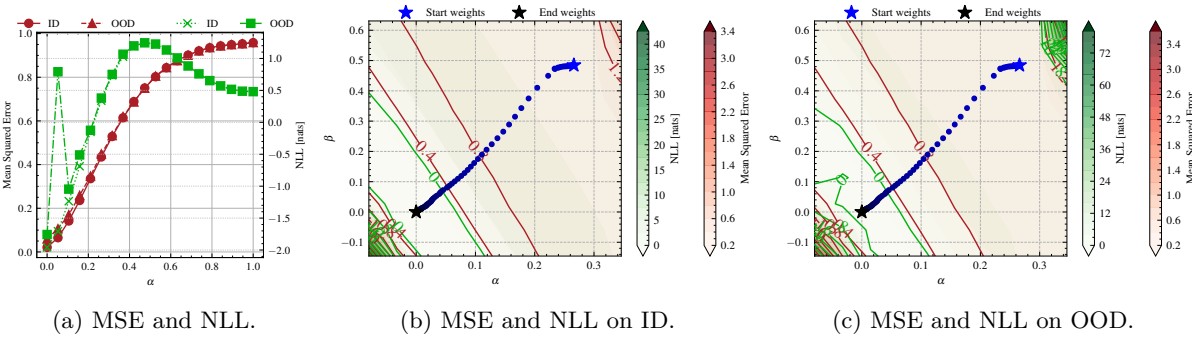

(a) MSE and NLL.  (b) MSE and NLL on ID.  (c) MSE and NLL on OOD.

Figure 48: Weight DropConnect on Yacht. *Observations*: While the shape of the 1D curves looks similar to no noise, the MSE and NLL magnitudes are different. The 2D plots demonstrate a wider landscape of feasible solutions than no noise.

