# OpenReview forum: "Navigating Noise: A Study of How Noise Influences Generalisation and Calibration of Neural Networks"
_TMLR — Accepted by TMLR_

### Review · Reviewer_QK5J · 2024-01-08

**Summary Of Contributions:**

Enhancing the generalization capabilities of neural networks (NNs) by incorporating noise, such as MixUp or Dropout, during training has become a potent and adaptable technique. Despite the established effectiveness of noise in NN training, a consensus is lacking regarding the optimal sources, types, and placements of noise that yield maximum benefits in generalization and confidence calibration. This study extensively explores diverse noise modalities to assess their impacts on NN generalization and calibration in both in-distribution and out-of-distribution settings. Additionally, experiments delve into the metric landscapes of the learned representations across various NN architectures, tasks, and datasets.

The research findings indicate that AugMix and weak augmentation demonstrate effectiveness across different tasks in computer vision, underscoring the importance of tailoring noise to specific domains. While it highlights the success of combining different noise sources and achieving hyperparameter transfer within a single domain, challenges arise in extending these benefits to other domains. Furthermore, the study emphasizes the intricacy of concurrently optimizing for both generalization and calibration, stressing the importance for practitioners to thoughtfully consider noise combinations and hyperparameter tuning to achieve optimal performance in specific tasks and datasets.

**Audience:**

Yes

**Claims And Evidence:**

Yes

**Requested Changes:**

- In the “Introduction”, this paper should give more details about the definition of noise.
- It is somewhat confusing about the target noise such as label smoothing, as it seems that this way does not introduce noise. Also, it would be better if the paper could add some discussions about label noise [R1-R2] as target noise.
- It seems that there is an overlap between different types of noise. For example, input-target noise includes target noise. A more accurate classification would be better.
- Could the different noise produce conflicts that finally hurt the model generalization?
- The green color is not clear in the algorithm flow.
- It seems that the proposed method of this paper sometimes does not work well. However, there is no clear description of this phenomenon. More details about this are expected to be provided.

----
[R1] Are Anchor Points Really Indispensable in Label-noise Learning? NeurIPS 2019.
[R2] Robust Early-learning: Hindering the Memorization of Noisy Labels. ICLR 2021.

**Strengths And Weaknesses:**

**Strengths**
- The motivation of this paper is strong.
- The writing and organization of this paper are overall good.
- Experimental results are overall comprehensive.

**Weaknesses**
- The technical details of the proposed method are not clear.
- There are some unclear descriptions, explanations, and justifications.

---

> ### Author Response · Authors · 2024-02-02
> **Response**
>
> We thank the reviewer for their valuable feedback, and we provide responses to the points raised. We especially appreciate that the reviewer recognises the motivation for our work, the comprehensiveness of experiments as well as quality of presentation.
>
> **Clarification on Noise Definition**: We have expanded the discussion on the definition of noise within the Related Work section on page 13 in the revised version to enhance the clarity and understanding of our study's context. This revision aims to comprehensively delineate the types and noise sources explored in our research.
>
> **Distinction Between Label Noise and Label Smoothing**: In response to the reviewer's comments, we clarified the distinction between label noise and label smoothing in the Related Work section on page 13 in the new paper version. We emphasise that our focus is on artificial noise—intentionally introduced during training—to investigate its impact on model performance. Conversely, natural noise, such as incorrect labels (label noise), represents a separate challenge. We have included references to seminal works on learning with noisy labels to guide readers' interest in the Related Work on page 2 in the revision.
>
> **Classification of Noise Types**: We have revised the classification of noise types in the Related Work section on page 13 in the revision to address the overlap noted by the reviewer. Specifically, we have separated input-target noise from target noise to prevent confusion and ensure a more precise taxonomy of noise types.
>
> **Potential Conflicts Among Different Noises**: Our findings, as illustrated in Figures 1, 3, and Table 3, generally show that combining noise sources can enhance model generalisation and calibration. Nonetheless, we acknowledge the potential for conflicts between noise types, which could detract from model performance if not carefully managed. We have elaborated on our methodology to describe how we mitigate such conflicts through tuning on page 4 in the revision, allowing for the adjustment of noise frequency and magnitude to prevent detrimental interactions.
>
> **Improvements to Algorithm Clarity**: To address the reviewer's point about the clarity of the green colour in the algorithm flow, we have revised the algorithm presentation on page 4 in the revised version to improve legibility and understanding. This includes more detailed explanations of the conditional application of noise and enhancements to the visual representation of the algorithm steps.
>
> **Performance Variability of Noise Types**: We recognise the reviewer's concern regarding instances where certain noise types underperform. Our study identifies variability in noise's effectiveness across different tasks, datasets, and architectures. This variability underscores the necessity of selecting appropriate noise types for specific contexts. Across all experiments, at least one noise type improves performance, demonstrating the potential benefits of incorporating noise in NN training. We have refined the Key Takeaways section on page 13 in the revised version of the paper to highlight the adaptability of our methodology, which accommodates a wide range of noise types and combinations to optimise performance across various scenarios.

---

### Review · Reviewer_EoNq · 2024-01-18

**Summary Of Contributions:**

This paper performs an extensive evaluation of an array of data augmentation and regularization techniques (termed as "noise" in the paper) on multiple datasets in the image, text and tabular data domains, on classification and regression tasks, in both in-domain and out-of-distribution settings. Both prediction performance and calibration are evaluated. Based on these results, the paper makes several general observations, for instance, that adding "noise" helps for the most part except on the tested image regression tasks and that combining multiple "noises" further improves performance. The relative ranking of the methods could be of interest to a practitioner.

**Audience:**

Yes

**Broader Impact Concerns:**

No concerns.

**Claims And Evidence:**

No

**Requested Changes:**

- Clarify the testing protocol - what data splits were used
- Perform more realistic OOD evaluation at least for tabular data, if not for all domains
- Clarify the motivation for simultaneously optimizing generalization and calibration
- Include additional datasets/tasks for text (e.g. sentiment analysis)
- Include more realistic regression tasks on image data (e.g. facial keypoint detection) (optional)
- Explain how the noise methods included were selected (optional)
- Provide further insight into why the different methods perform differently (optional)

**Strengths And Weaknesses:**

Strengths:
- The paper's experiments cover quite a range of "noise" methods spanning different types (input, labels, weights, gradients), and are evaluated over several domains and datasets from each domain. Hyper-parameter tuning is also done. Overall it is quite an effort to perform all the experiments.
- Investigation of transferability of hyperparameters is something quite useful for practitioners, though it would be more interesting to see whether there is a default set of hyperparameters that generally performs well, which could then serve as a starting point for further tuning.
- The submission includes code to reproduce the results, which is to be commended.

Weaknesses:
- The OOD evaluation for tabular data does not seem well-justified. Zeroing out features and adding noise could change the labels, unlike the common corruptions used for image data. More broadly, it is advised to use real-world distribution shifted data for a more realistic evaluation (e.g. WILDS - https://wilds.stanford.edu/datasets/). There is also no OOD analysis provided for the text-domain.
- Some of the datasets/tasks used in the experiments seem rather "toy", such that the conclusions obtained from these may not generalize in practice. In particular, the image regression task of rotation prediction and the NewsGroup text classification task; in addition for the text-domain only a single dataset is used.
- When combining different noises, using the top-k noise combinations requires knowing which of them is the top (i.e., tuning on all methods chosen) - how does this compare to doing a search over all combinations of methods but with the same tuning budget (# methods * budget per method)?
- It is unclear why there is an emphasis on simultaneously optimizing for generalization and calibration as calibration can be done post-hoc using various methods (e.g. temperature scaling).

Other comments:
- It is arguable whether data augmentation methods should be termed as "noise" or perturbations.
- Would be useful to explain how the methods used in the evaluation were selected
- Clarify the testing protocol - what train/validation/test splits were used?
- Some of the stated conclusions are not very surprising (e.g. "Noise injection methods can improve NN performance across various tasks and datasets" - this would be shown when the methods were published in the first place; "AugMix remained highly ranked for robustness" - AugMix was specifically designed for robustness)

Overall, the paper presents results from many experiments, but is largely descriptive and would likely benefit from being more focused on particular domains/tasks to provide more generalizable conclusions and to provide a more complete picture (e.g. lack of OOD analysis and only a single dataset for text data); more analysis/insight into the differences between methods would also enrich the paper.

---

> ### Author Response · Authors · 2024-02-02
> **Response (1/3)**
>
> We thank the reviewer for their valuable feedback, and we provide responses to the points raised. We especially appreciate that the reviewer recognises the findings could be of interest to a practitioner and that the experiments are extensive, with hyper-parameter tuning included.
>
> **OOD Evaluation for Tabular Data**: We acknowledge the reviewer's concern regarding the OOD evaluation for tabular data. While vision data has ImageNet-C, to our knowledge, there is no similar benchmark for tabular data. WILDS does not have tabular data. Our methodology for introducing noise and zeroing out features is designed to simulate a wide range of potential distribution shifts in real-world scenarios, such as instrumentation errors, missing data, and adversarial attacks. We crafted the OOD evaluation to be similar to ImageNet-C corruptions in terms of the magnitude and severity of the noise, which allows us to systematically evaluate the models' robustness. The techniques we employ can be easily replicated and applied by practitioners. We have added our justification for the OOD evaluation in the Experiments section on page 5 in the revised paper. Regarding the OOD evaluation for vision data, we have used ImageNet-C-style corruptions since pairings with the tested datasets are commonly used, and the datasets used were within our available computational budget.
>
> We have now adopted a specific strategy to enhance the quality of augmented tabular data and avoid incorrect label changes (label-flipping). We calculate a unique scaling factor for each dataset to adjust how much we augment the data, ensuring the changes are manageable. We use a K-nearest neighbour (KNN) classifier, specifically a 1-neighbour KNN, trained on a dataset's original, unmodified data. This classifier then predicts labels for the augmented (modified) data. We adjust the scaling factor for each dataset so that the KNN classifier's mean accuracy on the augmented data exceeds 99% or the mean squared error is less than 0.01. This approach ensures that the augmentations are subtle enough to maintain the integrity of the data, meaning the nearest neighbour—the closest match in the original dataset—remains the same, and the labels are not incorrectly changed. Under this regime, we have added new results for the OOD evaluation for tabular data, along with the description of the methodology in the revised version of the Appendix on page 18. The code will include an additional Jupyter notebook with the code demonstrating the process. Alternatively, we could remove the OOD evaluation for tabular data if the reviewer desires. Still, we believe the results are valuable to the community.
>
> **Selection of Datasets/Tasks**: We acknowledge the reviewer's concern regarding the "toyness" of the datasets used in the experiments. We have used publicly available datasets commonly used in the literature that fit into our computational budget since, as the reviewer admits, conducting all the presented experiments was resource-consuming. We have already mentioned this unavoidable limitation in the paper. Rotation prediction is a common task in the literature for self-supervised pre-training [1], and hence, we deemed it a reasonable task to evaluate. We expanded the motivation for the rotation prediction task in the Experiments setup on page 5 of the revision. We have expanded the text domain to include an evaluation of SST-2 with respect to both the transformer and the CNN architecture across all relevant experiments. Note that OOD evaluation for text requires suitable datasets, but these are larger in scale, making it challenging to integrate them into our evaluation within the given compute and time resources.

---

> ### Author Response · Authors · 2024-02-02
> **Response (2/3)**
>
> **Noise Combinations**: This paper aimed to identify the most promising noise methods so that future researchers/practitioners do not have to perform a full search similar to ours and could instead start with the top-k combinations. However, if the users wish to perform a full search, we have provided a methodology and code. If we were to consider all combinations of methods, e.g. for computer vision, there are 12 methods. Each of them has 1 real-valued hyperparameter for the frequency of application, and let's assume at least one additional magnitude hyperparameter for each method, giving at least 24 hyperparameters for joint tuning. We have empirically found that tuning that many hyperparameters is not feasible even with a larger computational budget. Hence, given the required computational budget, we deemed it reasonable to use the top-k combinations as a form of incremental improvement in which we take only one step (Top-2) or two steps at a time (Top-3) and retune or reuse the previously successful parameters. This is also what practitioners [3] would do unless they have a much larger computational budget. We have noted this in the respective sections and limitations of the revised paper on pages 9 and  14. Further, our results in the Combination of Noises section suggest that a significantly larger tuning budget would be needed when jointly tuning hyperparameters for several noises.
>
> **Optimising Generalisation and Calibration**: Besides generalisation, confidence calibration is a desirable model property, especially in safety-critical applications where confidence scores must be aligned with the model's accuracy to make informed decisions. As shown in the literature [2], achieving good calibration is not trivial, especially on out-of-distribution data where temperature scaling is often insufficient. Temperature scaling is also not supported for regression problems. Therefore, many methods are proposed in the literature to optimise for generalisation and calibration, among which we focused on noise injection methods. While focused on improving generalisation, e.g. MixUp, AugMix, label smoothing, Dropout, DropConnect, etc., these noises were also empirically shown to enhance the calibration of their respective publications. However, as we argue in the paper, the methods are usually compared only within their respective domains and datasets. Hence, we wanted to evaluate them in a more general setting. Additionally, we have considered further noises that were not previously evaluated for calibration. We wanted to explore whether they can improve calibration and if the noise needs to be applied to specific components of the training process. We have expanded the motivation for the calibration evaluation in the Introduction on page 1 in the revised version.

---

> ### Author Response · Authors · 2024-02-02
> **Response (3/3)**
>
> **Selection of Noise Methods**: The selection of noise methods was guided by a literature review - we aimed to encompass a diverse array of artificial noise types, each chosen for its potential to enhance model robustness and generalisation. Our extensive literature review identified the most fundamental noise types, e.g. Dropout, which introduced a novel paradigm in adding noise to the activations and their applications in the literature. This review helped us narrow down the candidates, from which we then chose popular representative examples of each noise injection category. We noted the selection process in the Related Work on page 2 in the revision.
>
> **Other Comments**: We opted to use noise for all the methods as it is a more general term encompassing both perturbations and augmentations. We have added the testing/validation/training splits in the Appendix for all tasks across pages 18 and 19 in the revised version of the paper. As we replied to reviewer QK5J, AugMix was chosen for its illustrative value in demonstrating the potential benefits of incorporating domain-specific inductive biases. We commented on this in the Key Takeaways on page 14 in the new version. Our study had to strike a balance between breadth and depth. Therefore, we opted for 2 separate tasks across 3 modalities and different network architectures. We included our analysis of Learnt Representation Landscapes to provide insight into the differences between methods. Our analysis provides a more qualitative comparison and demonstrates if the learned representations significantly differ between methods. We commented on every method’s landscape across two domains and four tasks, and as seen in the results, the empirical differences between many methods seem minor. We agree that more theoretical analysis would be beneficial. However, this is beyond this paper's scope and can be highly non-trivial, so we suggest it as a future direction on page 14 in the revision. Lastly, our Combination of Noises and Transferability of Hyperparameters Across Datasets and Models sections indicate that there is limited transferability of the hyperparameters, indicating that default parameters are rare.
>
> [1] Unsupervised Representation Learning by Predicting Image Rotations, ICLR 2018
>
> [2] Can You Trust Your Model's Uncertainty? Evaluating Predictive Uncertainty Under Dataset Shift, NeurIPS 2019
>
> [3] Deep Learning Tuning Playbook https://github.com/google-research/tuning_playbook

---

> ### Comment · Reviewer_EoNq · 2024-02-15
>
> Thanks to the authors for providing a comprehensive response and revision that has address some of my concerns. Here are some further comments:
>
> **OOD Evaluation for Tabular Data**: I appreciate the authors' efforts in refining the protocol, but unfortunately the lack of a way to actually validate the perturbations are not label changing significantly diminishes the value of the experiments. Some more realistic tabular OOD data can be found in the medical domain (e.g. see https://proceedings.mlr.press/v136/ulmer20a.html), as an example.
>
> **Selection of Datasets/Tasks**: My point was that the tasks should be selected to be sufficiently convincing in spite of the computational limitations. This would have, for example, involved focusing on a single domain but on realistic tasks to draw more generalizable conclusions. The utility of rotation prediction for self-supervision does not automatically make it a reasonable task for this work, as it is unlikely to be representative of real-world classification tasks.
>
> **Optimising Generalisation and Calibration**: Thanks for the clarification and revisions that address my concerns.
>
> **Noise Combinations**: Thanks for the clarification and revisions that address my concerns.
>
> **Selection of Noise Methods**: Thanks for the clarification and revisions that address my concerns.
>
> **Other Comments**:
> - I think the definition of "noise" as used in the paper should come in the introduction rather than related work as the version used in this paper is not a common definition of the term (e.g. reviewer QK5J's comments).
> - Analysis does not have to be theoretical, but can be the testing of various hypotheses about why some methods work better than others, that are then empirically tested.
> - On AugMix, I believe what reviewer bpxd and myself are pointing out is that AugMix was designed to be robust to corruptions and demonstrated to be so on the same datasets evaluated here in their paper. In this sense, this study does not add much new knowledge about AugMix's performance.

---

> > ### Author Response · Authors · 2024-02-15
> > **Response**
> >
> > Thank you very much for getting back to us. We appreciate that several of the points have been resolved. We give further clarifications to the remaining points raised.
> >
> > **OOD Evaluation for Tabular Data**: We acknowledge the reviewer's concern regarding the OOD evaluation for tabular data. In comparison to data from a medical domain, we aimed to use well-established UCI datasets to ensure that the results are reproducible and comparable to other works, also giving others the opportunity to easily build on top of our work without requiring a custom architecture or a complex training protocol, which could introduce additional confounding factors. However, no OOD data exists for these datasets, hence we had to simulate it. In the work shared by the reviewer, under Section 5.2 OOD experiments, the authors write that OOD data can be a result of a "corruption in the data due to machine or human error", and to simulate such OOD data, the authors proceed to create artificial corruptions in Section 5.2.1 by scaling the features randomly by a constant factor across multiple severities, mimicking the type of OOD evaluation we have performed. As we have stated in our previous response, the new corruptions have been specifically designed to simulate a wide range of potential distribution shifts in real-world scenarios, such as instrumentation errors, missing data, and adversarial attacks. We added a Jupyter notebook to demonstrate that by choosing an appropriate scaling factor per dataset for each corruption type, we can ensure that the corruptions are subtle enough, such that the nearest neighbour, arguably the simplest model, can still find the correct match in the original dataset. We chose the scaling factor such that the nearest neighbour classifier's mean accuracy on the augmented data exceeds 99% or the mean squared error is less than 0.01. We agree with the reviewer that unless observing this exact data in the wild, there is no way to validate that the perturbations are not label-changing. However, we believe we reasonably approximate the real-world distribution shifts with measured guarantees.
> >
> > If the reviewer still thinks the way we simulate OOD tabular data is not reasonable, we can always remove it with the explanation that OOD is easy to simulate only for CV data from the domains that we have considered.
> >
> > **Selection of Datasets/Tasks**: A related study could certainly focus on one domain, but comparing the impact of noise across multiple domains also leads to valuable insights and ones that cannot be derived if only one domain was used. More specifically, we observe that different noise types are needed for different domains - e.g. CV and tabular data benefit from very different types of noise. Even though the datasets we use are smaller in scale to make a study with our focus tractable, these are still commonly used datasets for machine learning research. The rotation prediction task can be seen as borderline in terms of practical usefulness. Still, it can be used to estimate better how different noises impact CV regression, as this task is an example of it.
> >
> > **Noise Definition**: We are happy to clarify this further in the paper and include a definition of noise in the introduction.
> >
> > **Analysis**: In order to analyse the differences between methods, we relied on the learnt representation landscapes. We hypothesised that the learnt representations would reflect the differences between methods. As demonstrated across image and tabular domains, the empirical differences between many methods seem minor. This indicates that there are more fundamental determinants of performance. Given that this study is the first to compare such a wide range of noise methods across different settings, we felt that our analysis is a solid foundation for future work to build on and investigate in-depth differences between subgroups of noise methods.
> >
> > **On AugMix**: We recognise your and reviewer bpxd's concerns regarding the novelty of our findings about AugMix's performance. Our intention was to evaluate AugMix in the broader context of our study, comparing it against uninformative noise methods and demonstrating that by considering inductive biases in generating the noise, e.g. corruptions or object/environment variations that would be present in the real world, the performance of the model can be improved more efficiently rather than by using other unconstrained noise methods. Our aim was not to add new knowledge about AugMix's performance against image corruptions, given its well-documented performance in previous works, but to demonstrate that inductive biases should be considered when generating noise for better performance.

---

> > > ### Comment · Reviewer_EoNq · 2024-02-16
> > >
> > > Thanks for getting back. I think for many of these remaining issues instead of performing additional experiments (that may not be feasible given the available resources), it suffices to explicitly acknowledge the limitations of the study and the generalizability of the results so that the claims are supported by the evidence presented in the paper, e.g. in terms of datasets used and how the results may be more realistic if done on real-world OOD data, the tasks used and their relevance to real-world tasks.

---

> > > > ### Author Response · Authors · 2024-02-16
> > > > **Response**
> > > >
> > > > Thank you again for your feedback and discussion. We will address/re-emphasise these points in the paper accordingly.

---

### Review · Reviewer_bpxd · 2024-01-21

**Summary Of Contributions:**

This paper investigates the impact of various noise injection techniques on the generalization and calibration of neural networks (NNs) across different tasks, datasets, and architectures. The study reveals that methods like AugMix, weak augmentation, and Gaussian noise enhance NN performance diversely, highlighting the importance of task-specific noise application. The research shows that combining different noises can be more effective than using them individually, particularly in classification tasks, while regression might benefit from a singular noise approach. Despite variations in effectiveness across tasks, the study confirms that noise injections contribute to improved performance both in-distribution (ID) and out-of-distribution (OOD), with notable differences in their efficiency across different tasks and datasets.

**Audience:**

Yes

**Broader Impact Concerns:**

none noted.

**Claims And Evidence:**

No

**Requested Changes:**

1. Recent studies have shown the limitation of evaluating a model with fixed benchmark [1], and as an example, AugMix, which is primarily designed for the noises it's introduced for, shows weaker performances when the evaluation is more dynamic. I will recommend the authors to consider such evaluation protocol for a more dynamic evaluation process.

2. more powerful methods should probably be in the discussion, e.g., DeepAug [2], there are many other follow-ups also, but probably less prominent.


[1] Foundation Model-oriented Robustness: Robust Image Model Evaluation with Pretrained Models.
[2] The many faces of robustness: A critical analysis of out-of-distribution generalization

**Strengths And Weaknesses:**

Strengths

+ The study has an interesting research aim, which aims to systematically investigate the effects of diverse noise injections on NN generalization and calibration across multiple datasets, tasks, and architectures. This problem is largely understudied.
+ The study draws specific conclusion on the practical usage of the noises, giving suggestions on how to use the model in practice.


Weakness

- While the ideal scope of this study is admirable, as the authors admit themselves, a practical study need to limit the study scope due to many practical limitations. For example, a limited opitimization scope that is tuned before hand, while different techniques are will likely benefit from different parameters, tuning strategies etc. Thus, the conclusions are less likely to be meaningful as in practice, because in practice, people who choose to use this method for a specific application is likely to tune heavily.

- The conclusion that AugMix is the most promising is no surprising and likely misleading. The OOD datasets the authors choose to evaluate is where AugMix is defined for. In fact, there are many other datasets that AugMix is no longer the best performing one.

- There are also more powerful augmentations, such as DeepAug, that are not compared by the authors.

---

> ### Author Response · Authors · 2024-02-02
> **Response**
>
> We thank the reviewer for their valuable feedback, and we provide responses to the points raised. We especially appreciate that the reviewer recognizes the motivation for our work as well as that we draw specific and practical conclusions.
>
> **On the Optimization Budget and Scope Limitation**: We acknowledge the reviewer's concern regarding the limited optimisation scope necessitated by practical constraints, including computational costs. Our study, spanning multiple domains, datasets, and noise injection methods, required us to make pragmatic decisions regarding the optimisation budget. We concur with the reviewer's observation that our research provides a valuable starting point for practitioners by highlighting promising noise sources for further exploration and optimisation in the new version of Key Takeaways on page 13. Our work demonstrates, even with the limited optimisation budget, that some noise methods are more accessible, enabling a practitioner to at least predict the scale of the potential benefits and how much effort is required to achieve them. For example, AugMix, incorporating domain-specific inductive biases, was transferable and effective despite a limited budget. To address this, we have reiterated the Limitations on page 14 in the revision regarding computational resources in our paper and emphasised the utility of our findings for practitioners as a preliminary guide.
>
> **Regarding the Evaluation of AugMix**: We appreciate the opportunity to clarify our stance on AugMix. Contrary to the perception that AugMix directly overlaps with ImageNet-C augmentations, we reference a statement from the AugMix paper: "Crucially, we exclude operations which overlap with ImageNet-C corruptions." This supports our belief in the fairness and relevance of our evaluation methodology. Our study's primary aim was not to surpass state-of-the-art results but to systematically assess the impact of prevalent noise injection techniques on model generalisation and calibration. AugMix was chosen for its illustrative value in demonstrating the potential benefits of incorporating domain-specific inductive biases. Nonetheless, we recognise the evolving landscape of noise injection methods and, in response to the reviewer's suggestion, have included DeepAug in the revised Future Directions on page 14 as a representative of advanced developments in this area, and we improved the scoping of our paper in the Introduction on page 2 in the revised version.
>
> **On Out-of-Distribution (OOD) Evaluation**: We conducted our OOD evaluations using the ImageNet-C benchmark, a peer-reviewed and widely recognised standard in the field. Our objective drove this choice to ensure a rigorous and widely applicable assessment of the noise injection methods under study.

---

### Author Response · Authors · 2024-02-02
**Overall Response**

We are grateful to the reviewers for reviewing our paper and providing valuable suggestions for improving it. In particular, we appreciate that the reviewers recognise the strong motivation for our paper, the comprehensiveness of experiments, and highlighting that we have specific practical conclusions.

We have improved our paper based on the reviewers' suggestions, with the revised parts of the paper in blue. In particular, we add the following and recompute the results accordingly:
* Additional NLP dataset: sentiment classification using SST-2
* Improved OOD evaluation for tabular data

We also provide detailed responses to the questions from each reviewer. We are happy to engage in further discussion and address additional reviewers' questions/clarification requests.  We hope our responses and the improvements to the paper, give reasonable justifications to recommend acceptance for our paper.

---

### Decision · Action_Editor_rLQx · 2024-03-03

**Recommendation:** Accept with minor revision

**Comment:**

This paper has mixed opinions. Most negative opinions are from the fundamental limitations of the study. In my opinion, two claims can be critical "AugMix noises are exactly defined on the target OOD dataset (which is hard to say "OOD")" and "the evaluation is conducted on the fixed benchmark". One possible direction to resolve this concern is an additional OOD experiment on completely unknown distribution shifts. For example, one can use adversarial OOD samples [1] or the NINCO (No ImageNet Class Objects) dataset [2] for a fairer evaluation. I suggest to examine these datasets (especially [2]) for enhancing the results.

- [1] NeurIPS 2020. Certifiably Adversarially Robust Detection of Out-of-Distribution Data https://github.com/j-cb/GOOD
- [2] ICML 2023. In or Out? Fixing ImageNet Out-of-Distribution Detection Evaluation https://github.com/j-cb/NINCO

Note that NINCO [2] is only for ImageNet-1k classifiers. Hence, it would be better to resize NINCO images and evaluate them using TinyImageNet classifiers. I also clarify that it is still okay if the NINCO results (or GOOD results) are not perfectly aligned with the original results, because there are other experiments (e.g., Tabular, NLP classification and Tabular regression) that show consistent results. Plus, I think it is an optional experiment to resolve the concerns raised by the reviewers.

Another concern from me (not raised by the reviewers) is that similar studies with similar settings already exist. In my opinion, it is better to cite the previous studies and add a discussion of the difference between the previous studies and this study. In my point of view, using various datasets is a valuable contribution compared to the previous studies.

- [3] NeurIPS 2018. Generalisation in humans and deep neural networks
- [4] ICML workshop 2019. Transfer of Adversarial Robustness Between Perturbation Types
- [5] ICML workshop 2019. An Empirical Evaluation on Robustness and Uncertainty of Regularization methods
- [6] NeurIPS 2020. Measuring robustness to natural distribution shifts in image classification

In summary, I suggest to the authors revise the paper by (1) adding a discussion of the difference between the previous studies and this study and (2) considering to add more OOD experiments to resolve the concerns regarding to AugMix. I think these changes will be minor modifications that can be sufficiently handled by a minor revision.

**Audience:**

The relationship between regularization methods (including augmentations) and the generalizability of neural networks is a widely studied field. I believe that there are audiences who are interested in this paper's findings.

**Claims And Evidence:**

This paper investigates the effect of noise injection methods on neural network (NN) generalization and calibration across diverse datasets, tasks, and architectures for both in-domain and out-of-distribution scenarios. Based on the observations, this paper claims that using various noises is more effective than single noise injection. Also, this paper shows that noise injection can improve both in-domain and out-of-distribution scenarios.

As the comments by the reviewers, this study has many practical limitations. For example, the limited optimization scope, AugMix noises are exactly defined on the target OOD dataset (which is hard to say "OOD"), and the evaluation is conducted on the fixed benchmark.

However, I think that the paper's claim is well-supported in the limited experimental setting. Although there are many relevant studies (especially for vision datasets -- see the "Comment" section for details), this paper extends the previous observations to the various domains, such as CV classification and regression, tabular data classification and regression, and NLP classification.

---

> ### Author Response · Authors · 2024-03-26
> **Response**
>
> Thank you very much for handling our paper, and we appreciate the decision to accept our paper with minor revision. We have updated the paper with 1.) further discussion of the differences compared to earlier studies and this study, 2.) extended discussion of limitations as suggested by Reviewer EoNq and 3.) additional OOD experiments.
>
> **References**: We have added references [3-5] to the Related Work section and also highlighted the differences between our work and the previous studies in the revised version of the paper on pages 3-4. Specifically, we reference [6] in connection to the limitations of the study and the practical implications of the findings connected to out-of-domain evaluation and transferability between synthetic and real-world domain shifts on page 15 of the revised paper.
>
> **Limitations, Clarifications and Future Work**: We have added a comment on the evaluation of AugMix in the revised version of the paper on page 14, clarifying the relevance of AugMix to the study and its illustrative value in demonstrating the potential benefits of incorporating domain-specific inductive biases. We commented on the observed minor differences between noise methods in the learned representation landscapes and suggested that future work could delve deeper into the theoretical analysis of these differences on page 15 in the revised version of the paper. We added a definition of noise in the introduction to clarify the terminology used in the paper, directly on page 2 in the revised version. We have added the requested comment about the limitation of the empirical guarantee of the out-of-domain evaluation for tabular data to the revised version of the paper on page 19. Lastly, we emphasise that our evaluation was done on experimental data, suggesting that larger-scale datasets could be used for further validation in the future on page 14 in the revised version of the paper.
>
> **Out-of-Domain Evaluation**:  For additional OOD experiments, we added evaluation of TinyImageNet pre-trained models on a subset of ImageNet-Sketch (1) as it represents a significantly larger domain shift compared to the earlier. We selected a subset of 200 classes, shared between TinyImageNet and ImageNet-Sketch, giving us around 10000 samples, resized them to 64x64 and evaluated the models pre-trained on TinyImageNet on this subset of ImageNet-Sketch. The results highlight AugMix maintains its strong performance even if the domain shift is stronger than in our main OOD evaluation. We include these results as part of the Appendix (Tables 9-12) on pages 23-24, and we also include a short discussion in Section 4.3 of the main text. We note that AugMix augmentations are distinct from the ones we test within OOD, so that evaluation is valid, even if it is not as challenging for AugMix due to using related augmentations during training. We carefully read the referenced papers [1,2] but found they focus on shifts in terms of classes or adversarial attacks, while we evaluate shifts in terms of domains and assume the classes remain the same. Adversarial attacks could be considered as a special case of OOD evaluation, but the evaluation of ImageNet-Sketch gives a more natural example of a stronger domain shift.
>
> Please let us know if you have any questions and we will be happy to answer.
>
> (1) Wang, Haohan, et al. "Learning robust global representations by penalizing local predictive power." Advances in Neural Information Processing Systems 32 (2019).